# A Theoretical Analysis of Deep Q-Learning

## Abstract

Despite the great empirical success of deep reinforcement learning, its theoretical foundation is less well understood. In this work, we make the first attempt to theoretically understand the deep Q-network (DQN) algorithm (Mnih et al., 2015) from both algorithmic and statistical perspectives. In specific, we focus on the fitted Q iteration (FQI) algorithm with deep neural networks, which is a slight simplification of DQN that captures the tricks of experience replay and target network used in DQN. Under mild assumptions, we establish the algorithmic and statistical rates of convergence for the action-value functions of the iterative policy sequence obtained by FQI. In particular, the statistical error characterizes the bias and variance that arise from approximating the action-value function using deep neural network, while the algorithmic error converges to zero at a geometric rate. As a byproduct, our analysis provides justifications for the techniques of experience replay and target network, which are crucial to the empirical success of DQN. Furthermore, as a simple extension of DQN, we propose the Minimax-DQN algorithm for zero-sum Markov game with two players, which is deferred to the appendix due to space limitations.

## 1 Introduction

Reinforcement learning (RL) attacks the multi-stage decision-making problems by interacting with the environment and learning from the experiences. With the breakthrough in deep learning, deep reinforcement learning (DRL) demonstrates tremendous success in solving highly challenging problems, such as the game of Go (Silver et al., 2016; 2017), robotics (Kober & Peters, 2012), and dialogue systems (Chen et al., 2017). In DRL, the value or policy functions are often represented as deep neural networks and the related deep learning techniques can be readily applied. For example, deep Q-network (DQN) (Mnih et al., 2015), asynchronous advantage actor-critic (A3C) and (Mnih et al., 2016) demonstrate superhuman performance in various applications and become standard algorithms for artificial intelligence.

Despite its great empirical success, there exists a gap between the theory and practice of DRL. In particular, most existing theoretical work on reinforcement learning focuses on the tabular case where the state and action spaces are finite, or the case where the value function is linear. Under these restrictive settings, the algorithmic and statistical perspectives of reinforcement learning are well-understood via the tools developed for convex optimization and linear regression. However, in presence of nonlinear function approximators such as deep neural network, the theoretical analysis of reinforcement learning becomes intractable as it involves solving a highly nonconvex statistical optimization problem.

To bridge such a gap in DRL, we make the first attempt to theoretically understand DQN, which can be cast as an extension of the classical Q-learning algorithm (Watkins & Dayan, 1992) that uses deep neural network to approximate the action-value function. Although the algorithmic and statistical properties of the classical Q-learning algorithm are well-studied, theoretical analysis of DQN is highly challenging due to its differences in the following two aspects.

First, in online gradient-based temporal-difference reinforcement learning algorithms, approximating the action-value function often leads to instability. Baird (1995) proves that this is the case even with linear function approximation. The key technique to achieve stability in DQN is experience replay (Lin, 1992; Mnih et al., 2015). In specific, a replay memory is used to store the trajectory of the Markov decision process (MDP). At each iteration of DQN, a mini-batch of states, actions, rewards, and next states are sampled from the replay memory as observations to train the Q-network, which approximates the action-value function. The intuition behind experience replay is to achieve stability by breaking the temporal dependency among the observations used in the training of the deep neural network.

Second, in addition to the aforementioned Q-network, DQN uses another neural network named target network to obtain an unbiased estimator of the mean-squared Bellman error used in training the Q-network. The target network is synchronized with the Q-network after each period of iterations, which leads to a coupling between the two networks. Moreover, even if we fix the target network and focus on updating the Q-network, the subproblem of training a neural network still remains less well-understood in theory.

In this paper, we focus on a slight simplification of DQN, which is amenable to theoretical analysis while fully capturing the above two aspects. In specific, we simplify the technique of experience replay with an independence assumption, and focus on deep neural networks with rectified linear units (ReLU) (Nair & Hinton, 2010) and large batch size. Under this setting, DQN is reduced to the neural fitted Q-iteration (FQI) algorithm (Riedmiller, 2005) and the technique of target network can be cast as the value iteration. More importantly, by adapting the approximation results for ReLU networks to the analysis of Bellman operator, we establish the algorithmic and statistical rates of convergence for the iterative policy sequence obtained by DQN. As shown in the main results in §3, the statistical error characterizes the bias and variance that arise from approximating the action-value function using neural network, while the algorithmic error geometrically decays to zero as the number of iteration goes to infinity.

Our contribution is two-fold. First, we establish the algorithmic and statistical errors of the neural FQI algorithm, which can be viewed as a slight simplification of DQN. Under mild assumptions, our results show that the proposed algorithm obtains a sequence of Q-networks that geometrically converges to the optimal action-value function up to an intrinsic statistical error induced by the approximation bias of ReLU network and finite sample size. Second, as a byproduct, our analysis justifies the techniques of experience replay and target network used in DQN, where the latter can be viewed as the value iteration. In addition, we also extend our algorithm to zero-sum Markov games. Due to space limit, we defer these results to the appendix.

**Notation.** For a measurable space with domain $\mathcal{S}$, we denote by $\mathcal{B}(\mathcal{S}, V)$ the set of measurable functions on $\mathcal{S}$ that are bounded by $V$ in absolute value. Let $\mathcal{P}(\mathcal{S})$ be the set of all probability measures over $\mathcal{S}$. For any $\nu \in \mathcal{P}(\mathcal{S})$ and any measurable function $f: \mathcal{S} \to \mathbb{R}$, we denote by $\|f\|_{p,\nu}$ the $\ell_p$-norm of $f$ with respect to measure $\nu$ for $p \geq 1$. In addition, for simplicity, we write $\|f\|_\nu$ for $\|f\|_{2,\nu}$. In addition, let $\{f(n), g(n)\}_{n \geq 1}$ be two positive series. We write $f(n) \lesssim g(n)$ if there exists a constant $C$ such that $f(n) \leq C \cdot g(n)$ for all $n$ larger than some $n_0 \in \mathbb{N}$. In addition, we write $f(n) \asymp g(n)$ if $f(n) \lesssim g(n)$ and $g(n) \lesssim f(n)$.

## 2 BACKGROUND

In this section, we introduce the background. We first lay out the formulation of the reinforcement learning problem, and then define the family of ReLU neural networks.

### 2.1 REINFORCEMENT LEARNING

A discounted Markov decision process is defined by a tuple $(\mathcal{S}, \mathcal{A}, P, R, \gamma)$. Here $\mathcal{S}$ is the set of all states, which can be countable or uncountable, $\mathcal{A}$ is the set of all actions, $P: \mathcal{S} \times \mathcal{A} \to \mathcal{P}(\mathcal{S})$

is the Markov transition kernel, $R\colon \mathcal{S} \times \mathcal{A} \to \mathcal{P}(\mathbb{R})$ is the distribution of the immediate reward, and $\gamma \in (0,1)$ is the discount factor. In specific, upon taking any action $a \in \mathcal{A}$ at any state $s \in \mathcal{S}$, $P(\cdot \,|\, s, a)$ defines the probability distribution of the next state and $R(\cdot \,|\, s, a)$ is the distribution of the immediate reward. Moreover, for regularity, we further assume that $\mathcal{S}$ is a compact subset of $\mathbb{R}^d$ which can be infinite, $\mathcal{A} = \{a_1, a_2, \ldots, a_M\}$ has finite cardinality $M$, and the rewards are uniformly bounded by $R_{\max}$, i.e., $R(\cdot \,|\, s, a)$ is supported on $[-R_{\max}, R_{\max}]$ for any $s \in \mathcal{S}$ and $a \in \mathcal{A}$.

A policy $\pi\colon \mathcal{S} \to \mathcal{P}(\mathcal{A})$ for the MDP maps any state $s \in \mathcal{S}$ to a probability distribution $\pi(\cdot \,|\, s)$ over $\mathcal{A}$. For policy $\pi$, the corresponding value function $V^\pi\colon \mathcal{S} \to \mathbb{R}$ is defined as the cumulative discounted reward obtained by when the actions are executed according to $\pi$, that is,

$$V^\pi(s) = \mathbb{E}\left[\sum_{t=0}^{\infty} \gamma^t \cdot R_t \,\bigg|\, S_0 = s, A_t \sim \pi(\cdot \,|\, S_t), S_{t+1} \sim P(\cdot \,|\, S_t, A_t)\right]. \tag{2.1}$$

Similarly, the action-value function $Q^\pi \colon \mathcal{S} \times \mathcal{A} \to \mathbb{R}$ is defined as

$$Q^\pi(s,a) = \mathbb{E}\left[\sum_{t=0}^{\infty} \gamma^t \cdot R_t \,\bigg|\, S_0 = s, A_0 = a, A_t \sim \pi(\cdot \,|\, S_t), S_{t+1} \sim P(\cdot \,|\, S_t, A_t)\right]. \tag{2.2}$$

For any given action-value function $Q\colon \mathcal{S} \times \mathcal{A} \to \mathbb{R}$, define the one-step greedy policy $\pi_Q$ as any policy that selects the action with the largest $Q$-value, that is, for any $s \in \mathcal{S}$, $\pi_Q(\cdot \,|\, s)$ satisfies

$$\pi_Q(a \,|\, s) = 0 \text{ if } Q(s,a) \neq \max_{a' \in \mathcal{A}} Q(s, a'). \tag{2.3}$$

Moreover, we define operator $P^\pi$ by

$$(P^\pi Q)(s,a) = \mathbb{E}\big[Q(S', A') \,\big|\, S' \sim P(\cdot \,|\, s, a), A' \sim \pi(\cdot \,|\, S')\big], \tag{2.4}$$

and define the Bellman operator $T^\pi$ by $(T^\pi Q)(s,a) = r(s,a) + \gamma \cdot (P^\pi Q)(s,a)$, where $r(s,a) = \int r R(\mathrm{d}r \,|\, s, a)$ is the expected reward obtained at state $s$ when taking action $a$. Then it can be verified that $Q^\pi$ is the unique fixed point of $T^\pi$.

The goal of reinforcement learning is to find the optimal policy, which achieves the largest cumulative reward. To characterize optimality, we define optimal action-value function $Q^*$ as

$$Q^*(s,a) = \sup_{\pi} Q^\pi(s,a), \tag{2.5}$$

where the supremum is taken over all policies. Based on $Q^*$, we define the optimal policy $\pi^*$ as any policy that is greedy with respect to $Q^*$. It can be shown that $Q^* = Q^{\pi^*}$. Finally, we define the Bellman optimality operator $T$ via

$$(TQ)(s,a) = r(s,a) + \gamma \cdot \mathbb{E}\big[\max_{a' \in \mathcal{A}} Q(S', a') \,\big|\, S' \sim P(\cdot \,|\, s, a)\big]. \tag{2.6}$$

Then we have Bellman optimality equation $TQ^* = Q^*$.

## 2.2 Deep Neural Network

We study the performance of DQN with rectified linear unit (ReLU) activation function $\sigma(u) = \max(u, 0)$. For any positive integer $L$ and $\{d_j\}_{j=0}^{L+1} \subseteq \mathbb{N}$, a ReLU network $f\colon \mathbb{R}^{d_0} \to \mathbb{R}^{d_{L+1}}$ with $L$ hidden layers and width $\{d_j\}_{j=0}^{L+1}$ is of form

$$f(x) = W_{L+1}\sigma(W_L\sigma(W_{L-1}\ldots\sigma(W_2\sigma(W_1 x + v_1) + v_2)\ldots v_{L-1}) + v_L), \tag{2.7}$$

where $W_\ell \in \mathbb{R}^{d_\ell \times d_{\ell-1}}$ and $v_\ell \in \mathbb{R}^{d_\ell}$ are the weight matrix and the shift vector in the $\ell$-th layer, respectively. Here we apply $\sigma$ to to each entry of its argument in (2.7). In deep learning, the network structure is fixed, and the goal is to learn the network parameters (weights) $\{W_\ell, v_\ell\}_{\ell \in [L+1]}$ with the convention that $v_{L+1} = 0$. For deep neural networks, the number of parameters greatly exceeds the input dimension $d_0$. To restrict the model class, we focus on the class of ReLU networks where most parameters are zero.

**Definition 2.1** (Sparse ReLU Network). For any $L, s \in \mathbb{N}$, $\{d_j\}_{j=0}^{L+1} \subseteq \mathbb{N}$, and $V > 0$, the family of sparse ReLU networks bounded by $V$ with $L$ hidden layers, network width $d$, and weight sparsity $s$ is defined as

$$\mathcal{F}(L, \{d_j\}_{i=0}^{L+1}, s, V) = \left\{ f \colon \max_{\ell \in [L+1]} \|\widetilde{W}_\ell\|_\infty \le 1, \sum_{\ell=1}^{L+1} \|\widetilde{W}_\ell\|_0 \le s, \max_{j \in [d_{L+1}]} \|f_j\|_\infty \le V \right\}, \quad (2.8)$$

where we denote $(W_\ell, v_\ell)$ by $\widetilde{W}_\ell$. Moreover, $f$ in (2.8) is expressed as in (2.7), and $f_j$ is the $j$-th component of $f$.

Here we focus on functions that are uniformly bounded because the value functions in (2.1) and (2.2) are always bounded by $V_{\max} = R_{\max}/(1 - \gamma)$. In the sequel, we write $\mathcal{F}(L, \{d_j\}_{j=0}^{L+1}, s, V_{\max})$ as $\mathcal{F}(L, \{d_j\}_{j=0}^{L+1}, s)$ to simplify the notation. In addition, we restrict the networks weights to be sparse, i.e., $s$ is much smaller compared with the total number of parameters. Such an assumption implies that the network has sparse connections, which are useful for applying deep learning in memory-constrained situations such as mobile devices (Han et al., 2015; Liu et al., 2015).

Moreover, we introduce the notion of Hölder smoothness as follows, which is a generalization of Lipschitz continuity, and is widely used to characterize the regularity of functions.

**Definition 2.2** (Hölder Smooth Function). Let $\mathcal{D}$ be a compact subset of $\mathbb{R}^r$, where $r \in \mathbb{N}$. We define the set of Hölder smooth functions on $\mathcal{D}$ as

$$\mathcal{C}_r(\mathcal{D}, \beta, H) = \left\{ f \colon \mathcal{D} \to \mathbb{R} \colon \sum_{\boldsymbol{\alpha} \colon |\boldsymbol{\alpha}| < \beta} \|\partial^{\boldsymbol{\alpha}} f\|_\infty + \sum_{\boldsymbol{\alpha} \colon \|\boldsymbol{\alpha}\|_1 = \lfloor \beta \rfloor} \sup_{x,y \in \mathcal{D}, x \neq y} \frac{|\partial^{\boldsymbol{\alpha}} f(x) - \partial^{\boldsymbol{\alpha}}(y)|}{\|x - y\|_\infty^{\beta - \lfloor \beta \rfloor}} \le H \right\},$$

where $\beta > 0$ and $H > 0$ are parameters and $\lfloor \beta \rfloor$ is the largest integer no greater than $\beta$. In addition, here we use the multi-index notation by letting $\boldsymbol{\alpha} = (\alpha_1, \ldots, \alpha_r)^\top \in \mathbb{N}^r$, and $\partial^{\boldsymbol{\alpha}} = \partial^{\alpha_1} \ldots \partial^{\alpha_r}$.

Finally, we conclude this section by defining functions that can be written as a composition of multiple Hölder functions, which captures complex mappings in real-world applications such as multi-level feature extraction.

**Definition 2.3** (Composition of Hölder Functions). Let $q \in \mathbb{N}$ and $\{p_j\}_{j \in [q]} \subseteq \mathbb{N}$ be integers, and let $\{a_j, b_j\}_{j \in [q]} \subseteq \mathbb{R}$ such that $a_j < b_j$ $j \in [q]$. Moreover, let $g_j \colon [a_j, b_j]^{p_j} \to [a_{j+1}, b_{j+1}]^{p_{j+1}}$ be a function, $\forall j \in [q]$. Let $(g_{jk})_{k \in [p_{j+1}]}$ be the components of $g_j$, and we assume that each $g_{jk}$ is Hölder smooth, and depends on at most $t_j$ of its input variables, where $t_j$ could be much smaller than $p_j$, i.e., $g_{jk} \in \mathcal{C}_{t_j}([a_j, b_j]^{t_j}, \beta_j, H_j)$. Finally, we denote by $\mathcal{G}(\{p_j, t_j, \beta_j, H_j\}_{j \in [q]})$ the family of functions that can be written as compositions of $\{g_j\}_{j \in [q]}$, with the convention that $p_{q+1} = 1$. That is, for any $f \in \mathcal{G}(\{p_j, t_j, \beta_j, H_j\}_{j \in [q]})$, we can write

$$f = g_q \circ g_{q-1} \circ \ldots \circ g_2 \circ g_1,$$

with $g_{jk} \in \mathcal{C}_{t_j}([a_j, b_j]^{t_j}, \beta_j, H_j)$ for each $k \in [p_{j+1}]$ and $j \in [q]$.

## 3 UNDERSTANDING DEEP Q-NETWORK

In the DQN algorithm, a deep neural network $Q_\theta \colon \mathcal{S} \times \mathcal{A} \to \mathbb{R}$ is used to approximate $Q^*$, where $\theta$ is the parameter. For completeness, we state the DQN as Algorithm 2 in §A. As shown in the experiments in Mnih et al. (2015), two tricks are pivotal for the success of DQN.

First, DQN use the trick of experience replay (Lin, 1992). Specifically, at each time $t$, we store the transition $(S_t, A_t, R_t, S_{t+1})$ into the replay memory $\mathcal{M}$, and then sample a minibatch of independent samples from $\mathcal{M}$ to train the neural network via stochastic gradient descent. Since the trajectory of MDP has strong temporal correlation, the goal of experience replay is to obtain uncorrelated samples, which yields accurate gradient estimation for the stochastic optimization problem.

Another trick is to use a target network $Q_{\theta^\star}$ with parameter $\theta^\star$. Specifically, with independent samples $\{(s_i, a_i, r_i, s_i')\}_{i \in [n]}$ from the replay memory, to update the parameter $\theta$ of the Q-network,

we compute the target $Y_i = r_i + \gamma \cdot \max_{a \in \mathcal{A}} Q_{\theta^\star}(s_i', a)$, and update $\theta$ by the gradient of

$$L(\theta) = \frac{1}{n} \sum_{i=1}^{n} [Y_i - Q_\theta(s_i, a_i)]^2. \tag{3.1}$$

Whereas parameter $\theta^\star$ is updated once every $T_{\text{target}}$ steps by letting $\theta^\star = \theta$. That is, the target network is hold fixed for $T_{\text{target}}$ steps, and is thus updated in a slower pace.

To demystify DQN, it is crucial to understand the role played by these two tricks. For experience replay, in practice, the replay memory size is usually very large. For example, the replay memory size is $10^6$ in Mnih et al. (2015). Moreover, DQN use the $\epsilon$-greedy policy, which enables exploration over $\mathcal{S} \times \mathcal{A}$. Thus, when the replay memory is large, experience replay is close to sampling independent transitions from an explorative policy. This reduces the variance of the $\nabla L(\theta)$, which is used to update $\theta$.

Thus, experience replay stabilizes the training of DQN, which benefits the algorithm in terms of computation. To understand the statistical property of DQN, we replace the experience replay by sampling independent transitions from a fixed distribution $\sigma \in \mathcal{P}(\mathcal{S} \times \mathcal{A})$. That is, instead of sampling from the replay memory, we sample i.i.d. observations $\{(S_i, A_i)\}_{i \in [n]}$ from $\sigma$. Moreover, for any $i \in [n]$, let $R_i$ and $S_i'$ be the immediate reward and the next state when taking action $A_i$ at state $S_i$. Under this setting, we have $\mathbb{E}(Y_i \mid S_i, A_i) = (TQ_{\theta^\star})(S_i, A_i)$, where $Q_{\theta^\star}$ is the target network, which, as we show as follows, is motivated from a statistical consideration.

Let us first neglect the target network and set $\theta^\star = \theta$. Using bias-variance decomposition, the the expected value of $L(\theta)$ in (3.1) is

$$\mathbb{E}[L(\theta)] = \|Q_\theta - TQ_\theta\|_\sigma^2 + \mathbb{E}\Big\{ \big[Y_1 - (TQ_\theta)(S_1, A_1)\big]^2 \Big\}. \tag{3.2}$$

Here the first term in (3.2) is known as the mean-squared Bellman error (MSBE), and the second term is the variance of $Y_1$. Whereas $L(\theta)$ can be viewed as the empirical version of the MSBE, which has bias $\mathbb{E}\{[Y_1 - (TQ_\theta)(S_1, A_1)]^2\}$ that also depends on $\theta$. Thus, without the target network, minimizing $L(\theta)$ can be drastically different from minimizing the MSBE.

To resolve this problem, we use a target network in (3.1), which has expectation

$$\mathbb{E}[L(\theta)] = \|Q_\theta - TQ_{\theta^*}\|_\sigma^2 + \mathbb{E}\Big\{ \big[Y_1 - (TQ_{\theta^*})(S_1, A_1)\big]^2 \Big\},$$

where the variance of $Y_1$ does not depend on $\theta$. Thus, minimizing $L(\theta)$ is close to solving

$$\underset{\theta \in \Theta}{\text{minimize}} \, \|Q_\theta - TQ_{\theta^\star}\|_\sigma^2, \tag{3.3}$$

where $\Theta$ is the parameter space. Note that in DQN we hold $\theta^\star$ still and update $\theta$ for $T_{\text{target}}$ steps. When $T_{\text{target}}$ is sufficiently large and we neglect the fact that the objective in (3.3) is nonconvex, we would update $\theta$ by the minimizer of (3.3) for fixed $\theta^\star$.

Therefore, in the ideal case, DQN aims to solve the minimization problem (3.3) with $\theta^\star$ fixed, and then update $\theta^\star$ by the minimizer $\theta$. Interestingly, this view of DQN offers a statistical interpretation of the target network. In specific, if $\{Q_\theta : \theta \in \Theta\}$ is sufficiently large such that it contains $TQ_{\theta^\star}$, then (3.3) has solution $Q_\theta = TQ_{\theta^\star}$, which can be viewed as one-step of value iteration (Sutton & Barto, 2011) for neural networks. In addition, in the sample setting, $Q_{\theta^\star}$ is used to construct $\{Y_i\}_{i \in [n]}$, which serve as the response in the regression problem defined in (3.1), with $(TQ_{\theta^\star})$ being the regression function.

Furthermore, turning the discussion above into a realizable algorithm, we obtain the neural fitted Q-iteration (FQI) algorithm, which generates a sequence of value functions. Specifically, let $\mathcal{F}$ be a class of function defined on $\mathcal{S} \times \mathcal{A}$. In the $k$-th iteration of FQI, let $\widetilde{Q}_k$ be current estimate of $Q^*$. Similar to (3.1) and (3.3), we define $Y_i = R_i + \gamma \cdot \max_{a \in \mathcal{A}} \widetilde{Q}_k(S_i', a)$, and update $\widetilde{Q}_k$ by

$$\widetilde{Q}_{k+1} = \underset{f \in \mathcal{F}}{\text{argmin}} \, \frac{1}{n} \sum_{i=1}^{n} \big[Y_i - f(S_i, A_i)\big]^2. \tag{3.4}$$

This gives the fitted-Q iteration algorithm, which is stated in Algorithm 1.

When $\mathcal{F}$ is the family of neural networks, Algorithm 1 is known as the neural FQI, which is proposed in Riedmiller (2005). Thus, we can view neural FQI as an modification of DQN, where we replace experience replay by sampling from a fixed distribution $\sigma$, so as to understand its the statistical property. As a byproduct, such a modification naturally justifies the trick of target network in DQN. In addition, note that the optimization problem in (3.4) appears in each iteration of FQI, which is nonconvex when neural networks are used. However, since we focus solely on the statistical aspect, we make the assumption that the global optima of (3.4) can be reached, which is also contained $\mathcal{F}$. Interestingly, a recent line of research on deep learning (Du et al., 2018b;a; Zou et al., 2018; Chizat & Bach, 2018; Allen-Zhu et al., 2018a;b; Jacot et al., 2018; Cao & Gu, 2019; Arora et al., 2019; Ma et al., 2019; Mei et al., 2019; Yehudai & Shamir, 2019) has established global convergence of gradient-based algorithms for empirical risk minimization when the neural networks are overparametrized. We provide more discussions on the computation aspect in §B. Moreover, we make the i.i.d. assumption in Algorithm 1 to simplify the analysis. Antos et al. (2008b) study the performance of fitted value iteration with fixed data used in the regression sub-problems repeatedly, where the data is sampled from a single trajectory based on a fixed policy such that the induced Markov chain satisfies certain conditions on the mixing time. Using similar analysis as in Antos et al. (2008b), our algorithm can also be extended to handled fixed data that is collected beforehand.

---

**Algorithm 1** Fitted Q-Iteration Algorithm

**Input:** MDP $(\mathcal{S}, \mathcal{A}, P, R, \gamma)$, function class $\mathcal{F}$, sampling distribution $\sigma$, number of iterations $K$, number of samples $n$, the initial estimator $\widetilde{Q}_0$.
**for** $k = 0, 1, 2, \ldots, K-1$ **do**
  Sample i.i.d. observations $\{(S_i, A_i), i \in [n]\}$ from $\sigma$, obtain $R_i \sim R(\cdot \,|\, S_i, A_i)$ and $S_i' \sim P(\cdot \,|\, S_i, A_i)$.
  Compute $Y_i = R_i + \gamma \cdot \max_{a \in \mathcal{A}} \widetilde{Q}_k(S_i', a)$.
  Update the action-value function:

$$\widetilde{Q}_{k+1} \leftarrow \operatorname*{argmin}_{f \in \mathcal{F}} \frac{1}{n} \sum_{i=1}^{n} \big[Y_i - f(S_i, A_i)\big]^2.$$

**end for**
Define policy $\pi_K$ as the greedy policy with respect to $\widetilde{Q}_K$.
**Output:** An estimator $\widetilde{Q}_K$ of $Q^*$ and policy $\pi_K$.

---

## 4 THEORETICAL RESULTS

We establish statistical guarantees for DQN with ReLU networks. Specifically, let $Q^{\pi_K}$ be the action-value function corresponding to $\pi_K$, which is returned by Algorithm 1. In the following, we obtain an upper bound for $\|Q^{\pi_K} - Q^*\|_{1,\mu}$, where $\mu \in \mathcal{P}(\mathcal{S} \times \mathcal{A})$ is allowed to be different from $\nu$. In addition, we assume that the state space $\mathcal{S}$ is a compact subset in $\mathbb{R}^r$ and the action space $\mathcal{A}$ is finite. Without loss of generality, we let $\mathcal{S} = [0,1]^r$ hereafter, where $r$ is a fixed integer. To begin with, we first specify the function class $\mathcal{F}$ in Algorithm 1.

**Definition 4.1** (Function Classes). Following Definition 2.1, let $\mathcal{F}(L, \{d_j\}_{j=0}^{L+1}, s)$ be the family of sparse ReLU networks defined on $\mathcal{S}$ with $d_0 = r$ and $d_{L+1} = 1$. Then we define $\mathcal{F}_0$ by

$$\mathcal{F}_0 = \big\{f \colon \mathcal{S} \times \mathcal{A} \to \mathbb{R} \colon f(\cdot, a) \in \mathcal{F}(L, \{d_j\}_{i=0}^{L+1}, s) \text{ for any } a \in \mathcal{A}\big\}. \tag{4.1}$$

In addition, let $\mathcal{G}(\{p_j, t_j, \beta_j, H_j\}_{j \in [q]})$ be set of composition of Hölder smooth functions defined on $\mathcal{S} \subseteq \mathbb{R}^r$. Similar to $\mathcal{F}_0$, we define a function class $\mathcal{G}_0$ as

$$\mathcal{G}_0 = \big\{f \colon \mathcal{S} \times \mathcal{A} \to \mathbb{R} \colon f(\cdot, a) \in \mathcal{G}(\{p_j, t_j, \beta_j, H_j\}_{j \in [q]}) \text{ for any } a \in \mathcal{A}\big\}. \tag{4.2}$$

By this definition, for any function $f \in \mathcal{F}_0$ and any action $a \in \mathcal{A}$, $f(\cdot, a)$ is a ReLU network defined on $\mathcal{S}$, which is standard for Q-networks. Moreover, $\mathcal{G}_0$ contains a broad family of smooth functions on $\mathcal{S} \times \mathcal{A}$. In the following, we make a mild assumption on $\mathcal{F}_0$ and $\mathcal{G}_0$.

**Assumption 4.2.** We assume that for any $f \in \mathcal{F}_0$, we have $Tf \in \mathcal{G}_0$, where $T$ is the Bellman optimality operator defined in (2.6). That is, for any $f \in \mathcal{F}$ and any $a \in \mathcal{A}$, $(Tf)(s, a)$ can be written as compositions of Hölder smooth functions as a function of $s \in \mathcal{S}$.

We remark that this assumption holds when the MDP satisfies some smoothness conditions. For any state-action pair $(s, a) \in \mathcal{S} \times \mathcal{A}$, let $P(\cdot \mid s, a)$ be the density of the next state. By the definition of the Bellman optimality operator in (2.6), we have

$$(Tf)(s, a) = r(s, a) + \gamma \cdot \int_{\mathcal{S}} \left[ \max_{a' \in \mathcal{A}} f(s', a') \right] \cdot P(s' \mid s, a) \mathrm{d}s'. \tag{4.3}$$

For any $s' \in \mathcal{S}$ and $a \in \mathcal{A}$, we define functions $g_1, g_2$ by letting $g_1(s) = r(s, a)$ and $g_2(s) = P(s' \mid s, a)$. Suppose both $g_1$ and $g_2$ are Hölder smooth functions on $\mathcal{S} = [0, 1]^r$ with parameters $\beta$ and $H$. Since $\|f\|_\infty \leq V_{\max}$, by changing the order of integration and differentiation with respect to $s$ in (4.3), we obtain that function $s \to (Tf)(s, a)$ belongs to the Hölder class $\mathcal{C}_r(\mathcal{S}, \beta, H')$ with $H' = H(1 + V_{\max})$. Furthermore, in the more general case, suppose for any fixed $a \in \mathcal{A}$, we can write $P(s' \mid s, a)$ as $h_1[h_2(s, a), h_3(s')]$, where $h_2 \colon \mathcal{S} \to \mathbb{R}^{r_1}$, and $h_3 \colon \mathcal{S} \to \mathbb{R}^{r_2}$ can be viewed as feature mappings, and $h_1 \colon \mathbb{R}^{r_1+r_2} \to \mathbb{R}$ is a bivariate function. We define function $h_4 \colon \mathbb{R}^{r_1} \to \mathbb{R}$ by

$$h_4(u) = \int_{\mathcal{S}} \left[ \max_{a' \in \mathcal{A}} f(s', a') \right] h_1[u, h_3(s')] \mathrm{d}s'.$$

Then by (4.3) we have $(Tf)(s, a) = g_1(s) + h_4 \circ h_2(s, a)$. Then Assumption 4.2 holds if $h_4$ is Hölder smooth and both $g_1$ and $h_2$ can be represented as compositions of Hölder functions. Thus, Assumption 4.2 holds if both the reward function and the transition density of the MDP are sufficiently smooth.

Moreover, even when the transition density is not smooth, we could also expect Assumption 4.2 to hold. Consider the extreme case where the MDP has deterministic transitions, that is, the next state $s'$ is a function of $s$ and $a$, which is denoted by $s' = h(s, a)$. In this case, for any ReLU network $f$, we have $(Tf)(s, a) = r(s, a) + \gamma \cdot \max_{a' \in \mathcal{A}} f[h(s, a), a']$. Since

$$\left| \max_{a' \in \mathcal{A}} f(s_1, a') - \max_{a' \in \mathcal{A}} f(s_2, a') \right| \leq \max_{a' \in \mathcal{A}} \left| f(s_1, a') - f(s_2, a') \right|$$

for any $s_1, s_2 \in \mathcal{S}$, and network $f(\cdot, a)$ is Lipschitz continuous for any fixed $a \in \mathcal{A}$, function $m_1(s) = \max_{a'} f(s, a')$ is Lipschitz on $\mathcal{S}$. Thus, for any fixed $a \in \mathcal{A}$, if both $g_1(s) = r(s, a)$ and $m_2(s) = h(s, a)$ are compositions of Hölder functions, so is $(Tf)(s, a) = g_1(s) + m_1 \circ m_2(s)$. Therefore, even if the MDP has deterministic dynamics, if both the reward function $r(s, a)$ and the transition function $h(s, a)$ are sufficiently nice, Assumption 4.2 still holds true.

In the following, we define the concentration coefficients, which measures the similarity between two probability distributions under the MDP.

**Assumption 4.3** (Concentration Coefficients). Let $\nu_1, \nu_2 \in \mathcal{P}(\mathcal{S} \times \mathcal{A})$ be two probability measures that are absolutely continuous with respect to the Lebesgue measure on $\mathcal{S} \times \mathcal{A}$. Let $\{\pi_t\}_{t \geq 1}$ be a sequence of policies. Suppose the initial state-action pair $(S_0, A_0)$ of the MDP has distribution $\nu_1$, and we take action $A_t$ according to policy $\pi_t$. For any integer $m$, we denote by $P^{\pi_m} P^{\pi_{m-1}} \cdots P^{\pi_1} \nu_1$ the distribution of $(S_m, A_m)$. Then we define the $m$-th concentration coefficient as

$$\kappa(m; \nu_1, \nu_2) = \sup_{\pi_1, \dots, \pi_m} \left[ \mathbb{E}_{\nu_2} \left| \frac{\mathrm{d}(P^{\pi_m} P^{\pi_{m-1}} \cdots P^{\pi_1} \nu_1)}{\mathrm{d}\nu_2} \right|^2 \right]^{1/2}, \tag{4.4}$$

where the supremum is taken over all possible policies.

Furthermore, let $\sigma$ be the sampling distribution in Algorithm 1 and let $\mu$ be a fixed distribution on $\mathcal{S} \times \mathcal{A}$. We assume that there exists a constant $\phi_{\mu,\sigma} < \infty$ such that

$$(1 - \gamma)^2 \cdot \sum_{m \geq 1} \gamma^{m-1} \cdot m \cdot \kappa(m; \mu, \sigma) \leq \phi_{\mu,\sigma}, \tag{4.5}$$

where $(1 - \gamma)^2$ in (4.5) is a normalization term, since $\sum_{m \geq 1} \gamma^{m-1} \cdot m = (1 - \gamma)^{-2}$.

By definition, concentration coefficients in (4.4) quantifies the similarity between $\nu_2$ and the distribution of the future states of the MDP when starting from $\nu_1$. Moreover, (4.5) is a standard assumption in the literature. See, e.g., Munos & Szepesvári (2008); Lazaric et al. (2016); Scherrer et al. (2015); Farahmand et al. (2010; 2016). This assumption holds for large class of systems MDPs and specifically for MDPs whose top-Lyapunov exponent is finite. See Munos & Szepesvári (2008); Antos et al. (2007) for more detailed discussions on this assumption.

Now we are ready to present the main theorem.

**Theorem 4.4.** Under Assumptions 4.2 and 4.3, let $\mathcal{F}_0$ be defined in (4.1) based on the family of sparse ReLU networks $\mathcal{F}(L^*, \{d_j^*\}_{j=0}^{L^*+1}, s^*)$ and let $\mathcal{G}_0$ be given in (4.2). Moreover, for any $j \in [q-1]$, we define $\beta_j^* = \beta_j \cdot \prod_{\ell=j+1} \min(\beta_\ell, 1)$; let $\beta_q^* = 1$. In addition, let $\alpha^* = \max_{j \in [q]} t_j / (2\beta_j^* + t_j)$. For the parameters of $\mathcal{G}_0$, we assume that there exists a constant $\xi > 0$ such that

$$\max_{j \in [q]} t_j \leq \xi, \quad \sum_{j \in [q]} \log t_j \lesssim (\log n)^\xi \quad \text{and} \quad \max_{j \in [q]} p_j \lesssim (\log n)^\xi. \tag{4.6}$$

For the hyperparameters $L^*$, $\{d_j^*\}_{j=0}^{L^*+1}$, and $s^*$ of the ReLU network, we set $d_0^* = 0$ and $d_{L^*+1}^* = 1$. Moreover, we set $L^* \lesssim (\log n)^{\xi'}$,

$$\max_{j \in [q]} \{p_{j+1} \cdot t_j\} \cdot n^{\alpha^*} \lesssim \min_{i \in [L^*]} d_j^* \leq \max_{j \in [L^*]} d_j^* \lesssim n^{\xi'}, \quad \text{and} \quad s^* \asymp n^{\alpha^*} \cdot (\log n)^{\xi'} \tag{4.7}$$

for some constant $\xi' > 0$. For any $K \in \mathbb{N}$, let $Q^{\pi_K}$ be the action-value function corresponding to policy $\pi_K$, which is returned by Algorithm 1 based on function class $\mathcal{F}_0$. Then there exists constants $\xi^*$ and $C$ such that

$$\|Q^* - Q^{\pi_K}\|_{1,\mu} \leq C \cdot \frac{\phi_{\mu,\sigma} \cdot \gamma}{(1-\gamma)^2} \cdot |\mathcal{A}| \cdot (\log n)^{\xi^*} \cdot n^{(\alpha^*-1)/2} + \frac{4\gamma^{K+1}}{(1-\gamma)^2} \cdot R_{\max}. \tag{4.8}$$

This theorem implies that the statistical rate of convergence is the sum of a statistical error and an algorithmic error. The algorithmic error converges to zero in linear rate as the algorithm proceeds, whereas the statistical error reflects the fundamental difficulty of the problem. Thus, when the number of iterations satisfy

$$K \geq C' \cdot \left[ \log |A| + (1 - \alpha^*) \cdot \log n \right] / \log(1/\gamma)$$

iterations, where $C'$ is a sufficiently large constant, the algorithmic error is dominated by the statistical error. In this case, if we view both $\gamma$ and $\phi_{\mu,\sigma}$ as constants and ignore the polylogarithmic term, Algorithm 1 achieves error rate

$$|\mathcal{A}| \cdot n^{(\alpha^*-1)/2} = |\mathcal{A}| \cdot \max_{j \in [q]} n^{-\beta_j^*/(2\beta_j^* + t_j)}, \tag{4.9}$$

which scales linearly with the capacity of the action space, and decays to zero when the $n$ goes to infinity. Furthermore, the rates $\{n^{-\beta_j^*/(2\beta_j^* + t_j)}\}_{j \in [q]}$ in (4.9) recovers the statistical rate of nonparametric regression in $\ell_2$-norm, whereas our statistical rate $n^{(\alpha^*-1)/2}$ in (4.9) is the fastest among these nonparametric rates, which illustrates the benefit of compositional structure of $\mathcal{G}_0$.

Furthermore, as a concrete example, we assume that both the reward function and the Markov transition kernel are Hölder smooth with smoothness parameter $\beta$. As stated below Assumption 4.2, for any $f \in \mathcal{F}_0$, we have $(Tf)(\cdot, a) \in \mathcal{C}_r(\mathcal{S}, \beta, H')$. Then Theorem 4.4 implies that Algorithm 1 achieves error rate $|\mathcal{A}| \cdot n^{-\beta/(2\beta+r)}$ when $K$ is sufficiently large. Since $|\mathcal{A}|$ is finite, this rate achieves the minimax-optimal statistical rate of convergence within the class of Hölder smooth functions defined on $[0, 1]^d$ (Stone, 1982) and thus cannot be further improved.

In addition, as a simple extension of DQN, we propose the Minimax-DQN algorithm for zero-sum Markov game with two players. Specifically, in this setting, there are two players with action spaces $\mathcal{A}$ and $\mathcal{B}$. The action-value function $Q^*(s, a, b): \mathcal{S} \times \mathcal{A} \times \mathcal{B} \to \mathbb{R}$ can be similarly defined, which correspond to the value obtained by a pair of policies that constitute the Nash equilibrium. Minimax-DQN differs from the original DQN mainly in the computation of target, which is obtained by solving

a zero-sum matrix game via linear programming. Using similar proof technique, we establish both the algorithmic and statistical convergence rates of the action-value functions associated with the sequence of policies returned by the Minimax-DQN algorithm. Due to space limit, we defer the algorithm and its theory to §E.1 in the appendix.

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

## A    DEEP Q-NETWORK

We are present the DQN algorithm for MDP in details, which is proposed by Mnih et al. (2015) and adapted here to discounted MDP. As shown in Algorithm 2 below, DQN features two key tricks that lead to its empirical success, namely, experience replay and target network.

---

**Algorithm 2** Deep Q-Network (DQN)

---

**Input:** MDP $(\mathcal{S}, \mathcal{A}, P, R, \gamma)$, replay memory $\mathcal{M}$, number of iterations $T$, minibatch size $n$, exploration probability $\epsilon \in (0, 1)$, a family of deep Q-networks $Q_\theta \colon \mathcal{S} \times \mathcal{A} \to \mathbb{R}$, an integer $T_{\text{target}}$ for updating the target network, and a sequence of stepsizes $\{\alpha_t\}_{t \geq 0}$.
Initialize the replay memory $\mathcal{M}$ to be empty.
Initialize the Q-network with random weights $\theta$.
Initialize the weights of the target network with $\theta^\star = \theta$.
Initialize the initial state $S_0$.
**for** $t = 0, 1, \ldots, T$ **do**
  With probability $\epsilon$, choose $A_t$ uniformly at random from $\mathcal{A}$, and with probability $1 - \epsilon$, choose $A_t$ such that $Q_\theta(S_t, A_t) = \max_{a \in \mathcal{A}} Q_\theta(S_t, a)$.
  Execute $A_t$ and observe reward $R_t$ and the next state $S_{t+1}$.
  Store transition $(S_t, A_t, R_t, S_{t+1})$ in $\mathcal{M}$.
  Experience replay: Sample random minibatch of transitions $\{(s_i, a_i, r_i, s_i')\}_{i \in [n]}$ from $\mathcal{M}$.
  For each $i \in [n]$, compute the target $Y_i = r_i + \gamma \cdot \max_{a \in \mathcal{A}} Q_{\theta^\star}(s_i', a)$.
  Update the Q-network: Perform a gradient descent step

$$\theta \leftarrow \theta - \alpha_t \cdot \frac{1}{n} \sum_{i \in [n]} \big[ Y_i - Q_\theta(s_i, a_i) \big] \cdot \nabla_\theta Q_\theta(s_i, a_i).$$

  Update the target network: Update $\theta^\star \leftarrow \theta$ every $T_{\text{target}}$ steps.
**end for**
Define policy $\overline{\pi}$ as the greedy policy with respect to $Q_\theta$.
**Output:** Action-value function $Q_\theta$ and policy $\overline{\pi}$.

---

## B    COMPUTATIONAL ASPECT OF DQN

Recall that in Algorithm 1 we assume that the global optima of the nonlinear least-squares problem in (3.1) is be obtained in each iteration. We make such assumption as our focus is on the statistical analysis. In terms of optimization, it has been shown recently that, when the neural network is overparametrized, (stochastic) gradient descent converges to the global minima of the empirical function. Moreover, the generalization error of the obtained neural network can also be established. The intuition behind these results is that, when the neural network is overparametrized, it behaves similar to the random feature model (Rahimi & Recht, 2008). See, e.g., Li & Liang (2018); Du et al. (2018b;a); Zou et al. (2018); Chizat & Bach (2018); Allen-Zhu et al. (2018a;b); Jacot et al. (2018); Cao & Gu (2019); Arora et al. (2019); Ma et al. (2019); Mei et al. (2019); Yehudai & Shamir (2019) and the references therein. Also see Fan et al. (2019) for a detailed survey. In the following, we make an initial attempt in providing a unified statistical and computational analysis of DQN.

In the following, we consider the reinforcement learning problem with the state space $\mathcal{S} = [0, 1]^r$ and a finite action space $\mathcal{A}$. To simplify the notation, we represent action $a$ using one-hot embedding and thus identify it as an element in $\{0, 1\}^{|\mathcal{A}|}$. Thus, we can pack the state $s$ and the action $a$ together and obtain a vector $(s, a)$ in $\mathbb{R}^d$, where we denote $r + |\mathcal{A}|$ by $d$.

We represent the Q-network by the family of two-layer neural networks

$$Q(s, a; b, W) = \frac{1}{\sqrt{m}} \sum_{j=1}^{m} b_j \cdot \sigma[W_j^\top(s, a)], \qquad \forall (s, a) \in \mathcal{S} \times \mathcal{A}. \tag{B.1}$$

Here $m$ is the number of neurons, $b_j \in \mathbb{R}$ and $W_j \in \mathbb{R}^d$ for all $j \in [m]$, and $\sigma(u) = \max\{u, 0\}$ is the ReLU activation function. Here $b = (b_1, \ldots, b_m)^\top \in \mathbb{R}^m$ and $W = (W_1, \ldots, W_m) \in \mathbb{R}^{d \times m}$ are the weights of the neural network. Then, in the $k$-th iteration of Algorithm 1, the optimization problem in (3.1) becomes

$$\underset{b, W}{\text{minimize}} \frac{1}{2n} \sum_{i=1}^{n} \big[Y_i - Q(S_i, A_i; b, W)\big]^2, \tag{B.2}$$

where $Y_i = R_i + \gamma \cdot \max_{a \in \mathcal{A}} \widetilde{Q}_k(S_i', a)$ is the target and $\widetilde{Q}_k$ is the Q-network computed in the previous iteration. Notice that this problem is the least-squares regression with two-layer neural networks in the overparametrized setting. We solve the optimization in (B.2) via gradient descent. Specifically, we initialize. the parameters via $b_j \overset{\text{i.i.d.}}{\sim} \text{Unif}(\{-1, 1\})$ and $W_j \overset{\text{i.i.d.}}{\sim} N(0, I_d/d)$, where $I_d$ is the identity matrix in $\mathbb{R}^d$. Moreover, for ease of presentation, during training we keeping $\{b_1, \ldots, b_m\}$ fixed as the random initialization and only optimize over $W$. Moreover, let $W(0) \in \mathbb{R}^{d \times m}$ be the initialization of $W$. We restrict the weight $W$ to a Frobenius ball centered at $W(0)$ with radius $B > 0$, i.e., we define

$$\mathcal{B}_B = \big\{ W \in \mathbb{R}^{d \times m} \colon \|W - W(0)\|_{\text{fro}} \le B \big\}, \tag{B.3}$$

where $B$ is a sufficiently large constant. Thus, the problem in (B.2) is transformed into

$$\underset{W \in \mathcal{B}_B}{\text{minimize}} \, L_n(W) = \frac{1}{2n} \sum_{i=1}^{n} \big[Y_i - Q(S_i, A_i; b, W)\big]^2, \tag{B.4}$$

where $b$ is fixed to the initial value. We solve this optimization problem via projected gradient descent, which generates a sequence of weights $\{W(t)\}_{t \ge 0} \subseteq \mathcal{B}_B$ satisfying

$$W(t+1) = \Pi_{\mathcal{B}_B} \Big[ W(t) - \frac{\eta}{n} \sum_{i=1}^{n} \big[Y_i - Q\big(S_i, A_i; b, W(t)\big)\big] \cdot \nabla_W Q\big(S_i, A_i; b, W(t)\big)\Big], \quad \text{(B.5)}$$

where $\Pi_{\mathcal{B}_B}$ is the projection operator onto $\mathcal{B}_B$ and $\eta > 0$ is the step size.

To understand the convergence of the updates in (B.4), we utilize the fact that overparametrized two-layer neural networks behave similar to the random feature model. Specifically, notice that we have

$$Q(s, a; b, W) = \frac{1}{\sqrt{m}} \sum_{j=1}^{m} b_j \cdot \mathbb{1}\{W_j^\top(s, a) > 0\} \cdot W_j^\top(s, a), \tag{B.6}$$

$$\nabla_{W_j} Q(s, a; b, W) = \frac{1}{\sqrt{m}} b_j \cdot \mathbb{1}\{W_j^\top(s, a) > 0\}, \qquad \forall j \in [m]. \tag{B.7}$$

We define function class

$$\mathcal{F}_{B,m}^{(t)} = \Big\{ Q(s, a) = \frac{1}{\sqrt{m}} \sum_{j=1}^{m} b_j \cdot \mathbb{1}\{W_j(t)^\top(s, a) > 0\} \cdot W_j^\top(s, a) \colon W \in \mathcal{B}_B \Big\}. \tag{B.8}$$

By (B.6) and (B.7), every $Q \in \mathcal{F}_{B,m}^{(t)}$ can be written as

$$Q(s, a) = Q\big(s, a; b, W(t)\big) + \frac{1}{\sqrt{m}} \sum_{j=1}^{m} b_j \cdot \mathbb{1}\{W_j(t)^\top(s, a) > 0\} \cdot [W_j - W_j(t)]^\top(s, a)$$

$$= Q\big(s, a; b, W(t)\big) + \big\langle \nabla_W Q\big(s, a; b, W(t)\big), W - W(t) \big\rangle.$$

Thus, $\mathcal{F}_{B,m}^{(t)}$ contains first-order approximations of $Q(s, a; b, W(t))$. Furthermore, as shown in Du et al. (2018b;a); Arora et al. (2019), when $m$ is sufficiently large, the overall effect of the scaled indicators $\{1/\sqrt{m} \cdot \mathbb{1}\{W_j(t)^\top(s, a) > 0\}$ are well approximated by the indicators produced by the initial weights. That is, when $m$ approaches infinity, for each fixed $t$, $\mathcal{F}_{B,m}^{(t)}$ in (B.8) is close to

$$\mathcal{F}_{B,m} = \left\{ nQ(s, a) = \frac{1}{\sqrt{m}} \sum_{j=1}^{m} b_j \cdot \mathbb{1}\{W_j(0)^\top(s, a) > 0\} \cdot W_j^\top(s, a) \colon W \in \mathcal{B}_B \right\}. \quad \text{(B.9)}$$

Notice that $\{\phi_j(s, a) = 1/\sqrt{m} \cdot b_j \cdot \mathbb{1}\{W_j(0)^\top(s, a) > 0\}$ are i.i.d. random variables. Thus, function class $\mathcal{F}_{B,m}$ is family of functions that can be written as combinations of random features, i.e.,

$$\mathcal{F}_{B,m} = \left\{ Q(s, a) = Q(s, a; b, W(0)) + \sum_{j=1}^{m} \phi_j(s, a)^\top W_j \colon \|W\|_{\text{fro}} \leq B \right\}, \quad \text{(B.10)}$$

where we utilize the definition of $\mathcal{B}_B$. More importantly, when $m$ goes to infinity, the empirical distribution of the random features $\{\phi_j(s, a)\}_{j \in [m]}$ converges to its population distribution. We denote $\phi(\cdot, \cdot; \beta, w)$ as the random feature, where $\beta \in \text{Unif}(\{-1, 1\})$ and $w \sim N(I_d/d)$. We denote $\mu$ as the joint distribution of $\beta$ and $w$. Then, $\mathcal{F}_{B,m}$ in (B.10) converges to a set $\mathcal{F}_B^*$ given by

$$\mathcal{F}_B^* = \left\{ Q(s, a) = Q_0(s, a) + \int \phi(s, a; \beta, w)^\top \alpha(\beta, w) \, d\mu(\beta, w), \colon \int \|\alpha(\beta, w)\|_2^2 \, d\mu(\beta, w) \leq B^2 \right\}, \quad \text{(B.11)}$$

where $\alpha$ is a function over $\{-1, 1\} \times \mathbb{R}^d$ and $Q_0(s, a) = \lim_{m \to \infty} Q(s, a; b, W(0))$.

Furthermore, it can be shown that $\mathcal{F}_B^*$ is a subset of a reproducing kernel Hilbert space (Rahimi & Recht, 2008) $\mathcal{H}$ generated by kernel

$$K((s, a), (s', a')) = \mathbb{E}_{N(I_d/d)}[\mathbb{1}\{w^\top(s, a) > 0\} \cdot \mathbb{1}\{w^\top(s', a') > 0\}\langle(s, a), (s', a')\rangle]. \quad \text{(B.12)}$$

Besides, the inner product induced by the RKHS norm between two functions

$$f_1 = \int \phi(\cdot, \cdot; \beta, w)^\top \alpha_1(\beta, w) \, d\mu(\beta, w) \qquad \text{and} \qquad f_2 = \int \phi(\cdot, \cdot; \beta, w)^\top \alpha_1(\beta, w) \, d\mu(\beta, w)$$

is given by $\langle f_1, f_2 \rangle_{\mathcal{H}} = \int \langle \alpha_1(\beta, w), \alpha_1(\beta, w) \rangle \, d\mu(\beta, w)$. Thus, $\mathcal{F}_B^*$ given in (B.11) can be written a RKHS-norm ball

$$\mathcal{F}_B^* = \{Q = Q_0 + f \colon \|f\|_{\mathcal{H}} \leq B\}$$

with radius $B$. As a result, when both $m$ and $n$ go to infinity, the population problem correspond to (B.2) becomes $\text{minimize}_{Q \in \mathcal{F}_B^*} \|Q - T\widetilde{Q}_k\|_\sigma^2$, whose solution is denoted by $Q_{k+1}^\star$. Therefore, utilizing the recent result in optimization for two-layer neural networks Arora et al. (2019), the Q-networks obtained by the projected gradient descent in (B.5) satisfy

$$\|Q(\cdot, \cdot, W(t)) - Q_{k+1}^\star\|_\sigma^2 \lesssim 1/\sqrt{n} \quad \text{(B.13)}$$

when both $m$ and $t$ are sufficiently large. That is, when the network is overparametrized, projected gradient descent produces a solution $\widetilde{Q}_{t+1}$ with $\mathcal{O}(1/\sqrt{n})$ generalization error. Moreover, using inequality $(a + b)^2 \leq 2a^2 + 2b^2$, we have

$$\|\widetilde{Q}_{t+1} - T\widetilde{Q}_k\|_\sigma^2 \leq 2 \cdot \|\widetilde{Q}_{t+1} - Q_{k+1}^\star\|_{\sigma^2} + 2\|Q_{k+1}^\star - T\widetilde{Q}_k\|_\sigma^2 \lesssim 1/\sqrt{n} + \text{dist}(\mathcal{F}_B^*, \sigma, \mathcal{T}), \quad \text{(B.14)}$$

where $\text{dist}(\mathcal{F}_B^*, \sigma, \mathcal{T})$ is defined as

$$\text{dist}(\mathcal{F}_B^*, \sigma, \mathcal{T}) = \inf_{f \in \mathcal{F}_B^*} \sup_{g \in \mathcal{F}_B^*} \|f - Tg\|_\sigma^2,$$

which measures the approximation error of functions in the RKHS ball $\mathcal{F}_B^*$ with respect the Bellman operator and the sampling distribution $\sigma$. Finally, combining (B.14) with the error propagation result in Theorem D.1, we obtain the error of neural-FQI with overparametrized two-layer neural networks.

## C   RELATED WORK

There is a huge literature in deep reinforcement learning, where algorithms are based on Q-learning or policy gradient (Sutton et al., 2000). We refer the reader to Arulkumaran et al. (2017) for a survey of the recent developments of DRL. In addition, the DQN algorithm is first proposed in Mnih et al. (2015), which applies DQN to Artari 2600 games (Bellemare et al., 2013). The extensions of DQN include double DQN (van Hasselt et al., 2016), dueling DQN (Wang et al., 2016), deep recurrent Q-network (Hausknecht & Stone, 2015), and asynchronous DQN (Mnih et al., 2016). All of these algorithms are corroborated only by numerical experiments, without theoretical guarantees. Moreover, these algorithms not only inherit the tricks of experience replay and the target network proposed in the original DQN, but develop even more tricks to enhance the performance. Furthermore, recent work such as Schaul et al. (2016); Liu & Zou (2017); Zhang & Sutton (2017) study the effect of experience replay and propose various modifications.

In addition, our work is closely related to the literature on batch reinforcement learning (Lange et al., 2012), where the goal is to estimate the value function given transition data. These problems are usually formulated into least-squares regression, for which various algorithms are proposed with finite-sample analysis. However, most existing work focus on the settings where the value function are approximated by linear functions. See Bradtke & Barto (1996); Boyan (2002); Lagoudakis & Parr (2003); Lazaric et al. (2016); Farahmand et al. (2010); Lazaric et al. (2012); Tagorti & Scherrer (2015) and the references therein for results of the least-squares policy iteration (LSPI) and Bellman residue minimization (BRM) algorithms. Beyond linear function approximation, a recent work Farahmand et al. (2016) study the performance of LSPI and BRM when the value function belongs to a reproducing kernel Hilbert space. However, we study the fitted Q-iteration algorithm, which is a batch RL counterpart of DQN. The fitted Q-iteration algorithm is proposed in Ernst et al. (2005), and Riedmiller (2005) propose the neural FQI algorithm. A finite-sample bound for FQI is established in Munos & Szepesvári (2008) for a large class of regressors. However, their results are not applicable to ReLU networks due to the huge capacity of deep neural networks. Furthermore, various extensions of FQI are studied in Antos et al. (2008a); Farahmand et al. (2009); Tosatto et al. (2017); Geist et al. (2019) to handle continuous actions space, ensemble learning, and entropy regularization.

Furthermore, our work is also related to works that apply reinforcement learning to zero-sum Markov games. The Minimax-Q learning is proposed by Littman (1994), which is an online algorithm that is an extension Q-learning. Subsequently, for Markov games, various online algorithms are also proposed with theoretical guarantees. These work consider either the tabular case or linear function approximation. See, e.g., Bowling (2001); Conitzer & Sandholm (2007); Prasad et al. (2015); Wei et al. (2017); Pérolat et al. (2018); Srinivasan et al. (2018); Wei et al. (2017) and the references therein. In addition, batch reinforcement learning is also applied to zero-sum Markov games by Lagoudakis & Parr (2002); Perolat et al. (2015); Pérolat et al. (2016a;b); Zhang et al. (2018), which are closely related to our work. All of these works consider either linear function approximation or a general function class with bounded pseudo-dimension (Anthony & Bartlett, 2009). However, there results cannot directly imply finite-sample bounds for Minimax-DQN due to the huge capacity of deep neural networks.

Finally, our work is also related a line of research on the model capacity of ReLU deep neural networks, which leads to understanding the generalization property of deep learning (Mohri et al., 2012; Kawaguchi et al., 2017). Specifically, Bartlett (1998); Neyshabur et al. (2015b;a); Bartlett et al. (2017a); Golowich et al. (2017); Liang et al. (2017) propose various norms computed from the networks parameters and establish capacity bounds based upon these norms. In addition, Maass (1994); Bartlett et al. (1999); Schmidt-Hieber (2017); Bartlett et al. (2017b); Klusowski & Barron (2016); Suzuki (2018); Bauer et al. (2019) study the Vapnik-Chervonenkis (VC) dimension of neural networks and Dziugaite & Roy (2017); Neyshabur et al. (2017a) establish the PAC-Bayes bounds for neural networks. Among these work, our work is more related to Schmidt-Hieber (2017); Suzuki (2018), which relate the VC dimension of the ReLU networks to a set of hyperparameters used to define the networks. Based on the VC dimension, they study the statistical error of nonparametric

regression using ReLU networks. In sum, theoretical understanding of deep learning is pertinent to the study of DRL algorithms. See Kawaguchi et al. (2017); Neyshabur et al. (2017b); Fan et al. (2019) and the references therein for recent developments on theoretical analysis of the generalization property of deep learning.

# D    PROOF OF THE MAIN THEOREM

In this section, we present a detailed proof of Theorem 4.4.

*Proof.* The proof requires two key ingredients. First in Theorem D.1 we quantify how the error of action-value function approximation propagates through each iteration of Algorithm 1. Then in Theorem D.2 we analyze such one-step approximation error for ReLU networks.

**Theorem D.1** (Error Propagation). Recall that $\{\widetilde{Q}_k\}_{0 \le k \le K}$ are the iterates of Algorithm 1. Let $\pi_K$ be the one-step greedy policy with respect to $\widetilde{Q}_K$, and let $Q^{\pi_K}$ be the action-value function corresponding to $\pi_K$. Under Assumption 4.3, we have

$$\|Q^* - Q^{\pi_K}\|_{1,\mu} \le \frac{2\phi_{\mu,\sigma}\gamma}{(1-\gamma)^2} \cdot \varepsilon_{\max} + \frac{4\gamma^{K+1}}{(1-\gamma)^2} \cdot R_{\max}, \tag{D.1}$$

where we define the maximum one-step approximation error $\varepsilon_{\max} = \max_{k \in [K]} \|T\widetilde{Q}_{k-1} - \widetilde{Q}_k\|_{\sigma}$. Here $\phi_{\mu,\sigma}$ is a constant that only depends on the probability distributions $\mu$ and $\sigma$.

*Proof.* See §F.1 for a detailed proof.                                                                  □

In the sequel, we establish an upper bound for the one-step approximation error $\|T\widetilde{Q}_{k-1} - \widetilde{Q}_k\|_{\sigma}$ for each $k \in [K]$.

**Theorem D.2** (One-step Approximation Error). Let $\mathcal{F} \subseteq \mathcal{B}(\mathcal{S} \times \mathcal{A}, V_{\max})$ be a class of measurable functions on $\mathcal{S} \times \mathcal{A}$ that are bounded by $V_{\max} = R_{\max}/(1-\gamma)$, and let $\sigma$ be a probability distribution on $\mathcal{S} \times \mathcal{A}$. Also, let $\{(S_i, A_i)\}_{i \in [n]}$ be $n$ i.i.d. random variables in $\mathcal{S} \times \mathcal{A}$ following $\sigma$. For each $i \in [n]$, let $R_i$ and $S_i'$ be the reward and the next state corresponding to $(S_i, A_i)$. In addition, for any fixed $Q \in \mathcal{F}$, we define $Y_i = R_i + \gamma \cdot \max_{a \in \mathcal{A}} Q(S_i', a)$. Based on $\{(X_i, A_i, Y_i)\}_{i \in [n]}$, we define $\widehat{Q}$ as the solution to the least-squares problem

$$\underset{f \in \mathcal{F}}{\text{minimize}} \ \frac{1}{n} \sum_{i=1}^{n} \big[f(S_i, A_i) - Y_i\big]^2. \tag{D.2}$$

Meanwhile, for any $\delta > 0$, let $\mathcal{N}(\delta, \mathcal{F}, \|\cdot\|_{\infty})$ be the minimal $\delta$-covering of $\mathcal{F}$ with respect to $\ell_{\infty}$-norm, and we denote by $N_{\delta}$ its cardinality. Then for any $\epsilon \in (0, 1]$ and any $\delta > 0$, we have

$$\|\widehat{Q} - TQ\|_{\sigma}^2 \le (1+\epsilon)^2 \cdot \omega(\mathcal{F}) + C \cdot V_{\max}^2/(n \cdot \epsilon) \cdot \log N_{\delta} + C' \cdot V_{\max} \cdot \delta, \tag{D.3}$$

where $C$ and $C'$ are two positive absolute constants and $\omega(\mathcal{F})$ is defined as

$$\omega(\mathcal{F}) = \sup_{g \in \mathcal{F}} \inf_{f \in \mathcal{F}} \|f - Tg\|_{\sigma}^2. \tag{D.4}$$

*Proof.* See §F.2 for a detailed proof.                                                                  □

To obtain an upper bound for $\|T\widetilde{Q}_{k-1} - \widetilde{Q}_k\|_{\sigma}$ as required in Theorem D.1, we set $Q = \widetilde{Q}_{k-1}$ in Theorem D.2. Then according to Algorithm 1, $\widehat{Q}$ defined in (D.2) becomes $\widetilde{Q}_k$. We set the function class $\mathcal{F}$ in Theorem D.2 to be the family of ReLU Q-networks $\mathcal{F}_0$ defined in (4.1). By setting $\epsilon = 1$ and $\delta = 1/n$ in Theorem D.2, we obtain

$$\|\widetilde{Q}_{k+1} - T\widetilde{Q}_k\|_{\sigma}^2 \le 4 \cdot \omega(\mathcal{F}_0) + C \cdot V_{\max}^2/n \cdot \log N_0, \tag{D.5}$$

where $C$ is a positive absolute constant and

$$N_0 = \big| \mathcal{N}(1/n, \mathcal{F}_0, \| \cdot \|_\infty) \big| \tag{D.6}$$

is the $1/n$-covering number of $\mathcal{F}_0$. In the subsequent proof, we establish upper bounds for $\omega(\mathcal{F}_0)$ defined in (D.4) and $\log N_0$, respectively. Recall that the family of composite Hölder smooth functions $\mathcal{G}_0$ is defined in (4.2). By Assumption 4.2, we have $Tg \in \mathcal{G}_0$ for any $g \in \mathcal{F}_0$. Hence, we have

$$\omega(\mathcal{F}_0) = \sup_{f' \in \mathcal{G}_0} \inf_{f \in \mathcal{F}_0} \| f - f' \|_\sigma^2 \leq \sup_{f' \in \mathcal{G}_0} \inf_{f \in \mathcal{F}_0} \| f - f' \|_\infty^2, \tag{D.7}$$

where the right-hand side is the $\ell_\infty$-error of approximating the functions in $\mathcal{G}_0$ using the family of ReLU networks $\mathcal{F}_0$.

By the definition of $\mathcal{G}_0$ in (4.2), for any $f \in \mathcal{G}_0$ and any $a \in \mathcal{A}$, $f(\cdot, a) \in \mathcal{G}(\{(p_j, t_j, \beta_j, H_j)\}_{j \in [q]})$ is a composition of Hölder smooth functions, that is, $f(\cdot, a) = g_q \circ \cdots \circ g_1$. Recall that, as defined in Definition 2.3, $g_{jk}$ is the $k$-th entry of the vector-valued function $g_j$. Here $g_{jk} \in \mathcal{C}_{t_j}([a_j, b_j]^{t_j}, \beta_j, H_j)$ for each $k \in [p_{j+1}]$ and $j \in [q]$. In the sequel, we construct a ReLU network to approximate $f(\cdot, a)$ and establish an upper bound of the approximation error on the right-hand side of (D.7). We first show that $f(\cdot, a)$ can be reformulated as a composition of Hölder functions defined on a hypercube. We define $h_1 = g_1/(2H_1) + 1/2$,

$$h_j(u) = g_j(2H_{j-1}u - H_{j-1})/(2H_j) + 1/2, \quad \text{for all } j \in \{2, \ldots, q-1\},$$

and $h_q(u) = g_q(2H_{q-1}u - H_{q-1})$. Then we immediately have

$$f(\cdot, a) = g_q \circ \cdots \circ g_1 = h_q \circ \cdots \circ h_1. \tag{D.8}$$

Furthermore, by the definition of Hölder smooth functions in Definition 2.2, for any $k \in [p_2]$, we have that $h_{1k}$ takes value in $[0, 1]$ and $h_{1k} \in \mathcal{C}_{t_1}([0, 1]^{t_1}, \beta_1, 1)$. Similarly, for any $j \in \{2, \ldots, q-1\}$ and $k \in [p_{j+1}]$, $h_{jk}$ also takes value in $[0, 1]$ and

$$h_{jk} \in \mathcal{C}_{t_j}\big([0, 1]^{t_j}, \beta_j, (2H_{j-1})^{\beta_j}\big).$$

Finally, recall that we use the convention that $p_{q+1} = 1$, that is, $h_q$ is a scalar-valued function that satisfies

$$h_q \in \mathcal{C}_{t_q}\big([0, 1]^{t_q}, \beta_q, H_q(2H_{q-1})^{\beta_q}\big).$$

Now we employ the following lemma of Schmidt-Hieber (2017) to construct a ReLU network that approximates each $h_{jk}$, which combined with (D.8) yields a ReLU network that is close to $f(\cdot, a)$. Recall that, as defined in Definition 2.2, we denote by $\mathcal{C}_r(\mathcal{D}, \beta, H)$ the family of Hölder smooth functions with parameters $\beta$ and $H$ on $\mathcal{D} \subseteq \mathbb{R}^r$.

**Lemma D.3** (Approximation of Hölder Smooth Function). For any integers $m \geq 1$ and $N \geq \max\{(\beta+1)^r, (H+1)\}$, let $L = 8 + (m+5) \cdot (1 + \lceil \log_2 r \rceil)$, $d_0 = r$, $d_j = 12rN$ for each $j \in [L]$, and $d_{L+1} = 1$. For any $g \in \mathcal{C}_r([0, 1]^r, \beta, H)$, there exists a ReLU network $f \in \mathcal{F}(L, \{d_j\}_{j=0}^{L+1}, s, V_{\max})$ as defined in Definition 2.1 such that

$$\| f - g \|_\infty \leq (2H + 1) \cdot 3^{r+1} \cdot N \cdot 2^{-m} + H \cdot 2^\beta \cdot N^{-\beta/r},$$

where the parameter $s$ satisfies $s \leq 94 \cdot r^2 \cdot (\beta + 1)^{2r} \cdot N \cdot (m + 6) \cdot (1 + \lceil \log_2 r \rceil)$.

*Proof.* See Theorem 3 of Schmidt-Hieber (2017) for a detailed proof. $\square$

We apply Lemma D.3 to $h_{jk} \colon [0, 1]^{t_j} \to [0, 1]$ for any $j \in [q]$ and $k \in [p_{j+1}]$. We set $m = \eta \cdot \lceil \log_2 n \rceil$ for a sufficiently large constant $\eta > 1$, and set $N$ to be a sufficiently large integer depending on $n$, which will be specified later. In addition, we set

$$L_j = 9 + (m + 5) \cdot (1 + \lceil \log_2 t_j \rceil) \tag{D.9}$$

and define

$$W = \max\big\{ \max_{1 \leq j \leq q-1} (2H_{j-1})^{\beta_j}, H_q(2H_{q-1})^{\beta_q} \big\}, \tag{D.10}$$

which is a constant. Without loss of generality, we assume $W \geq 1$ hereafter. Then by Lemma D.3, there exists a ReLU network $\widehat{h}_{jk}$ such that

$$\|\widehat{h}_{jk} - h_{jk}\|_\infty \leq (2W+1) \cdot 3^{t_j} \cdot N \cdot n^{-\eta} + W \cdot 2^{\beta_j} \cdot N^{-\beta_j/t_j}. \tag{D.11}$$

Furthermore, we have $\widehat{h}_{jk} \in \mathcal{F}(L_j, \{t_j, 12t_jN, \ldots, 12t_jN, 1\}, s_j)$ with

$$s_j \leq 94 \cdot t_j^2 \cdot (\beta_j + 1)^{2t_j} \cdot N \cdot (m+6) \cdot (1 + \lceil \log_2 t_j \rceil). \tag{D.12}$$

Meanwhile, since $h_{j+1} = (h_{(j+1)k})_{k \in [p_{j+2}]}$ takes input from $[0,1]^{t_{j+1}}$, we need to further transform $\widehat{h}_{jk}$ so that it takes value in $[0,1]$. In particular, we define $\sigma(u) = 1 - (1-u)_+ = \min\{\max\{u, 0\}, 1\}$ for any $u \in \mathbb{R}$. Note that $\sigma$ can be represented by a two-layer ReLU network with four nonzero weights. Then we define $\widetilde{h}_{jk} = \sigma \circ \widehat{h}_{jk}$ and $\widetilde{h}_j = (\widetilde{h}_{jk})_{k \in [p_{j+1}]}$. Note that by the definition of $\widetilde{h}_{jk}$, we have $\widetilde{h}_{jk} \in \mathcal{F}(L_j + 2, \{t_j, 12t_jN, \ldots, 12t_jN, 1\}, s_j + 4)$, which yields

$$\widetilde{h}_j \in \mathcal{F}\big(L_j + 2, \{t_j, 12t_jN \cdot p_{j+1}, \ldots, 12t_jN \cdot p_{j+1}, p_{j+1}\}, (s_j + 4) \cdot p_{j+1}\big).$$

Moreover, since both $\widetilde{h}_{jk}$ and $h_{jk}$ take value in $[0,1]$, by (D.11) we have

$$\|\widetilde{h}_{jk} - h_{jk}\|_\infty = \|\sigma \circ \widehat{h}_{jk} - \sigma \circ h_{jk}\|_\infty \leq \|\widehat{h}_{jk} - h_{jk}\|_\infty$$
$$\leq (2W+1) \cdot 3^{t_j} \cdot N \cdot n^{-\eta} + W \cdot 2^{\beta_j} \cdot N^{-\beta_j/t_j}, \tag{D.13}$$

where the constant $W$ is defined in (D.10). Now we define $\widetilde{f} \colon \mathcal{S} \to \mathbb{R}$ by $\widetilde{f} = \widetilde{h}_q \circ \cdots \circ \widetilde{h}_1$, which falls in the function class

$$\mathcal{F}(\widetilde{L}, \{r, \widetilde{d}, \ldots, \widetilde{d}, 1\}, \widetilde{s}), \tag{D.14}$$

where we define $\widetilde{L} = \sum_{j=1}^q (L_j + 2)$, $\widetilde{d} = \max_{j \in [q]} 12t_j \cdot p_{j+1} \cdot N$, and $\widetilde{s} = \sum_{j=1}^q (s_j + 4) \cdot p_{j+1}$. Recall that, as defined in (D.9), we have $L_j = 9 + (m+5) \cdot (1 + \lceil \log_2 t_j \rceil)$. Then when $n$ is sufficiently large, we have

$$\widetilde{L} \leq \sum_{i=1}^q \big[11 + (\log_2 n + 5) \cdot (1 + \lceil \log_2 t_j \rceil)\big] \leq \sum_{i=1}^q 4 \log_2 t_i \cdot \log_2 n \lesssim (\log n)^{1+\xi}, \tag{D.15}$$

where $\xi > 0$ is an absolute constant. Here the last inequality follows from (4.6). Moreover, for $\widetilde{d}$ in (D.14), by (4.6) we have

$$N \cdot \max_{j \in [q]} \{p_{j+1} \cdot t_j\} \lesssim \widetilde{d} \leq 12 \cdot N \cdot \big(\max_{j \in [q]} p_j\big) \cdot \big(\max_{j \in [q]} t_j\big) \lesssim N \cdot (\log n)^\xi. \tag{D.16}$$

In addition, combining (D.12), (4.6), and the fact that $t_j \leq p_j$, we obtain

$$\widetilde{s} \lesssim N \cdot \log n \cdot \big(\max_{j \in [q]} p_j\big) \cdot \Big(\sum_{j=1}^q \log t_j\Big) \lesssim N \cdot (\log n)^{1+2\xi}. \tag{D.17}$$

Now we show that the function class in (D.14) can be embedded in $\mathcal{F}(L^*, \{d_j^*\}_{j=1}^{L^*+1}, s^*)$, where $L^*$, $\{d_j^*\}_{j=1}^{L^*+1}$, and $s^*$ are specified in (4.7). To this end, we set

$$N = \big\lceil \max_{1 \leq j \leq q} C \cdot n^{t_j/(2\beta_j^* + t_j)} \big\rceil, \tag{D.18}$$

where the absolute constant $C > 0$ is sufficiently large. Note that we define $\alpha^* = \max_{j \in [q]} t_j/(2\beta_j^* + t_j)$. Then (D.18) implies that $N \asymp n^{\alpha^*}$. When $\xi'$ in (4.7) satisfies $\xi' \geq 1 + 2\xi$, by (D.15) we have

$$\widetilde{L} \leq L^* \lesssim (\log n)^{\xi'}.$$

In addition, (D.16) and (4.7) implies that we can set $d_j^* \geq \widetilde{d}$ for all $j \in [L^*]$. Finally, by (D.17) and (D.18), we have $\widetilde{s} \lesssim n^{\alpha^*} \cdot (\log n)^{\xi'}$, which implies $\widetilde{s} + (L^* - \widetilde{L}) \cdot r \leq s^*$. For an $\widetilde{L}$-layer ReLU

network in (D.14), we can make it an $L^*$-layer ReLU network by inserting $L^* - \widetilde{L}$ identity layers, since the inputs of each layer are nonnegative. Thus, ReLU networks in (D.14) can be embedded in

$$\mathcal{F}\big[L^*, \{r, r, \ldots, r, \widetilde{d}, \ldots, \widetilde{d}, 1\}, \widetilde{s} + (L^* - \widetilde{L})r\big],$$

which is a subset of $\mathcal{F}(L^*, \{d_j^*\}_{j=1}^{L+1}, s^*)$ by (4.7).

To obtain the approximation error $\|\widetilde{f} - f(\cdot, a)\|_\infty$, we define $G_j = h_j \circ \cdots \circ h_1$ and $\widetilde{G}_j = \widetilde{h}_j \circ \cdots \circ \widetilde{h}_1$ for any $j \in [q]$. By triangle inequality, for any $j > 1$ we have

$$\begin{aligned}
\|G_j - \widetilde{G}_j\|_\infty &\le \|h_j \circ \widetilde{G}_{j-1} - h_j \circ G_{j-1}\|_\infty + \|\widetilde{h}_j \circ \widetilde{G}_{j-1} - h_j \circ \widetilde{G}_{j-1}\|_\infty \\
&\le W \cdot \|G_{j-1} - \widetilde{G}_{j-1}\|_\infty^{\beta_j \wedge 1} + \|h_j - \widetilde{h}_j\|_\infty,
\end{aligned} \tag{D.19}$$

where the second inequality holds since $h_j$ is Hölder smooth. To simplify the notation, we define $\lambda_j = \prod_{\ell=j+1}^q (\beta_\ell \wedge 1)$ for any $j \in [q-1]$, and set $\lambda_q = 1$. By applying recursion to (D.19), we obtain

$$\|f(\cdot, a) - \widetilde{f}\|_\infty = \|G_q - \widetilde{G}_q\|_\infty \le W \sum_{j=1}^q \|\widetilde{h}_j - h_j\|_\infty^{\lambda_j}, \tag{D.20}$$

where the constant $W$ is defined in (D.10). Here in (D.20) we use the fact that $(a+b)^\alpha \le a^\alpha + b^\alpha$ for all $\alpha \in [0,1]$ and $a, b > 0$.

In the sequel, we combine (D.7), (D.13), (D.20), and (D.18) to obtain the final bound on $\omega(\mathcal{F}_0)$. Since we can set the constant $\eta$ in (D.13) to be sufficiently large, the second term on the right-hand side of (D.13) is the leading term asymptotically, that is,

$$\|\widetilde{h}_{jk} - h_{jk}\|_\infty \lesssim N^{-\beta_j/t_j}. \tag{D.21}$$

Also note that $\beta_j^* = \beta_j \cdot \prod_{\ell=j+1}^q (\beta_\ell \wedge 1) = \beta_j \cdot \lambda_j$ for all $j \in [q-1]$. Thus we have $\beta_j^* = \beta_j \cdot \lambda_j$ for all $j \in [q]$. Combining (D.20) and (D.21), we have

$$\|f(\cdot, a) - \widetilde{f}\|_\infty \lesssim \sum_{j=1}^q \big(N^{-\beta_j/t_j}\big)^{\lambda_j} = \sum_{j=1}^q N^{-\beta_j^*/t_j} \lesssim \max_{j \in [q]} N^{-\beta_j^*/t_j}. \tag{D.22}$$

Thus, we combine (D.7), (D.18), and (D.22) to obtain

$$\omega(\mathcal{F}_0) \le \big(\max_{j \in [q]} N^{-\beta_j^*/t_j}\big)^2 \asymp \max_{j \in [q]} n^{-2\beta_j^*/(2\beta_j^* + t_j)} = n^{\alpha^* - 1}. \tag{D.23}$$

As the final step of the proof, it remains to control the covering number of $\mathcal{F}_0$ defined in (4.1). By definition, for any $f \in \mathcal{F}_0$, we have $f(\cdot, a) \in \mathcal{F}(L^*, \{d_j^*\}_{j=1}^{L^*+1}, s^*)$ for any $a \in \mathcal{A}$. For notational simplicity, we denote by $\mathcal{N}_\delta$ the $\delta$-covering of $\mathcal{F}(L^*, \{d_j^*\}_{j=1}^{L^*+1}, s^*)$, that is, we define

$$\mathcal{N}_\delta = \mathcal{N}\big[\delta, \mathcal{F}(L^*, \{d_j^*\}_{j=1}^{L^*+1}, s^*), \|\cdot\|_\infty\big].$$

By the definition of covering, for any $f \in \mathcal{F}_0$ and any $a \in \mathcal{A}$, there exists $g_a \in \mathcal{N}_\delta$ such that $\|f(\cdot, a) - g_a\|_\infty \le \delta$. Then we define a function $g \colon \mathcal{S} \times \mathcal{A} \to \mathbb{R}$ by $g(s, a) = g_a(s)$ for any $(s, a) \in \mathcal{S} \times \mathcal{A}$. By the definition of $g$, it holds that $\|f - g\|_\infty \le \delta$. Therefore, the cardinality of $\mathcal{N}(\delta, \mathcal{F}_0, \|\cdot\|_\infty)$ satisfies

$$\big|\mathcal{N}(\delta, \mathcal{F}_0, \|\cdot\|_\infty)\big| \le |\mathcal{N}_\delta|^{|\mathcal{A}|}. \tag{D.24}$$

Now we utilize the following lemma in Anthony & Bartlett (2009) to obtain an upper bound of the cardinality of $\mathcal{N}_\delta$.

**Lemma D.4** (Covering Number of ReLU Network). Recall that the family of ReLU networks $\mathcal{F}(L, \{d_j\}_{j=0}^{L+1}, s, V_{\max})$ is given in Definition 2.1. Let $D = \prod_{\ell=1}^{L+1}(d_\ell + 1)$. For any $\delta > 0$, we have

$$\log\big|\mathcal{N}\big[\delta, \mathcal{F}(L, \{d_j\}_{j=0}^{L+1}, s, V_{\max}), \|\cdot\|_\infty\big]\big| \le (s+1) \cdot \log\big[2\delta^{-1} \cdot (L+1) \cdot D^2\big].$$

*Proof.* See Theorem 14.5 of Anthony & Bartlett (2009) for a detailed proof. □

Recall that we denote $\mathcal{N}(1/n, \mathcal{F}_0, \|\cdot\|_\infty)$ by $N_0$ in (D.6). By combining (D.24) with Lemma D.4 and setting $\delta = 1/n$, we obtain that

$$\log N_0 \le |\mathcal{A}| \cdot \log |\mathcal{N}_\delta| \le |\mathcal{A}| \cdot (s^* + 1) \cdot \log \big[2n \cdot (L^* + 1) \cdot D^2\big],$$

where $D = \prod_{\ell=1}^{L^*+1}(d_\ell^* + 1)$. By the choice of $L^*$, $s^*$, and $\{d_j^*\}_{j=0}^{L^*+1}$ in (4.7), we conclude that

$$\log N_0 \lesssim |\mathcal{A}| \cdot s^* \cdot L^* \max_{j \in [L^*]} \log(d_j^*) \lesssim n^{\alpha^*} \cdot (\log n)^{1+2\xi'}. \tag{D.25}$$

Finally, combining (D.1), (D.5), (D.23), and (D.25), we conclude the proof of Theorem 4.4. □

## E EXTENSION TO TWO-PLAYER ZERO-SUM MARKOV GAMES

In this section, we propose the Minimax-DQN algorithm, which combines DQN and the Minimax-Q learning for two-player zero-sum Markov games. We first present the background of zero-sum Markov games and introduce the the Minimax-DQN algorithm in §E.1. Borrowing the analysis for DQN in the previous section, we provide theoretical guarantees for the proposed algorithm in §E.2.

### E.1 MINIMAX-DQN ALGORITHM

As one of the simplistic extension of MDP to the multi-agent setting, two-player zero-sum Markov game is denoted by $(\mathcal{S}, \mathcal{A}, \mathcal{B}, P, R, \gamma)$, where $\mathcal{S}$ is state space, $\mathcal{A}$ and $\mathcal{B}$ are the action spaces of the first and second player, respectively. In addition, $P\colon \mathcal{S} \times \mathcal{A} \times \mathcal{B} \to \mathcal{P}(\mathcal{S})$ is the Markov transition kernel, and $R\colon \mathcal{S} \times \mathcal{A} \times \mathcal{B} \to \mathcal{P}(\mathbb{R})$ is the distribution of immediate reward received by the first player. At any time $t$, the two players simultaneously take actions $A_t \in \mathcal{A}$ and $B_t \in \mathcal{B}$ at state $S_t \in \mathcal{S}$, then the first player receives reward $R_t \sim R(S_t, A_t, B_t)$ and the second player obtains $-R_t$. The goal of each agent is to maximize its own cumulative discounted return.

Furthermore, let $\pi\colon \mathcal{S} \to \mathcal{P}(\mathcal{A})$ and $\nu\colon \mathcal{S} \to \mathcal{P}(\mathcal{B})$ be policies of the first and second players, respectively. Then, we similarly define the action-value function $Q^{\pi,\nu}\colon \mathcal{S} \times \mathcal{A} \times \mathcal{B} \to \mathbb{R}$ as

$$Q^{\pi,\nu}(s, a, b) = \mathbb{E}\bigg[\sum_{t=0}^{\infty} \gamma^t \cdot R_t \,\bigg|\, (S_0, A_0, B_0) = (s, a, b), A_t \sim \pi(\cdot\,|S_t), B_t \sim \nu(\cdot\,|S_t)\bigg], \quad \text{(E.1)}$$

and define the state-value function $V^{\pi,\nu}\colon \mathcal{S} \to \mathbb{R}$ as

$$V^{\pi,\nu}(s) = \mathbb{E}\big[Q^{\pi,\nu}(s, A, B) \,\big|\, A \sim \pi(\cdot\,|\,s), B \sim \nu(\cdot\,|\,s)\big]. \tag{E.2}$$

Note that these two value functions are defined by the rewards of the first player. Thus, at any state-action tuple $(s, a, b)$, the two players aim to solve $\max_\pi \min_\nu Q^{\pi,\nu}(s, a, b)$ and $\min_\nu \max_\pi Q^{\pi,\nu}(s, a, b)$, respectively. By the von Neumann's minimax theorem (Von Neumann & Morgenstern, 1947; Patek, 1997), there exists a minimax function of the game, $Q^*\colon \mathcal{S} \times \mathcal{A} \times \mathcal{B} \to \mathbb{R}$, such that

$$Q^*(s, a, b) = \max_\pi \min_\nu Q^{\pi,\nu}(s, a, b) = \min_\nu \max_\pi Q^{\pi,\nu}(s, a, b). \tag{E.3}$$

Moreover, for joint policy $(\pi, \nu)$ of two players, we define the Bellman operators $T^{\pi,\nu}$ and $T$ by

$$(T^{\pi,\nu}Q)(s, a, b) = r(s, a, b) + \gamma \cdot (P^{\pi,\nu}Q)(s, a, b), \tag{E.4}$$

$$(TQ)(s, a, b) = r(s, a, b) + \gamma \cdot (P^*Q)(s, a, b), \tag{E.5}$$

where $r(s, a, b) = \int r R(\mathrm{d}r \,|\, s, a, b)$, and we define operators $P_{\pi,\nu}$ and $P^*$ by

$$(P^{\pi,\nu}Q)(s, a, b) = \mathbb{E}_{s'\sim P(\cdot\,|\,s,a,b), a'\sim\pi(\cdot\,|\,s'), b'\sim\nu(\cdot\,|\,s')}\big[Q(s', a', b')\big],$$

$$(P^*Q)(s, a, b) = \mathbb{E}_{s'\sim P(\cdot\,|\,s,a,b)}\Big\{\max_{\pi'\in\mathcal{P}(\mathcal{A})}\min_{\nu'\in\mathcal{P}(\mathcal{B})}\mathbb{E}_{a'\sim\pi', b'\sim\nu'}\big[Q(s', a', b')\big]\Big\}.$$

Note that $P^*$ is defined by solving a zero-sum matrix game based on $Q(s', \cdot, \cdot) \in \mathbb{R}^{|\mathcal{A}| \times |\mathcal{B}|}$, which could be achieved via linear programming. It can be shown that both $T^{\pi, \nu}$ and $T$ are $\gamma$-contractive, with $Q^{\pi, \nu}$ defined in (E.1) and $Q^*$ defined in (E.3) being the unique fixed points, respectively. Furthermore, similar to (2.3), in zero-sum Markov games, for any action-value function $Q$, the equilibrium joint policy with respect to $Q$ is defined as

$$\big[\pi_Q(\cdot \,|\, s), \nu_Q(\cdot \,|\, s)\big] = \operatorname*{argmax}_{\pi' \in \mathcal{P}(\mathcal{A})} \operatorname*{argmin}_{\nu' \in \mathcal{P}(\mathcal{B})} \mathbb{E}_{a \sim \pi', b \sim \nu'}\big[Q(s, a, b)\big], \qquad \forall s \in \mathcal{S}. \qquad \text{(E.6)}$$

That is, $\pi_Q(\cdot \,|\, s)$ and $\nu_Q(\cdot \,|\, s)$ solves the zero-sum matrix game based on $Q(s, \cdot, \cdot)$ for all $s \in \mathcal{S}$. By this definition, we obtain that the equilibrium joint policy with respect to the minimax function $Q^*$ defined in (E.3) achieves the Nash equilibrium of the Markov game.

Therefore, to learn the Nash equilibrium, it suffices to estimate $Q^*$, which is the unique fixed point of the Bellman operator $T$. Similar to the standard Q-learning for MDP, Littman (1994) proposes the Minimax-Q learning algorithm, which constructs a sequence of action-value functions that converges to $Q^*$. Specifically, in each iteration, based on a transition $(s, a, b, s')$, Minimax-Q learning updates the current estimator of $Q^*$, denoted by $Q$, via

$$Q(s, a, b) \leftarrow (1 - \alpha) \cdot Q(s, a, b) + \alpha \cdot \left\{ r(s, a, b) + \gamma \cdot \max_{\pi' \in \mathcal{P}(\mathcal{A})} \min_{\nu' \in \mathcal{P}(\mathcal{B})} \mathbb{E}_{a' \sim \pi', b' \sim \nu'}\big[Q(s', a', b')\big] \right\},$$

where $\alpha \in (0, 1)$ is the stepsize.

Motivated by this algorithm, we propose the Minimax-DQN algorithm which extend DQN to two-player zero-sum Markov games. Specifically, we parametrize the action-value function using a deep neural network $Q_\theta \colon \mathcal{S} \times \mathcal{A} \times \mathcal{B} \to \mathbb{R}$ and store the transition $(S_t, A_t, B_t, R_t, S_{t+1})$ into the replay memory $\mathcal{M}$ at each time-step. Parameter $\theta$ of the Q-network is updated as follows. Let $Q_{\theta^*}$ be the target network. With $n$ independent samples $\{(s_i, a_i, b_i, r_i, s'_i)\}_{i \in [n]}$ from $\mathcal{M}$, for all $i \in [n]$, we compute the target

$$Y_i = r_i + \gamma \cdot \max_{\pi' \in \mathcal{P}(\mathcal{A})} \min_{\nu' \in \mathcal{P}(\mathcal{B})} \mathbb{E}_{a \sim \pi', b \sim \nu'}\big[Q_{\theta^*}(s'_i, a, b)\big], \qquad \text{(E.7)}$$

which can be attained via linear programming. Then we update $\theta$ in the direction of $\nabla_\theta L(\theta)$, where $L(\theta) = n^{-1} \sum_{i \in [n]} [Y_i - Q_\theta(s_i, a_i, b_i)]^2$. Finally, the target network $Q_{\theta^*}$ is updated every $T_{\text{target}}$ steps by letting $\theta^* = \theta$.

To understand the theoretical aspects of this algorithm, we similarly utilize the framework of batch reinforcement learning for statistical analysis. With the insights gained in §3, we consider a modification of Minimax-DQN based on neural fitted Q-iteration, whose details are stated in Algorithm 4. As in the MDP setting, we replace sampling from the replay memory by sampling i.i.d. state-action tuples from a fixed distribution $\sigma \in \mathcal{P}(\mathcal{S} \times \mathcal{A} \times \mathcal{B})$, and estimate $Q^*$ in (E.3) by solving a sequence of least-squares regression problems.

## E.2 Theoretical Results for Minimax-FQI

Following the theoretical results established in §4, in this subsection, we provide statistical guarantees for the Minimax-FQI algorithm with $\mathcal{F}$ being a family of deep neural networks with ReLU activation. Hereafter, without loss of generality, we assume $\mathcal{S} = [0, 1]^r$ with $r$ being a fixed integer, and the action spaces $\mathcal{A}$ and $\mathcal{B}$ are both finite. To evaluate the performance of the algorithm, we first introduce the best-response policy as follows.

**Definition E.1.** For any policy $\pi \colon \mathcal{S} \to \mathcal{P}(\mathcal{A})$ of player one, the best-response policy against $\pi$, denoted by $\nu_\pi^*$, is defined as the optimal policy of second player when the first player follows $\pi$. In other words, for all $s \in \mathcal{S}$, we have $\nu_\pi^*(\cdot \,|\, s) = \operatorname{argmin}_\nu V^{\pi, \nu}(s)$, where $V^{\pi, \nu}$ is defined in (E.2).

Note that when the first player adopt a fixed policy $\pi$, from the perspective of the second player, the Markov game becomes a MDP. Thus, $\nu_\pi^*$ is the optimal policy of the MDP induced by $\pi$. Moreover, it can be shown that, for any policy $\pi$, $Q^*(s, a, b) \geq Q^{\pi, \nu_\pi^*}(s, a, b)$ holds for every state-action tuple

---

**Algorithm 3** Minimax Deep Q-Network (Minimax-DQN) for the second player

---

**Input:** Zero-Sum Markov game $(\mathcal{S}, \mathcal{A}, \mathcal{B}, P, R, \gamma)$, replay memory $\mathcal{M}$, number of iterations $T$, minibatch size $n$, exploration probability $\epsilon \in (0, 1)$, a family of deep Q-networks $Q_\theta \colon \mathcal{S} \times \mathcal{A} \times \mathcal{B} \to \mathbb{R}$, an integer $T_{\text{target}}$ for updating the target network, and a sequence of stepsizes $\{\alpha_t\}_{t \geq 0}$.

Initialize the replay memory $\mathcal{M}$ to be empty.

Initialize the Q-network with random weights $\theta$.

Initialize the weights of the target network by letting $\theta^\star = \theta$.

Initialize the initial state $S_0$.

**for** $t = 0, 1, \ldots, T$ **do**

    With probability $\epsilon$, choose $B_t$ uniformly at random from $\mathcal{B}$, and with probability $1 - \epsilon$, sample $B_t$ according to the equilibrium policy $\widetilde{\nu}_{Q_\theta}(\cdot \,|\, S_t)$ defined in (**??**).

    Execute $B_t$ and observe the first player's action $A_t$, reward $R_t$ satisfying $-R_t \sim R(S_t, A_t, B_t)$, and the next state $S_{t+1} \sim P(\cdot \,|\, S_t, A_t, B_t)$.

    Store transition $(S_t, A_t, B_t, R_t, S_{t+1})$ in $\mathcal{M}$.

    Experience replay: Sample random minibatch of transitions $\{(s_i, a_i, b_i, r_i, s_i')\}_{i \in [n]}$ from $\mathcal{M}$. For each $i \in [n]$, compute the target

$$Y_i = r_i + \gamma \cdot \max_{\nu' \in \mathcal{P}(\mathcal{B})} \min_{\pi' \in \mathcal{P}(\mathcal{A})} \mathbb{E}_{a \sim \pi', b \sim \nu'} \big[Q_{\theta^*}(s_i', a, b)\big].$$

    Update the Q-network: Perform a gradient descent step

$$\theta \leftarrow \theta - \alpha_t \cdot \frac{1}{n} \sum_{i \in [n]} \big[Y_i - Q_\theta(s_i, a_i, b_i)\big] \cdot \nabla_\theta Q_\theta(s_i, a_i, b_i).$$

    Update the target network: Update $\theta^\star \leftarrow \theta$ every $T_{\text{target}}$ steps.

**end for**

**Output:** Q-network $Q_\theta$ and equilibrium joint policy with respect to $Q_\theta$.

---

**Algorithm 4** Fitted Q-Iteration Algorithm for Zero-Sum Markov Games (Minimax-FQI)

---

**Input:** Two-player zero-sum Markov game $(\mathcal{S}, \mathcal{A}, \mathcal{B}, P, R, \gamma)$, function class $\mathcal{F}$, distribution $\sigma \in \mathcal{P}(\mathcal{S} \times \mathcal{A} \times \mathcal{B})$, number of iterations $K$, number of samples $n$, the initial estimator $\widetilde{Q}_0 \in \mathcal{F}$.

**for** $k = 0, 1, 2, \ldots, K - 1$ **do**

    Sample $n$ i.i.d. observations $\{(S_i, A_i, B_i)\}_{i \in [n]}$ from $\sigma$, obtain $R_i \sim R(\cdot \,|S_i, A_i, B_i)$ and $S_i' \sim P(\cdot \,|S_i, A_i, B_i)$.

    Compute $Y_i = R_i + \gamma \cdot \max_{\pi' \in \mathcal{P}(\mathcal{A})} \min_{\nu' \in \mathcal{P}(\mathcal{B})} \mathbb{E}_{a \sim \pi', b \sim \nu'} \big[\widetilde{Q}_k(s_i', a, b)\big]..$

    Update the action-value function:

$$\widetilde{Q}_{k+1} \leftarrow \underset{f \in \mathcal{F}}{\operatorname{argmin}} \frac{1}{n} \sum_{i=1}^{n} \big[Y_i - f(S_i, A_i, B_i)\big]^2.$$

**end for**

Let $(\pi_K, \nu_K)$ be the equilibrium joint policy with respect to $\widetilde{Q}_K$, which is defined in (E.6).

**Output:** An estimator $\widetilde{Q}_K$ of $Q^*$ and joint policy $(\pi_K, \nu_K)$.

---

$(s, a, b)$. Thus, by considering the adversarial case where the opponent always plays the best-response policy, the difference between $Q^{\pi, \nu_\pi^*}$ and $Q^*$ servers as a characterization of the suboptimality of $\pi$. Hence, to quantify the performance of Algorithm 4, we consider the closeness between $Q^*$ and $Q^{\pi_K, \nu_{\pi_K}^*}$, which will be denoted by $Q_K^*$ hereafter for simplicity. Specifically, in the following we establish an upper bound for $\|Q^* - Q_K^*\|_{1, \mu}$ for some distribution $\mu \in \mathcal{P}(\mathcal{S} \times \mathcal{A} \times \mathcal{B})$.

We first specify the function class $\mathcal{F}$ in Algorithm 4 as follows.

**Assumption E.2** (Function Classes). Following Definition 4.1, let $\mathcal{F}(L, \{d_j\}_{j=0}^{L+1}, s)$ and $\mathcal{G}(\{p_j, t_j, \beta_j, H_j\}_{j\in[q]})$ be the family of sparse ReLU networks and the set of composition of Hölder smooth functions defined on $\mathcal{S}$, respectively. Similar to (4.1), we define $\mathcal{F}_1$ by

$$\mathcal{F}_1 = \big\{ f \colon \mathcal{S} \times \mathcal{A} \to \mathbb{R} \colon f(\cdot, a, b) \in \mathcal{F}(L, \{d_j\}_{i=0}^{L+1}, s) \text{ for any } (a, b) \in \mathcal{A} \times \mathcal{B} \big\}. \qquad \text{(E.8)}$$

For the Bellman operator $T$ defined in (E.5), we assume that for any $f \in \mathcal{F}_1$ and any state-action tuple $(s, a, b)$, we have $(Tf)(\cdot, a, b) \in \mathcal{G}(\{p_j, t_j, \beta_j, H_j\}_{j\in[q]})$.

We remark that this Assumption is in the same flavor as Assumption 4.2. As discussed in §4, this assumption holds if both the reward function and the transition density of the Markov game are sufficiently smooth.

In the following, we define the concentration coefficients for Markov games.

**Assumption E.3** (Concentration Coefficient for Zero-Sum Markov Games). Let $\{\tau_t \colon \mathcal{S} \to \mathcal{P}(\mathcal{A} \times \mathcal{B})\}$ be a sequence of joint policies for the two players in the zero-sum Markov game. Let $\nu_1, \nu_2 \in \mathcal{P}(\mathcal{S} \times \mathcal{A} \times \mathcal{B})$ be two absolutely continuous probability measures. Suppose the initial state-action pair $(S_0, A_0, B_0)$ has distribution $\nu_1$, the future states are sampled according to the Markov transition kernel, and the action $(A_t, B_t)$ is sampled from policy $\tau_t$. For any integer $m$, we denote by $P^{\tau_m} P^{\tau_{m-1}} \cdots P^{\tau_1} \nu_1$ the distribution of $(S_m, A_m, B_m)$. Then, the $m$-th concentration coefficient is defined as

$$\kappa(m; \nu_1, \nu_2) = \sup_{\tau_1, \ldots, \tau_m} \left[ \mathbb{E}_{\nu_2} \left| \frac{\mathrm{d}(P^{\tau_m} P^{\tau_{m-1}} \cdots P^{\tau_1} \nu_1)}{\mathrm{d}\nu_2} \right|^2 \right]^{1/2}, \qquad \text{(E.9)}$$

where the supremum is taken over all possible joint policy sequences $\{\tau_t\}_{t\in[m]}$.

Furthermore, for some $\mu \in \mathcal{P}(\mathcal{S} \times \mathcal{A} \times \mathcal{B})$, we assume that there exists a finite constant $\phi_{\mu,\sigma}$ such that $(1 - \gamma)^2 \cdot \sum_{m\geq 1} \gamma^{m-1} \cdot m \cdot \kappa(m; \mu, \sigma) \leq \phi_{\mu,\sigma}$, where $\sigma$ is the sampling distribution in Algorithm 4 and $\kappa(m; \mu, \sigma)$ is the $m$-th concentration coefficient defined in (E.9).

We remark that the definition of the $m$-th concentration coefficient is the same as in (4.4) if we replace the action space $\mathcal{A}$ of the MDP by $\mathcal{A} \times \mathcal{B}$ of the Markov game. Thus, Assumptions 4.3 and E.3 are of the same nature, which are standard in the literature.

Now we are ready to present the main theorem.

**Theorem E.4.** Under Assumptions E.2 and E.3, consider the Minimax-FQI algorithm with the function class $\mathcal{F}$ being $\mathcal{F}_1$ defined in (E.8) based on the family of sparse ReLU networks $\mathcal{F}(L^*, \{d_j^*\}_{j=0}^{L^*+1}, s^*)$. We make the same assumptions on $\mathcal{F}(L^*, \{d_j^*\}_{j=0}^{L^*+1}, s^*)$ and $\mathcal{G}(\{p_j, t_j, \beta_j, H_j\}_{j\in[q]})$ as in (4.6) and (4.7). Then for any $K \in \mathbb{N}$, let $(\pi_K, \nu_K)$ be the policy returned by the algorithm and let $Q_K^*$ be the action-value function corresponding to $(\pi_K, \nu_{\pi_K}^*)$. Then there exists constants $\xi^*$ and $C$ such that

$$\|Q^* - Q_K^*\|_{1,\mu} \leq C \cdot \frac{\phi_{\mu,\sigma} \cdot \gamma}{(1-\gamma)^2} \cdot |\mathcal{A}| \cdot |\mathcal{B}| \cdot (\log n)^{\xi^*} \cdot n^{(\alpha^*-1)/2} + \frac{4\gamma^{K+1}}{(1-\gamma)^2} \cdot R_{\max}, \qquad \text{(E.10)}$$

where $\alpha^* = \max_{j\in[q]} t_j / (2\beta_j^* + t_j)$ and $\phi_{\mu,\sigma}$ is specified in Assumption E.3.

Similar to Theorem 4.4, the bound in (E.10) shows that closeness between $(\pi_K, \nu_K)$ returned by Algorithm 4 and the Nash equilibrium policy $(\pi_{Q^*}, \nu_{Q^*})$, measured by $\|Q^* - Q_K^*\|_{1,\mu}$, is bounded by the sum of statistical error and an algorithmic error. Specifically, the statistical error balances the bias and variance of estimating the value functions using the family of deep ReLU neural networks, which exhibits the fundamental difficulty of the problem. Whereas the algorithmic error decay to zero geometrically as $K$ increases. Thus, when $K$ is sufficiently large, both $\gamma$ and $\phi_{\mu,\sigma}$ are constants, and the polylogarithmic term is ignored, Algorithm 4 achieves error rate

$$|\mathcal{A}| \cdot |\mathcal{B}| \cdot n^{\alpha^*-1} = |\mathcal{A}| \cdot |\mathcal{B}| \cdot \max_{j\in[q]} n^{-\beta_j^*/(2\beta_j^*+t_j)}, \qquad \text{(E.11)}$$

which scales linearly with the capacity of joint action space. Besides, if $|\mathcal{B}| = 1$, The Minimax-FQI algorithm reduces to Algorithm 1. In this case, (E.11) also recovers the error rate of Algorithm 1. Furthermore, the statistical rate $n^{(\alpha^* - 1)/2}$ achieves the optimal $\ell_2$-norm error of regression for nonparametric regression with a compositional structure, which indicates that the statistical error in (E.10) can not be further improved.

*Proof.* See §G for a detailed proof. □

## F    PROOFS OF AUXILIARY RESULTS

In this section, we present the proofs for Theorems D.1 and D.2, which are used in the §D to establish our main theorem.

### F.1    PROOF OF THEOREM D.1

*Proof.* Before we present the proof, we introduce some notation. For any $k \in \{0, \ldots, K - 1\}$, we denote $T\widetilde{Q}_k$ by $Q_{k+1}$ and define

$$\varrho_k = Q_k - \widetilde{Q}_k. \tag{F.1}$$

Also, we denote by $\pi_k$ the greedy policy with respect to $\widetilde{Q}_k$. In addition, throughout the proof, for two functions $Q_1, Q_2 \colon \mathcal{S} \times \mathcal{A} \to \mathbb{R}$, we use the notation $Q_1 \geq Q_2$ if $Q_1(s, a) \geq Q_2(s, a)$ for any $s \in \mathcal{S}$ and any $a \in \mathcal{A}$, and define $Q_1 \leq Q_2$ similarly. Furthermore, for any policy $\pi$, recall that in (2.4) we define the operator $P^\pi$ by

$$(P^\pi Q)(s, a) = \mathbb{E}\big[Q(S', A') \,\big|\, S' \sim P(\cdot \,|\, s, a), A' \sim \pi(\cdot \,|\, S')\big]. \tag{F.2}$$

In addition, we define the operator $T^\pi$ by

$$(T^\pi Q)(s, a) = r(s, a) + \gamma \cdot (P^\pi Q)(s, a).$$

Finally, we denote $R_{\max}/(1 - \gamma)$ by $V_{\max}$. Now we are ready to present the proof, which consists of three key steps.

**Step (i):** In the first step, we establish a recursion that relates $Q^* - \widetilde{Q}_{k+1}$ with $Q^* - \widetilde{Q}_k$ to measure the sub-optimality of the value function $\widetilde{Q}_k$. In the following, we first establish an upper bound for $Q^* - \widetilde{Q}_{k+1}$ as follows. For each $k \in \{0, \ldots, K - 1\}$, by the definition of $\varrho_{k+1}$ in (F.1), we have

$$\begin{aligned}
Q^* - \widetilde{Q}_{k+1} &= Q^* - (Q_{k+1} - \varrho_{k+1}) = Q^* - Q_{k+1} + \varrho_{k+1} = Q^* - T\widetilde{Q}_k + \varrho_{k+1} \\
&= Q^* - T^{\pi^*}\widetilde{Q}_k + (T^{\pi^*}\widetilde{Q}_k - T\widetilde{Q}_k) + \varrho_{k+1},
\end{aligned} \tag{F.3}$$

where $\pi^*$ is the greedy policy with respect to $Q^*$. Now we leverage the following lemma to show $T^{\pi^*}\widetilde{Q}_k \leq T\widetilde{Q}_k$.

**Lemma F.1.** For any action-value function $Q : \mathcal{S} \times \mathcal{A} \to \mathbb{R}$ and any policy $\pi$, it holds that

$$T^{\pi_Q}Q = TQ \geq T^\pi Q.$$

*Proof.* Note that we have $\max_{a'} Q(s', a') \geq Q(s', a')$ for any $s' \in \mathcal{S}$ and $a' \in \mathcal{A}$. Thus, it holds that

$$\begin{aligned}
(TQ)(s, a) &= r(s, a) + \gamma \cdot \mathbb{E}\big[\max_{a'} Q(S', a') \,\big|\, S' \sim P(\cdot \,|\, s, a)\big] \\
&\geq r(s, a) + \gamma \cdot \mathbb{E}\big[Q(S', A') \,\big|\, S' \sim P(\cdot \,|\, s, a), A' \sim \pi(\cdot \,|\, S')\big] = (T^\pi Q)(s, a).
\end{aligned}$$

Recall that $\pi_Q$ is the greedy policy with respect to $Q$ such that

$$\mathbb{P}\big[A \in \operatorname*{argmax}_a Q(s, a) \,\big|\, A \sim \pi_Q(\cdot \,|\, s)\big] = 1,$$

which implies

$$\mathbb{E}\big[Q(s', A') \,\big|\, A' \sim \pi_Q(\cdot \,|\, s')\big] = \max_{a'} Q(s', a').$$

Consequently, we have

$$\begin{aligned}
(T^{\pi_Q} Q)(s, a) &= r(s, a) + \gamma \cdot \mathbb{E}\big[Q(S', A') \,\big|\, S' \sim P(\cdot \,|\, s, a), A' \sim \pi_Q(\cdot \,|\, S')\big] \\
&= r(s, a) + \gamma \cdot \mathbb{E}\big[\max_{a'} Q(S', a') \,\big|\, S' \sim P(\cdot \,|\, s, a)\big] = (TQ)(s, a),
\end{aligned}$$

which concludes the proof of Lemma F.1. □

By Lemma F.1, we have $T\widetilde{Q}_k \geq T^{\pi^*}\widetilde{Q}_k$. Also note that $Q^*$ is the unique fixed point of $T^{\pi^*}$. Thus, by (F.3) we have

$$Q^* - \widetilde{Q}_{k+1} = (T^{\pi^*}Q^* - T^{\pi^*}\widetilde{Q}_k) + (T^{\pi^*}\widetilde{Q}_k - T\widetilde{Q}_k) + \varrho_{k+1} \leq (T^{\pi^*}Q^* - T^{\pi^*}\widetilde{Q}_k) + \varrho_{k+1}, \tag{F.4}$$

In the following, we establish a lower bound for $Q^* - \widetilde{Q}_{k+1}$ based on $\widetilde{Q}^* - \widetilde{Q}_k$. Note that, by Lemma F.1, we have $T^{\pi_k}\widetilde{Q}_k = T\widetilde{Q}_k$ and $TQ^* \geq T^{\pi_k}Q^*$. Similar to (F.3), since $Q^*$ is the unique fixed point of $T$, it holds that

$$\begin{aligned}
Q^* - \widetilde{Q}_{k+1} &= Q^* - T\widetilde{Q}_k + \varrho_{k+1} = Q^* - T^{\pi_k}\widetilde{Q}_k + \varrho_{k+1} = Q^* - T^{\pi_k}Q^* + (T^{\pi_k}Q^* - T^{\pi_k}\widetilde{Q}_k) + \varrho_{k+1} \\
&= (TQ^* - T^{\pi_k}Q^*) + (T^{\pi_k}Q^* - T^{\pi_k}\widetilde{Q}_k) + \varrho_{k+1} \geq (T^{\pi_k}Q^* - T^{\pi_k}\widetilde{Q}_k) + \varrho_{k+1}. 
\end{aligned} \tag{F.5}$$

Thus, combining (F.4) and (F.5) we obtain that, for any $k \in \{0, \dots, K-1\}$,

$$T^{\pi_k}Q^* - T^{\pi_k}\widetilde{Q}_k + \varrho_{k+1} \leq Q^* - \widetilde{Q}_{k+1} \leq T^{\pi^*}Q^* - T^{\pi^*}\widetilde{Q}_k + \varrho_{k+1}. \tag{F.6}$$

The inequalities in (F.6) show that the error $Q^* - \widetilde{Q}_{k+1}$ can be sandwiched by the summation of a term involving $Q^* - \widetilde{Q}_k$ and the error $\varrho_{k+1}$, which is defined in (F.1) and induced by approximating the action-value function. Using $P^\pi$ defined in (F.2), we can write (F.6) in a more compact form,

$$\gamma \cdot P^{\pi^*}(Q^* - \widetilde{Q}_k) + \varrho_{k+1} \geq Q^* - \widetilde{Q}_{k+1} \geq \gamma \cdot P^{\pi_k}(Q^* - \widetilde{Q}_k) + \varrho_{k+1}. \tag{F.7}$$

Meanwhile, note that $P^\pi$ defined in (F.2) is a linear operator. In fact, $P^\pi$ is the Markov transition operator for the Markov chain on $\mathcal{S} \times \mathcal{A}$ with transition dynamics

$$S_{t+1} \sim P(\cdot \,|\, S_t, A_t), \qquad A_{t+1} \sim \pi(\cdot \,|\, S_{t+1}).$$

By the linearity of the operator $P^\pi$ and the one-step error bound in (F.6), we have the following characterization of the multi-step error.

**Lemma F.2** (Error Propagation). For any $k, \ell \in \{0, 1, \dots, K-1\}$ with $k < \ell$, we have

$$Q^* - \widetilde{Q}_\ell \leq \sum_{i=k}^{\ell-1} \gamma^{\ell-1-i} \cdot (P^{\pi^*})^{\ell-1-i} \varrho_{i+1} + \gamma^{\ell-k} \cdot (P^{\pi^*})^{\ell-k}(Q^* - \widetilde{Q}_k), \tag{F.8}$$

$$Q^* - \widetilde{Q}_\ell \geq \sum_{i=k}^{\ell-1} \gamma^{\ell-1-i} \cdot (P^{\pi_{\ell-1}} P^{\pi_{\ell-2}} \cdots P^{\pi_{i+1}}) \varrho_{i+1} + \gamma^{\ell-k} \cdot (P^{\pi_{\ell-1}} P^{\pi_{\ell-2}} \cdots P^{\pi_k})(Q^* - \widetilde{Q}_k). \tag{F.9}$$

Here $\varrho_{i+1}$ is defined in (F.1) and we use $P^\pi P^{\pi'}$ and $(P^\pi)^k$ to denote the composition of operators.

*Proof.* Note that $P^\pi$ is a linear operator for any policy $\pi$. We obtain (F.8) and (F.9) by iteratively applying the inequalities in (F.7). □

Lemma F.2 gives the upper and lower bounds for the propagation of error through multiple iterations of Algorithm 1, which concludes the first step of our proof.

**Step (ii):** The results in the first step only concern the propagation of error $Q^* - \widetilde{Q}_k$. In contrast, the output of Algorithm 1 is the greedy policy $\pi_k$ with respect to $\widetilde{Q}_k$. In the second step, our goal is to quantify the suboptimality of $Q^{\pi_k}$, which is the action-value function corresponding to $\pi_k$. In the following, we establish an upper bound for $Q^* - Q^{\pi_k}$.

To begin with, we have $Q^* \geq Q^{\pi_k}$ by the definition of $Q^*$ in (2.5). Note that we have $Q^* = T^{\pi^*} Q^*$ and $Q^{\pi_k} = T^{\pi_k} Q^{\pi_k}$. Hence, it holds that

$$
\begin{aligned}
Q^* - Q^{\pi_k} &= T^{\pi^*} Q^* - T^{\pi_k} Q^{\pi_k} = T^{\pi^*} Q^* + (-T^{\pi^*} \widetilde{Q}_k + T^{\pi^*} \widetilde{Q}_k) + (-T^{\pi_k} \widetilde{Q}_k + T^{\pi_k} \widetilde{Q}_k) - T^{\pi_k} Q^{\pi_k} \\
&= (T^{\pi^*} \widetilde{Q}_k - T^{\pi_k} \widetilde{Q}_k) + (T^{\pi^*} Q^* - T^{\pi^*} \widetilde{Q}_k) + (T^{\pi_k} \widetilde{Q}_k - T^{\pi_k} Q^{\pi_k}).
\end{aligned} \tag{F.10}
$$

Now we quantify the three terms on the right-hand side of (F.10) respectively. First, by Lemma F.1, we have

$$
T^{\pi^*} \widetilde{Q}_k - T^{\pi_k} \widetilde{Q}_k = T^{\pi^*} \widetilde{Q}_k - T \widetilde{Q}_k \leq 0. \tag{F.11}
$$

Meanwhile, by the definition of the operator $P^\pi$ in (F.2), we have

$$
T^{\pi^*} Q^* - T^{\pi^*} \widetilde{Q}_k = \gamma \cdot P^{\pi^*} (Q^* - \widetilde{Q}_k), \qquad T^{\pi_k} \widetilde{Q}_k - T^{\pi_k} Q^{\pi_k} = \gamma \cdot P^{\pi_k} (\widetilde{Q}_k - Q^{\pi_k}). \tag{F.12}
$$

Plugging (F.11) and (F.12) into (F.10), we obtain

$$
\begin{aligned}
Q^* - Q^{\pi_k} &\leq \gamma \cdot P^{\pi^*} (Q^* - \widetilde{Q}_k) + \gamma \cdot P^{\pi_k} (\widetilde{Q}_k - Q^{\pi_k}) \\
&= \gamma \cdot (P^{\pi^*} - P^{\pi_k})(Q^* - \widetilde{Q}_k) + \gamma \cdot P^{\pi_k} (Q^* - Q^{\pi_k}),
\end{aligned}
$$

which further implies that

$$
(I - \gamma \cdot P^{\pi_k})(Q^* - Q^{\pi_k}) \leq \gamma \cdot (P^{\pi^*} - P^{\pi_k})(Q^* - \widetilde{Q}_k).
$$

Here $I$ is the identity operator. Since $T^\pi$ is a $\gamma$-contractive operator for any policy $\pi$, $I - \gamma \cdot P^\pi$ is invertible. Thus, we obtain

$$
0 \leq Q^* - Q^{\pi_k} \leq \gamma \cdot (I - \gamma \cdot P^{\pi_k})^{-1} \big[ P^{\pi^*} (Q^* - \widetilde{Q}_k) - P^{\pi_k}(Q^* - \widetilde{Q}_k) \big], \tag{F.13}
$$

which relates $Q^* - Q^{\pi_k}$ with $Q^* - \widetilde{Q}_k$. In the following, we plug Lemma F.2 into (F.13) to obtain the multiple-step error bounds for $Q^{\pi_k}$. First note that, by the definition of $P^\pi$ in (F.2), for any functions $f_1, f_2 : \mathcal{S} \times \mathcal{A} \to \mathbb{R}$ satisfying $f_1 \geq f_2$, we have $P^\pi f_1 \geq P^\pi f_2$. Combining this inequality with the upper bound in (F.8) and the lower bound in (F.9), we have that, for any $k < \ell$,

$$
P^{\pi^*}(Q^* - \widetilde{Q}_\ell) \leq \sum_{i=k}^{\ell-1} \gamma^{\ell-1-i} \cdot (P^{\pi^*})^{\ell-i} \varrho_{i+1} + \gamma^{\ell-k} \cdot (P^{\pi^*})^{\ell-k+1}(Q^* - \widetilde{Q}_k), \tag{F.14}
$$

$$
\begin{aligned}
P^{\pi_\ell}(Q^* - \widetilde{Q}_\ell) \geq \sum_{i=k}^{\ell-1} \gamma^{\ell-1-i} \cdot (P^{\pi_\ell} P^{\pi_{\ell-1}} \cdots P^{\pi_{i+1}}) \varrho_{i+1} \\
+ \gamma^{\ell-k} \cdot (P^{\pi_\ell} P^{\pi_{\ell-1}} \cdots P^{\pi_k})(Q^* - \widetilde{Q}_k).
\end{aligned} \tag{F.15}
$$

Then we plug (F.14) and (F.15) into (F.13) and obtain

$$
\begin{aligned}
0 \leq Q^* - Q^{\pi_\ell} \leq (I - \gamma \cdot P^{\pi_\ell})^{-1} \bigg\{ \sum_{i=k}^{\ell-1} \gamma^{\ell-i} \cdot \big[ (P^{\pi^*})^{\ell-i} - (P^{\pi_\ell} P^{\pi_{\ell-1}} \cdots P^{\pi_{i+1}}) \big] \varrho_{i+1} \\
+ \gamma^{\ell+1-k} \cdot \big[ (P^{\pi^*})^{\ell-k+1} - (P^{\pi_\ell} P^{\pi_{\ell-1}} \cdots P^{\pi_k}) \big](Q^* - \widetilde{Q}_k) \bigg\}
\end{aligned} \tag{F.16}
$$

for any $k < \ell$. To quantify the error of $Q^{\pi_K}$, we set $\ell = K$ and $k = 0$ in (F.16) to obtain

$$
0 \leq Q^* - Q^{\pi_K} \leq (I - \gamma P^{\pi_K})^{-1} \Bigg\{ \sum_{i=0}^{K-1} \gamma^{K-i} \cdot \big[ (P^{\pi^*})^{K-i} - (P^{\pi_K} P^{\pi_{K-1}} \cdots P^{\pi_{i+1}}) \big] \varrho_{i+1}
$$
$$
+ \gamma^{K+1} \cdot \big[ (P^{\pi^*})^{K+1} - (P^{\pi_K} P^{\pi_{K-1}} \cdots P^{\pi_0}) \big] (Q^* - \widetilde{Q}_0) \Bigg\}.
\tag{F.17}
$$

For notational simplicity, we define

$$
\alpha_i = \frac{(1 - \gamma) \gamma^{K-i-1}}{1 - \gamma^{K+1}}, \quad \text{for } 0 \leq i \leq K - 1, \text{ and } \alpha_K = \frac{(1 - \gamma) \gamma^K}{1 - \gamma^{K+1}}.
\tag{F.18}
$$

One can show that $\sum_{i=0}^{K} \alpha_i = 1$. Meanwhile, we define $K + 1$ linear operators $\{O_k\}_{k=0}^{K}$ by

$$
O_i = (1 - \gamma)/2 \cdot (I - \gamma P^{\pi_K})^{-1} \big[ (P^{\pi^*})^{K-i} + (P^{\pi_K} P^{\pi_{K-1}} \cdots P^{\pi_{i+1}}) \big], \quad \text{for } 0 \leq i \leq K - 1,
$$
$$
O_K = (1 - \gamma)/2 \cdot (I - \gamma P^{\pi_K})^{-1} \big[ (P^{\pi^*})^{K+1} + (P^{\pi_K} P^{\pi_{K-1}} \cdots P^{\pi_0}) \big].
$$

Using this notation, for any $(s, a) \in \mathcal{S} \times \mathcal{A}$, by (F.17) we have

$$
\big| Q^*(s, a) - Q^{\pi_K}(s, a) \big|
$$
$$
\leq \frac{2 \gamma (1 - \gamma^{K+1})}{(1 - \gamma)^2} \cdot \Bigg[ \sum_{i=0}^{K-1} \alpha_i \cdot \big( O_i |\varrho_{i+1}| \big)(s, a) + \alpha_K \cdot \big( O_K |Q^* - \widetilde{Q}_0| \big)(s, a) \Bigg],
\tag{F.19}
$$

where both $O_i |\varrho_{i+1}|$ and $O_K |Q^* - \widetilde{Q}_0|$ are functions defined on $\mathcal{S} \times \mathcal{A}$. Here (F.19) gives a uniform upper bound for $Q^* - Q^{\pi_K}$, which concludes the second step.

**Step (iii):** In this step, we conclude the proof by establishing an upper bound for $\|Q^* - Q^{\pi_K}\|_{1,\mu}$ based on (F.19). Here $\mu \in \mathcal{P}(\mathcal{S} \times \mathcal{A})$ is a fixed probability distribution. To simplify the notation, for any measurable function $f : \mathcal{S} \times \mathcal{A} \to \mathbb{R}$, we denote $\mu(f)$ to be the expectation of $f$ under $\mu$, that is, $\mu(f) = \int_{\mathcal{S} \times \mathcal{A}} f(s, a) \mathrm{d}\mu(s, a)$. Using this notation, by (F.19) we bound $\|Q^* - Q^{\pi_\ell}\|_{1,\mu}$ by

$$
\|Q^* - Q^{\pi_K}\|_{1,\mu} = \mu \big( |Q^* - Q^{\pi_K}| \big)
$$
$$
\leq \frac{2 \gamma (1 - \gamma^{K+1})}{(1 - \gamma)^2} \cdot \mu \Bigg[ \sum_{i=0}^{K-1} \alpha_i \cdot \big( O_i |\varrho_{i+1}| \big) + \alpha_K \cdot \big( O_K |Q^* - \widetilde{Q}_0| \big) \Bigg].
\tag{F.20}
$$

By the linearity of expectation, (F.20) implies

$$
\|Q^* - Q^{\pi_K}\|_{1,\mu} \leq \frac{2 \gamma (1 - \gamma^{K+1})}{(1 - \gamma)^2} \cdot \Bigg[ \sum_{i=0}^{K-1} \alpha_i \cdot \mu \big( O_i |\varrho_{i+1}| \big) + \alpha_K \cdot \mu \big( O_K |Q^* - \widetilde{Q}_0| \big) \Bigg].
\tag{F.21}
$$

Furthermore, since both $Q^*$ and $\widetilde{Q}_0$ are bounded by $V_{\max} = R_{\max}/(1 - \gamma)$ in $\ell_\infty$-norm, we have

$$
\mu \big( O_K |Q^* - \widetilde{Q}_0| \big) \leq 2 \cdot R_{\max}/(1 - \gamma).
\tag{F.22}
$$

Moreover, for any $i \in \{0, \ldots, K - 1\}$, by expanding $(1 - \gamma P^{\pi_K})^{-1}$ into a infinite series, we have

$$
\mu \big( O_i |\varrho_{i+1}| \big) = \mu \Bigg\{ \frac{1 - \gamma}{2} \cdot (1 - \gamma P^{\pi_K})^{-1} \big[ (P^{\pi^*})^{K-i} + (P^{\pi_K} P^{\pi_{K-1}} \cdots P^{\pi_{i+1}}) \big] |\varrho_{i+1}| \Bigg\}
$$
$$
= \frac{1 - \gamma}{2} \cdot \mu \Bigg\{ \sum_{j=0}^{\infty} \gamma^j \cdot \big[ (P^{\pi_K})^j (P^{\pi^*})^{K-i} + (P^{\pi_K})^{j+1} (P^{\pi_{K-1}} \cdots P^{\pi_{i+1}}) \big] |\varrho_{i+1}| \Bigg\}.
\tag{F.23}
$$

To upper bound the right-hand side of (F.23), we consider the following quantity

$$
\mu \big[ (P^{\pi_K})^j (P^{\tau_m} P^{\tau_{m-1}} \cdots P^{\tau_1}) f \big] = \int_{\mathcal{S} \times \mathcal{A}} \big[ (P^{\pi_K})^j (P^{\tau_m} P^{\tau_{m-1}} \cdots P^{\tau_1}) f \big](s, a) \mathrm{d}\mu(s, a).
\tag{F.24}
$$

Here $\tau_1, \ldots, \tau_m$ are $m$ policies. Recall that $P^\pi$ is the transition operator of a Markov process defined on $\mathcal{S} \times \mathcal{A}$ for any policy $\pi$. Then the integral on the right-hand side of (F.24) corresponds to the expectation of the function $f(X_t)$, where $\{X_t\}_{t \geq 0}$ is a Markov process defined on $\mathcal{S} \times \mathcal{A}$. Such a Markov process has initial distribution $X_0 \sim \mu$. The first $m$ transition operators are $\{P^{\tau_j}\}_{j \in [m]}$, followed by $j$ identical transition operators $P^{\pi_K}$. Hence, $(P^{\pi_K})^j (P^{\tau_m} P^{\tau_{m-1}} \cdots P^{\tau_1}) \mu$ is the marginal distribution of $X_{j+m}$, which we denote by $\widetilde{\mu}_j$ for notational simplicity. Hence, (F.24) takes the form

$$\mu\big[(P^{\pi_K})^j (P^{\tau_m} P^{\tau_{m-1}} \cdots P^{\tau_1}) f\big] = \mathbb{E}\big[f(X_{j+m})\big] = \widetilde{\mu}_j(f) = \int_{\mathcal{S} \times \mathcal{A}} f(s, a) \mathrm{d}\widetilde{\mu}_j(s, a) \quad \text{(F.25)}$$

for any measurable function $f$ on $\mathcal{S} \times \mathcal{A}$. By Cauchy-Schwarz inequality, we have

$$\widetilde{\mu}_j(f) \leq \left[ \int_{\mathcal{S} \times \mathcal{A}} \left| \frac{\mathrm{d}\widetilde{\mu}_j}{\mathrm{d}\sigma}(s, a) \right|^2 \mathrm{d}\sigma(s, a) \right]^{1/2} \left[ \int_{\mathcal{S} \times \mathcal{A}} |f(s, a)|^2 \mathrm{d}\sigma(s, a) \right]^{1/2}, \quad \text{(F.26)}$$

in which $\mathrm{d}\widetilde{\mu}_j / \mathrm{d}\sigma \colon \mathcal{S} \times \mathcal{A} \to \mathbb{R}$ is the Radon-Nikodym derivative. Recall that the $(m + j)$-th order concentration coefficient $\kappa(m + j; \mu, \sigma)$ is defined in (4.4). Combining (F.25) and (F.26), we obtain

$$\widetilde{\mu}_j(f) \leq \kappa(m + j; \mu, \sigma) \cdot \|f\|_\sigma.$$

Thus, by (F.23) we have

$$\mu\big(O_i | \varrho_{i+1}| \big) = \frac{1 - \gamma}{2} \cdot \sum_{j=0}^\infty \gamma^j \cdot \Big\{ \mu\big[(P^{\pi_K})^j (P^{\pi^*})^{K-i} | \varrho_{i+1}| \big] + \mu\big[(P^{\pi_K})^{j+1} (P^{\pi_{K-1}} \cdots P^{\pi_{i+1}}) | \varrho_{i+1}| \big] \Big\}$$

$$\leq (1 - \gamma) \cdot \sum_{j=0}^\infty \gamma^j \cdot \kappa(K - i + j; \mu, \sigma) \cdot \|\varrho_{i+1}\|_\sigma. \quad \text{(F.27)}$$

Now we combine (F.21), (F.22), and (F.27) to obtain

$$\|Q^* - Q^{\pi_K}\|_{1,\mu} \leq \frac{2\gamma(1 - \gamma^{K+1})}{(1 - \gamma)^2} \cdot \left[ \sum_{i=0}^{K-1} \alpha_i \cdot \mu\big(O_i | \varrho_{i+1}| \big) + \alpha_K \cdot \mu\big(O_K | Q^* - \widetilde{Q}_0| \big) \right]$$

$$\leq \frac{2\gamma(1 - \gamma^{K+1})}{(1 - \gamma)} \cdot \left[ \sum_{i=0}^{K-1} \sum_{j=0}^\infty \alpha_i \cdot \gamma^j \cdot \kappa(K - i + j; \mu, \sigma) \cdot \|\varrho_{i+1}\|_\sigma \right] + \frac{4\gamma(1 - \gamma^{K+1})}{(1 - \gamma)^3} \cdot \alpha_K \cdot R_{\max}.$$

Recall that in Theorem D.1 and (F.1) we define $\varepsilon_{\max} = \max_{i \in [K]} \|\varrho_i\|_\sigma$. We have that $\|Q^* - Q^{\pi_K}\|_{1,\mu}$ is further upper bounded by

$$\|Q^* - Q^{\pi_K}\|_{1,\mu} \quad \text{(F.28)}$$

$$\leq \frac{2\gamma(1 - \gamma^{K+1})}{(1 - \gamma)} \cdot \left[ \sum_{i=0}^{K-1} \sum_{j=0}^\infty \alpha_i \cdot \gamma^j \cdot \kappa(K - i + j; \mu, \sigma) \right] \cdot \varepsilon_{\max} + \frac{4\gamma(1 - \gamma^{K+1})}{(1 - \gamma)^3} \cdot \alpha_K \cdot R_{\max}$$

$$= \frac{2\gamma(1 - \gamma^{K+1})}{(1 - \gamma)} \cdot \left[ \sum_{i=0}^{K-1} \sum_{j=0}^\infty \frac{(1 - \gamma)\gamma^{K-i-1}}{1 - \gamma^{K+1}} \cdot \gamma^j \cdot \kappa(K - i + j; \mu, \sigma) \right] \cdot \varepsilon_{\max} + \frac{4\gamma^{K+1}}{(1 - \gamma)^2} \cdot R_{\max},$$

where the last equality follows from the definition of $\{\alpha_i\}_{0 \leq i \leq K}$ in (F.18). We simplify the summation on the right-hand side of (F.28) and use Assumption 4.3 to obtain

$$\sum_{i=0}^{K-1} \sum_{j=0}^\infty \frac{(1 - \gamma)\gamma^{K-i-1}}{1 - \gamma^{K+1}} \cdot \gamma^j \cdot \kappa(K - i + j; \mu, \sigma)$$

$$= \frac{1 - \gamma}{1 - \gamma^{K+1}} \sum_{j=0}^\infty \sum_{i=0}^{K-1} \gamma^{K-i+j-1} \cdot \kappa(K - i + j; \mu, \sigma)$$

$$\leq \frac{1 - \gamma}{1 - \gamma^{K+1}} \sum_{m=0}^\infty \gamma^{m-1} \cdot m \cdot \kappa(m; \mu, \sigma) \leq \frac{\phi_{\mu,\sigma}}{(1 - \gamma^{K+1})(1 - \gamma)}, \quad \text{(F.29)}$$

where the last inequality follows from (4.5) in Assumption 4.3. Finally, combining (F.28) and (F.29), we obtain

$$\|Q^* - Q^{\pi_K}\|_{1,\mu} \leq \frac{2\gamma \cdot \phi_{\mu,\sigma}}{(1-\gamma)^2} \cdot \varepsilon_{\max} + \frac{4\gamma^{K+1}}{(1-\gamma)^2} \cdot R_{\max},$$

which concludes the third step and hence the proof of Theorem D.1. $\qquad\square$

### F.2 PROOF OF THEOREM D.2

*Proof.* Recall that in Algorithm 1 we define $Y_i = R_i + \gamma \cdot \max_{a \in \mathcal{A}} Q(S_{i+1}, a)$, where $Q$ is any function in $\mathcal{F}$. By definition, we have $\mathbb{E}(Y_i \,|\, S_i = s, A_i = a) = (TQ)(s, a)$ for any $(s, a) \in \mathcal{S} \times \mathcal{A}$. Thus, $TQ$ can be viewed as the underlying truth of the regression problem defined in (D.2), where the covariates and responses are $\{(S_i, A_i)\}_{i \in [n]}$ and $\{Y_i\}_{i \in [n]}$, respectively. Moreover, note that $TQ$ is not necessarily in function class $\mathcal{F}$. We denote by $Q^*$ the best approximation of $TQ$ in $\mathcal{F}$, which is the solution to

$$\underset{f \in \mathcal{F}}{\text{minimize}}\, \|f - TQ\|_\sigma^2 = \mathbb{E}\Big\{ \big[f(S_i, A_i) - Q(S_i, A_i)\big]^2 \Big\}. \qquad (\text{F.30})$$

For notational simplicity, in the sequel we denote $(S_i, A_i)$ by $X_i$ for all $i \in [n]$. For any $f \in \mathcal{F}$, we define $\|f\|_n^2 = 1/n \cdot \sum_{i=1}^n [f(X_i)]^2$. Since both $\widehat{Q}$ and $TQ$ are bounded by $V_{\max} = R_{\max}/(1-\gamma)$, we only need to consider the case where $\log N_\delta \leq n$. Here $N_\delta$ is the cardinality of $\mathcal{N}(\delta, \mathcal{F}, \|\cdot\|_\infty)$. Moreover, let $f_1, \ldots, f_{N_\delta}$ be the centers of the minimal $\delta$-covering of $\mathcal{F}$. Then by the definition of $\delta$-covering, there exists $k^* \in [N_\delta]$ such that $\|\widehat{Q} - f_{k^*}\|_\infty \leq \delta$. It is worth mentioning that $k^*$ is a random variable since $\widehat{Q}$ is obtained from data.

In the following, we prove (D.3) in two steps, which are bridged by $\mathbb{E}[\|\widehat{Q} - TQ\|_n^2]$.

**Step (i):** We relate $\mathbb{E}[\|\widehat{Q} - TQ\|_n^2]$ with its empirical counterpart $\|\widehat{Q} - TQ\|_n^2$. Recall that we define $Y_i = R_i + \gamma \cdot \max_{a \in \mathcal{A}} Q(S_{i+1}, a)$ for each $i \in [n]$. By the definition of $\widehat{Q}$, for any $f \in \mathcal{F}$ we have

$$\sum_{i=1}^n \big[Y_i - \widehat{Q}(X_i)\big]^2 \leq \sum_{i=1}^n \big[Y_i - f(X_i)\big]^2. \qquad (\text{F.31})$$

For each $i \in [n]$, we define $\xi_i = Y_i - (TQ)(X_i)$. Then (F.31) can be written as

$$\|\widehat{Q} - TQ\|_n^2 \leq \|f - TQ\|_n^2 + \frac{2}{n} \sum_{i=1}^n \xi_i \cdot \big[\widehat{Q}(X_i) - f(X_i)\big]. \qquad (\text{F.32})$$

Since both $f$ and $Q$ are deterministic, we have $\mathbb{E}(\|f - TQ\|_n^2) = \|f - TQ\|_\sigma^2$. Moreover, since $\mathbb{E}(\xi_i \,|\, X_i) = 0$ by definition, we have $\mathbb{E}[\xi_i \cdot g(X_i)] = 0$ for any bounded and measurable function $g$. Thus, it holds that

$$\mathbb{E}\Big\{ \sum_{i=1}^n \xi_i \cdot \big[\widehat{Q}(X_i) - f(X_i)\big] \Big\} = \mathbb{E}\Big\{ \sum_{i=1}^n \xi_i \cdot \big[\widehat{Q}(X_i) - (TQ)(X_i)\big] \Big\}. \qquad (\text{F.33})$$

In addition, by triangle inequality and (F.33), we have

$$\Big| \mathbb{E}\Big\{ \sum_{i=1}^n \xi_i \cdot \big[\widehat{Q}(X_i) - (TQ)(X_i)\big] \Big\} \Big|$$

$$\leq \Big| \mathbb{E}\Big\{ \sum_{i=1}^n \xi_i \cdot \big[\widehat{Q}(X_i) - f_{k^*}(X_i)\big] \Big\} \Big| + \Big| \mathbb{E}\Big\{ \sum_{i=1}^n \xi_i \cdot \big[f_{k^*}(X_i) - (TQ)(X_i)\big] \Big\} \Big|, \qquad (\text{F.34})$$

where $f_{k^*}$ satisfies $\|f_{k^*} - \widehat{Q}\|_\infty \leq \delta$. In the following, we upper bound the two terms on the right-hand side of (F.34) respectively. For the first term, by applying Cauchy-Schwarz inequality

twice, we have

$$
\left| \mathbb{E}\left\{ \sum_{i=1}^{n} \xi_i \cdot \left[ \widehat{Q}(X_i) - f_{k^*}(X_i) \right] \right\} \right| \leq \sqrt{n} \cdot \left| \mathbb{E}\left[ \left( \sum_{i=1}^{n} \xi_i^2 \right)^{1/2} \cdot \| \widehat{Q} - f_{k^*} \|_n \right] \right|
$$

$$
\leq \sqrt{n} \cdot \left[ \mathbb{E}\left( \sum_{i=1}^{n} \xi_i^2 \right) \right]^{1/2} \cdot \left[ \mathbb{E}\left( \| \widehat{Q} - f_{k^*} \|_n^2 \right) \right]^{1/2} \leq n\delta \cdot \left[ \mathbb{E}(\xi_i^2) \right]^{1/2}, \qquad \text{(F.35)}
$$

where we use the fact that $\{\xi_i\}_{i \in [n]}$ have the same marginal distributions and $\| \widehat{Q} - f_{k^*} \|_n \leq \delta$. Since both $Y_i$ and $TQ$ are bounded by $V_{\max}$, $\xi_i$ is a bounded random variable by its definition. Thus, there exists a constant $C_\xi > 0$ depending on $\xi$ such that $\mathbb{E}(\xi_i^2) \leq C_\xi^2 \cdot V_{\max}^2$. Then (F.35) implies

$$
\left| \mathbb{E}\left\{ \sum_{i=1}^{n} \xi_i \cdot \left[ \widehat{Q}(X_i) - f_{k^*}(X_i) \right] \right\} \right| \leq C_\xi \cdot V_{\max} \cdot n\delta. \qquad \text{(F.36)}
$$

It remains to upper bound the second term on the right-hand side of (F.34). We first define $N_\delta$ self-normalized random variables

$$
Z_j = \frac{1}{\sqrt{n}} \sum_{i=1}^{n} \xi_i \cdot \left[ f_j(X_i) - (TQ)(X_i) \right] \cdot \| f_j - (TQ) \|_n^{-1} \qquad \text{(F.37)}
$$

for all $j \in [N_\delta]$. Here recall that $\{f_j\}_{j \in [N_\delta]}$ are the centers of the minimal $\delta$-covering of $\mathcal{F}$. Then we have

$$
\left| \mathbb{E}\left\{ \sum_{i=1}^{n} \xi_i \cdot \left[ f_{k^*}(X_i) - (TQ)(X_i) \right] \right\} \right| = \sqrt{n} \cdot \mathbb{E}\left[ \| f_{k^*} - TQ \|_n \cdot |Z_{k^*}| \right]
$$

$$
\leq \sqrt{n} \cdot \mathbb{E}\left\{ \left[ \| \widehat{Q} - TQ \|_n + \| \widehat{Q} - f_{k^*} \|_n \right] \cdot |Z_{k^*}| \right\} \leq \sqrt{n} \cdot \mathbb{E}\left\{ \left[ \| \widehat{Q} - TQ \|_n + \delta \right] \cdot |Z_{k^*}| \right\}, \qquad \text{(F.38)}
$$

where the first inequality follows from triangle inequality and the second inequality follows from the fact that $\| \widehat{Q} - f_{k^*} \|_\infty \leq \delta$. Then applying Cauchy-Schwarz inequality to the last term on the right-hand side of (F.38), we obtain

$$
\mathbb{E}\left\{ \left[ \| \widehat{Q} - TQ \|_n + \delta \right] \cdot |Z_{k^*}| \right\} \leq \left( \mathbb{E}\left\{ \left[ \| \widehat{Q} - TQ \|_n + \delta \right]^2 \right\} \right)^{1/2} \cdot \left[ \mathbb{E}(Z_{k^*}^2) \right]^{1/2}
$$

$$
\leq \left( \left\{ \mathbb{E}\left[ \| \widehat{Q} - TQ \|_n^2 \right] \right\}^{1/2} + \delta \right) \cdot \left[ \mathbb{E}\left( \max_{j \in [N]} Z_j^2 \right) \right]^{1/2}, \qquad \text{(F.39)}
$$

where the last inequality follows from

$$
\mathbb{E}\left[ \| \widehat{Q} - TQ \|_n \right] \leq \left\{ \mathbb{E}\left[ \| \widehat{Q} - TQ \|_n^2 \right] \right\}^{1/2}.
$$

Moreover, since $\xi_i$ is centered conditioning on $\{X_i\}_{i \in [n]}$ and is bounded by $2V_{\max}$, $\xi_i$ is a sub-Gaussian random variable. In specific, there exists an absolute constant $H_\xi > 0$ such that $\| \xi_i \|_{\psi_2} \leq H_\xi \cdot V_{\max}$ for each $i \in [n]$. Here the $\psi_2$-norm of a random variable $W \in \mathbb{R}$ is defined as

$$
\| W \|_{\psi_2} = \sup_{p \geq 1} p^{-1/2} \left[ \mathbb{E}(|W|^p) \right]^{1/p}.
$$

By the definition of $Z_j$ in (F.37), conditioning on $\{X_i\}_{i \in [n]}$, $\xi_i \cdot [f_j(X_i) - (TQ)(X_i)]$ is a centered and sub-Gaussian random variable with

$$
\left\| \xi_i \cdot \left[ f_j(X_i) - (TQ)(X_i) \right] \right\|_{\psi_2} \leq H_\xi \cdot V_{\max} \cdot \left| f_j(X_i) - (TQ)(X_i) \right|.
$$

Moreover, since $Z_j$ is a summation of independent sub-Gaussian random variables, by Lemma 5.9 of **?**, the $\psi_2$-norm of $Z_j$ satisfies

$$
\| Z_j \|_{\psi_2} \leq C \cdot H_\xi \cdot V_{\max} \cdot \| f_j - TQ \|_n^{-1} \cdot \left[ \frac{1}{n} \sum_{i=1}^{n} \left| [f_j(X_i) - (TQ)(X_i)] \right|^2 \right]^{1/2} \leq C \cdot H_\xi \cdot V_{\max},
$$

where $C > 0$ is an absolute constant. Furthermore, by Lemmas 5.14 and 5.15 of **?**, $Z_j^2$ is a sub-exponential random variable, and its the moment-generating function is bounded by

$$\mathbb{E}\big[\exp(t \cdot Z_j^2)\big] \leq \exp(C \cdot t^2 \cdot H_\xi^4 \cdot V_{\max}^4) \tag{F.40}$$

for any $t$ satisfying $C' \cdot |t| \cdot H_\xi^2 \cdot V_{\max}^2 \leq 1$, where $C$ and $C'$ are two positive absolute constants. Moreover, by Jensen's inequality, we bound the moment-generating function of $\max_{j \in [N_\delta]} Z_j^2$ by

$$\mathbb{E}\Big[\exp\big(t \cdot \max_{j \in [N_\delta]} Z_j^2\big)\Big] \leq \sum_{j \in [N_\delta]} \mathbb{E}\big[\exp(t \cdot Z_j^2)\big]. \tag{F.41}$$

Combining (F.40) and (F.41), we have

$$\mathbb{E}\big(\max_{j \in [N]} Z_j^2\big) \leq C^2 \cdot H_\xi^2 \cdot V_{\max}^2 \cdot \log N_\delta, \tag{F.42}$$

where $C > 0$ is an absolute constant. Hence, plugging (F.42) into (F.38) and (F.39), we upper bound the second term of the right-hand side of (F.33) by

$$\left| \mathbb{E}\bigg\{ \sum_{i=1}^n \xi_i \cdot \big[f_{k^*}(X_i) - (TQ)(X_i)\big] \bigg\} \right|$$
$$\leq \bigg( \big\{ \mathbb{E}\big[\|\widehat{Q} - TQ\|_n^2\big] \big\}^{1/2} + \delta \bigg) \cdot C \cdot H_\xi \cdot V_{\max} \cdot \sqrt{n \cdot \log N_\delta}. \tag{F.43}$$

Finally, combining (F.32), (F.36) and (F.43), we obtain the following inequality

$$\mathbb{E}\big[\|\widehat{Q} - TQ\|_n^2\big] \leq \inf_{f \in \mathcal{F}} \mathbb{E}\big[\|f - TQ\|_n^2\big] + C_\xi \cdot V_{\max} \cdot \delta \tag{F.44}$$
$$+ \bigg( \big\{ \mathbb{E}\big[\|\widehat{Q} - (TQ)\|_n^2\big] \big\}^{1/2} + \delta \bigg) \cdot C \cdot H_\xi \cdot V_{\max} \cdot \sqrt{\log N_\delta / n}$$
$$\leq C \cdot V_{\max} \sqrt{\log N_\delta / n} \cdot \big\{ \mathbb{E}\big[\|\widehat{Q} - (TQ)\|_n^2\big] \big\}^{1/2} + \inf_{f \in \mathcal{F}} \mathbb{E}\big[\|f - TQ\|_n^2\big] + C' \cdot V_{\max} \delta,$$

where $C$ and $C'$ are two positive absolute constants. Here in the first inequality we take the infimum over $\mathcal{F}$ because (F.31) holds for any $f \in \mathcal{F}$, and the second inequality holds because $\log N_\delta \leq n$.

Now we invoke a simple fact to obtain the final bound for $\mathbb{E}[\|\widehat{Q} - TQ\|_n^2]$ from (F.44). Let $a, b,$ and $c$ be positive numbers satisfying $a^2 \leq 2ab + c$. For any $\epsilon \in (0, 1]$, since $2ab \leq \epsilon \cdot a^2/(1+\epsilon) + (1+\epsilon) \cdot b^2/\epsilon$, we have

$$a^2 \leq (1 + \epsilon)^2 \cdot b^2/\epsilon + (1 + \epsilon) \cdot c. \tag{F.45}$$

Therefore, applying (F.45) to (F.44) with $a^2 = \mathbb{E}[\|\widehat{Q} - TQ\|_n^2]$, $b = C \cdot V_{\max} \cdot \sqrt{\log N_\delta / n}$, and $c = \inf_{f \in \mathcal{F}} \mathbb{E}[\|f - TQ\|_n^2] + C' \cdot V_{\max} \cdot \delta$, we obtain

$$\mathbb{E}\big[\|\widehat{Q} - TQ\|_n^2\big] \leq (1 + \epsilon) \cdot \inf_{f \in \mathcal{F}} \mathbb{E}\big[\|f - TQ\|_n^2\big] + C \cdot V_{\max}^2 \cdot \log N_\delta / (n\epsilon) + C' \cdot V_{\max} \cdot \delta, \tag{F.46}$$

where $C$ and $C'$ are two positive absolute constants. Now we conclude the first step.

**Step (ii).** In this step, we relate the population risk $\|\widehat{Q} - TQ\|_\sigma^2$ with $\mathbb{E}[\|\widehat{Q} - TQ\|_n^2]$, which is characterized in the first step. To begin with, we generate $n$ i.i.d. random variables $\{\widetilde{X}_i = (\widetilde{S}_i, \widetilde{A}_i)\}_{i \in [n]}$ following $\sigma$, which are independent of $\{(S_i, A_i, R_i, S_i')\}_{i \in [n]}$. Since $\|\widehat{Q} - f_{k^*}\|_\infty \leq \delta$, for any $x \in \mathcal{S} \times \mathcal{A}$, we have

$$\left| \big[\widehat{Q}(x) - (TQ)(x)\big]^2 - \big[f_{k^*}(x) - (TQ)(x)\big]^2 \right|$$
$$= \big|\widehat{Q}(x) - f_{k^*}(x)\big| \cdot \big|\widehat{Q}(x) + f_{k^*}(x) - 2(TQ)(x)\big| \leq 4V_{\max} \cdot \delta, \tag{F.47}$$

where the last inquality follows from the fact that $\|TQ\|_\infty \leq V_{\max}$ and $\|f\|_\infty \leq V_{\max}$ for any $f \in \mathcal{F}$. Then by the definition of $\|\widehat{Q} - TQ\|_\sigma^2$ and (F.47), we have

$$
\|\widehat{Q} - TQ\|_\sigma^2 = \mathbb{E}\left\{ \frac{1}{n} \sum_{i=1}^n \left[ \widehat{Q}(\widetilde{X}_i) - (TQ)(\widetilde{X}_i) \right]^2 \right\}
$$

$$
\leq \mathbb{E}\left\{ \|\widehat{Q} - TQ\|_n^2 + \frac{1}{n} \sum_{i=1}^n \left[ f_{k^*}(\widetilde{X}_i) - (TQ)(\widetilde{X}_i) \right]^2 - \frac{1}{n} \sum_{i=1}^n \left[ f_{k^*}(X_i) - (TQ)(\widetilde{X}_i) \right]^2 \right\} + 8V_{\max} \cdot \delta
$$

$$
= \mathbb{E}\left( \|\widehat{Q} - TQ\|_n^2 \right) + \mathbb{E}\left[ \frac{1}{n} \sum_{i=1}^n h_{k^*}(X_i, \widetilde{X}_i) \right] + 8V_{\max} \cdot \delta, \tag{F.48}
$$

where we apply (F.47) to obtain the first inequality, and in the last equality we define

$$
h_j(x, y) = \left[ f_j(y) - (TQ)(y) \right]^2 - \left[ f_j(x) - (TQ)(x) \right]^2, \tag{F.49}
$$

for any $(x, y) \in \mathcal{S} \times \mathcal{A}$ and any $j \in [N_\delta]$. Note that $h_{k^*}$ is a random function since $k^*$ is random. By the definition of $h_j$ in (F.49), we have $|h_j(x, y)| \leq 4V_{\max}^2$ for any $(x, y) \in \mathcal{S} \times \mathcal{A}$ and $\mathbb{E}[h_j(X_i, \widetilde{X}_i)] = 0$ for any $i \in [n]$. Moreover, the variance of $h_j(X_i, \widetilde{X}_i)$ is upper bounded by

$$
\mathrm{Var}\left[ h_j(X_i, \widetilde{X}_i) \right] = 2 \mathrm{Var}\left\{ \left[ f_j(X_i) - (TQ)(X_i) \right]^2 \right\}
$$

$$
\leq 2\mathbb{E}\left\{ \left[ f_j(X_i) - (TQ)(X_i) \right]^4 \right\} \leq 8\Upsilon^2 \cdot V_{\max}^2,
$$

where we define $\Upsilon$ by letting

$$
\Upsilon^2 = \max\left( 4V_{\max}^2 \cdot \log N_\delta / n, \ \max_{j \in [N_\delta]} \mathbb{E}\left\{ \left[ f_j(X_i) - (TQ)(X_i) \right]^2 \right\} \right). \tag{F.50}
$$

Furthermore, we define

$$
T = \sup_{j \in [N_\delta]} \left| \sum_{i=1}^n h_j(X_i, \widetilde{X}_i) / \Upsilon \right|. \tag{F.51}
$$

Combining (F.48) and (F.51), we obtain

$$
\|\widehat{Q} - TQ\|_\sigma^2 \leq \mathbb{E}\left[ \|\widehat{Q} - TQ\|_n^2 \right] + \Upsilon / n \cdot \mathbb{E}(T) + 8V_{\max} \cdot \delta. \tag{F.52}
$$

In the sequel, we utilize Bernstein's inequality to establish an upper bound for $\mathbb{E}(T)$, which is stated as follows for completeness.

**Lemma F.3** (Bernstein's Inequality). Let $U_1, \ldots U_n$ be $n$ independent random variables satisfying $\mathbb{E}(U_i) = 0$ and $|U_i| \leq M$ for all $i \in [n]$. Then for any $t > 0$, we have

$$
\mathbb{P}\left( \left| \sum_{i=1}^n U_i \right| \geq t \right) \leq 2\exp\left( \frac{-t^2}{2M \cdot t/3 + 2\sigma^2} \right),
$$

where $\sigma^2 = \sum_{i=1}^n \mathrm{Var}(U_i)$ is the variance of $\sum_{i=1}^n U_i$.

We first apply Bernstein's inequality by setting $U_i = h_j(X_i, \widetilde{X}_i) / \Upsilon$ for each $i \in [n]$. Then we take a union bound for all $j \in [N_\delta]$ to obtain

$$
\mathbb{P}(T \geq t) = \mathbb{P}\left[ \sup_{j \in [N_\delta]} \frac{1}{n} \left| \sum_{i=1}^n h_j(X_i, \widetilde{X}_i) / \Upsilon \right| \geq t \right] \leq 2N_\delta \cdot \exp\left\{ \frac{-t^2}{8V_{\max}^2 \cdot [t/(3\Upsilon) + n]} \right\}. \tag{F.53}
$$

Since $T$ is nonnegative, we have $\mathbb{E}(T) = \int_0^\infty \mathbb{P}(T \geq t)\mathrm{d}t$. Thus, for any $u \in (0, 3\Upsilon \cdot n)$, by (F.53) it holds that

$$
\mathbb{E}(T) \leq u + \int_u^\infty \mathbb{P}(T \geq t)\mathrm{d}t \leq u + 2N_\delta \int_u^{3\Upsilon \cdot n} \exp\left( \frac{-t^2}{16V_{\max}^2 \cdot n} \right) \mathrm{d}t + 2N_\delta \int_{3\Upsilon \cdot n}^\infty \exp\left( \frac{-3\Upsilon \cdot t}{16V_{\max}^2} \right) \mathrm{d}t
$$

$$
\leq u + 32N_\delta \cdot V_{\max}^2 \cdot n/u \cdot \exp\left( \frac{-u^2}{16V_{\max}^2 \cdot n} \right) + 32N_\delta \cdot V_{\max}^2 / (3\Upsilon) \cdot \exp\left( \frac{-9\Upsilon^2 \cdot n}{16V_{\max}^2} \right), \tag{F.54}
$$

where in the second inequality we use the fact that $\int_s^\infty \exp(-t^2/2)\mathrm{d}t \le 1/s \cdot \exp(-s^2/2)$ for all $s > 0$. Now we set $u = 4V_{\max}\sqrt{n \cdot \log N_\delta}$ in (F.54) and plug in the definition of $\Upsilon$ in (F.50) to obtain

$$\mathbb{E}(T) \le 4V_{\max}\sqrt{n \cdot \log N_\delta} + 8V_{\max}\sqrt{n/\log N_\delta} + 6V_{\max}\sqrt{n/\log N_\delta} \le 8V_{\max}\sqrt{n \cdot \log N_\delta},$$
(F.55)

where the last inequality holds when $\log N_\delta \ge 4$. Moreover, the definition of $\Upsilon$ in (F.50) implies that $\Upsilon \le \max[2V_{\max}\sqrt{\log N_\delta/n}, \|\widehat{Q} - TQ\|_\sigma + \delta]$. In the following, we only need to consider the case where $\Upsilon \le \|\widehat{Q} - TQ\|_\sigma + \delta$, since we already have (D.3) if $\|\widehat{Q} - TQ\|_\sigma + \delta \le 2V_{\max}\sqrt{\log N_\delta/n}$, which concludes the proof.

Then, when $\Upsilon \le \|\widehat{Q} - TQ\|_\sigma + \delta$ holds, combining (F.52) and (F.55) we obtain

$$\|\widehat{Q} - TQ\|_\sigma^2 \le \mathbb{E}\big[\|\widehat{Q} - TQ\|_n^2\big] + 8V_{\max}\sqrt{\log(N)/n} \cdot \|\widehat{Q} - TQ\|_\sigma + 8V_{\max}\sqrt{\log N_\delta/n} \cdot \delta + 8V_{\max} \cdot \delta$$
$$\le \mathbb{E}\big[\|\widehat{Q} - TQ\|_n^2\big] + 8V_{\max}\sqrt{\log N_\delta/n} \cdot \|\widehat{Q} - TQ\|_\sigma + 16V_{\max} \cdot \delta. \qquad \text{(F.56)}$$

We apply the inequality in (F.45) to (F.56) with $a = \|\widehat{Q} - TQ\|_\sigma$, $b = 8V_{\max}\sqrt{\log N_\delta/n}$, and $c = \mathbb{E}[\|\widehat{Q} - TQ\|_n^2] + 16V_{\max} \cdot \delta$. Hence we finally obtain that

$$\|\widehat{Q} - TQ\|_\sigma^2 \le (1 + \epsilon) \cdot \mathbb{E}\big[\|\widehat{Q} - TQ\|_n^2\big]$$
$$+ (1 + \epsilon)^2 \cdot 64V_{\max} \cdot \log(N)/(n \cdot \epsilon) + (1 + \epsilon) \cdot 18V_{\max} \cdot \delta, \qquad \text{(F.57)}$$

which concludes the second step of the proof.

Finally, combining these two steps together, namely, (F.46) and (F.57), we conclude that

$$\|\widehat{Q} - TQ\|_\sigma^2 \le (1 + \epsilon)^2 \cdot \inf_{f \in \mathcal{F}} \mathbb{E}\big[\|f - TQ\|_n^2\big] + C_1 \cdot V_{\max}^2 \cdot \log N_\delta/(n \cdot \epsilon) + C_2 \cdot V_{\max} \cdot \delta,$$

where $C_1$ and $C_2$ are two absolute constants. Moreover, since $Q \in \mathcal{F}$, we have

$$\inf_{f \in \mathcal{F}} \mathbb{E}\big[\|f - TQ\|_n^2\big] \le \sup_{Q \in \mathcal{F}}\Big\{\inf_{f \in \mathcal{F}} \mathbb{E}\big[\|f - TQ\|_n^2\big]\Big\},$$

which concludes the proof of Theorem D.2. $\qquad \square$

# G  PROOF OF THEOREM E.4

In this section, we present the proof of Theorem E.4. The proof is similar to that of Theorem 4.4, which is presented in §D in details. In the following, we follow the proof in §D and only highlight the differences for brevity.

*Proof.* The proof requires two key ingredients, namely the error propagation and the statistical error incurred by a single step of Minimax-FQI.

**Theorem G.1** (Error Propagation). Recall that $\{\widetilde{Q}_k\}_{0 \le k \le K}$ are the iterates of Algorithm 4 and $(\pi_K, \nu_K)$ is the equilibrium policy with respect to $\widetilde{Q}_K$. Let $Q_K^*$ be the action-value function corresponding to $(\pi_K, \nu_{\pi_K}^*)$, where $\nu_{\pi_K}^*$ is the best-response policy of the second player against $\pi_K$. Then under Assumption E.3, we have

$$\|Q^* - Q_K^*\|_{1,\mu} \le \frac{2\phi_{\mu,\rho} \cdot \gamma}{(1 - \gamma)^2} \cdot \varepsilon_{\max} + \frac{4\gamma^{K+1}}{(1 - \gamma)^2} \cdot R_{\max}, \qquad \text{(G.1)}$$

where we define the maximum one-step approximation error $\varepsilon_{\max} = \max_{k \in [K]} \|T\widetilde{Q}_{k-1} - \widetilde{Q}_k\|_\sigma$, and constant $\phi_{\mu,\nu}$ is specified in Assumption E.3.

*Proof.* We note that the proof of Theorem D.1 cannot be directly applied to prove this theorem. The main reason is that here we also need to consider the role played by the opponent, namely player two. Different from the MDP setting, here $Q_K^*$ is a fixed point of a nonlinear operator due to the fact that player two adopts the optimal policy against $\pi_K$. Thus, we need to conduct a more refined analysis. See §G.1 for a detailed proof. $\qquad \square$

By this theorem, we need to derive an upper bound of $\varepsilon_{\max}$. We achieve such a goal by studying the one-step approximation error $\|T\widetilde{Q}_{k-1} - \widetilde{Q}_k\|_\sigma$ for each $k \in [K]$.

**Theorem G.2** (One-step Approximation Error). Let $\mathcal{F} \subseteq \mathcal{B}(\mathcal{S} \times \mathcal{A} \times \mathcal{B}, V_{\max})$ be a family of measurable functions on $\mathcal{S} \times \mathcal{A} \times \mathcal{B}$ that are bounded by $V_{\max} = R_{\max}/(1 - \gamma)$. Also, let $\{(S_i, A_i, B_i)\}_{i\in[n]}$ be $n$ i.i.d. random variables following distribution $\sigma \in \mathcal{P}(\mathcal{S} \times \mathcal{A} \times \mathcal{B})$. . For each $i \in [n]$, let $R_i$ and $S_i'$ be the reward obtained by the first player and the next state following $(S_i, A_i, B_i)$. In addition, for any fixed $Q \in \mathcal{F}$, we define the response variable as

$$Y_i = R_i + \gamma \cdot \max_{\pi' \in \mathcal{P}(\mathcal{A})} \min_{\nu' \in \mathcal{P}(\mathcal{B})} \mathbb{E}_{a\sim\pi', b\sim\nu'}\big[Q(S_i', a, b)\big]. \tag{G.2}$$

Based on $\{(X_i, A_i, Y_i)\}_{i\in[n]}$, we define $\widehat{Q}$ as the solution to the least-squares problem

$$\min_{f\in\mathcal{F}} \frac{1}{n} \sum_{i=1}^{n} \big[f(S_i, A_i) - Y_i\big]^2. \tag{G.3}$$

Then for any $\epsilon \in (0, 1]$ and any $\delta > 0$, we have

$$\|\widehat{Q} - TQ\|_\sigma^2 \le (1 + \epsilon)^2 \cdot \sup_{g\in\mathcal{F}} \inf_{f\in\mathcal{F}} \|f - Tg\|_\sigma^2 + C \cdot V_{\max}^2/(n \cdot \epsilon) \cdot \log N_\delta + C' \cdot V_{\max} \cdot \delta, \tag{G.4}$$

where $C$ and $C'$ are two positive absolute constants, $T$ is the Bellman operator defined in (E.5), $N_\delta$ is the cardinality of the minimal $\delta$-covering of $\mathcal{F}$ with respect to $\ell_\infty$-norm.

*Proof.* By the definition of $Y_i$ in (G.2), for any $(s, a, b) \in \mathcal{S} \times \mathcal{A} \times \min_{\nu'\in\mathcal{P}(\mathcal{B})}$, we have

$$\mathbb{E}(Y_i \,|\, S_i = s, A_i = a, B_i = b)$$
$$= r(s, a) + \gamma \cdot \mathbb{E}_{s'\sim P(\cdot\,|\,s,a,b)}\Big\{ \max_{\pi'\in\mathcal{P}(\mathcal{A})} \min_{\nu'\in\mathcal{P}(\mathcal{B})} \mathbb{E}_{a'\sim\pi', b'\sim\nu'}\big[Q(s', a, b)\big] \Big\} = (TQ)(s, a, b).$$

Thus, $TQ$ can be viewed as the ground truth of the nonlinear least-squares regression problem in (G.3). Therefore, following the same proof of Theorem D.2, we obtain the desired result. □

Now we let $\mathcal{F}$ be the family of ReLU Q-networks $\mathcal{F}_1$ defined in (E.8) and set $Q = \widetilde{Q}_{k-1}$ in Theorem G.2. In addition, setting $\epsilon = 1$ and $\delta = 1/n$ in (G.4), we obtain

$$\|\widetilde{Q}_{k+1} - T\widetilde{Q}_k\|_\sigma^2 \le 4 \cdot \sup_{g\in\mathcal{F}_1} \inf_{f\in\mathcal{F}_1} \|f - Tg\|_\sigma^2 + C \cdot V_{\max}^2/n \cdot \log N_1$$
$$\le 4 \cdot \sup_{f'\in\mathcal{G}_1} \inf_{f\in\mathcal{F}_1} \|f - f'\|_\infty^2 + C \cdot V_{\max}^2/n \cdot \log N_1, \tag{G.5}$$

where $C$ is a positive absolute constant, $N_1$ is the $1/n$-covering number of $\mathcal{F}_1$, and function class $\mathcal{G}_1$ is defined as

$$\mathcal{G}_1 = \big\{f\colon \mathcal{S} \times \mathcal{A} \to \mathbb{R}\colon f(\cdot, a, b) \in \mathcal{G}(\{p_j, t_j, \beta_j, H_j\}_{j\in[q]}) \text{ for any } (a, b) \in \mathcal{A} \times \mathcal{B}\big\}. \tag{G.6}$$

Here the second inequality follows from Assumption E.2.

Thus, it remains to bound the $\ell_\infty$-error of approximating functions in $\mathcal{G}_1$ using ReLU Q-networks in $\mathcal{F}_1$ and the $1/n$-covering number of $\mathcal{F}_1$. In the sequel, obtain upper bounds for these two terms.

By the definition of $\mathcal{G}_1$ in (G.6), for any $f \in \mathcal{G}_1$ and any $(a, b) \in \mathcal{A} \times \mathcal{B}$, we have $f(\cdot, a, b) \in \mathcal{G}(\{(p_j, t_j, \beta_j, H_j)\}_{j\in[q]})$. Following the same construction as in §F.2, we can find a function $\widetilde{f}$ in $\mathcal{F}(L^*, \{d_j^*\}_{j=1}^{L^*+1}, s^*)$ such that

$$\|f(\cdot, a, b) - \widetilde{f}\|_\infty \lesssim \max_{j\in[q]} n^{-2\beta_j^*/(2\beta_j^*+t_j)} = n^{\alpha^*-1},$$

which implies that

$$\sup_{f'\in\mathcal{G}_1} \inf_{f\in\mathcal{F}_1} \|f - f'\|_\infty^2 \lesssim n^{\alpha^*-1}. \tag{G.7}$$

Moreover, for any $f \in \mathcal{F}_1$ and any $(a, b) \in \mathcal{A} \times \mathcal{B}$, we have $f(\cdot, a, b) \in \mathcal{F}(L^*, \{d_j^*\}_{j=1}^{L^*+1}, s^*)$. Let $\mathcal{N}_\delta$ be the $\delta$-covering of $\mathcal{F}(L^*, \{d_j^*\}_{j=1}^{L^*+1}, s^*)$ in the $\ell_\infty$-norm. Then for any $f \in \mathcal{F}_1$ and any $(a, b) \in \mathcal{A} \times \mathcal{B}$, there exists $g_{ab} \in \mathcal{N}_\delta$ such that $\|f(\cdot, a, b) - g_{a,b}\|_\infty \leq \delta$. Thus, the cardinality of the $\mathcal{N}(\delta, \mathcal{F}_1, \|\cdot\|_\infty)$ satisfies

$$\left|\mathcal{N}(\delta, \mathcal{F}_1, \|\cdot\|)\right| \leq |\mathcal{N}_\delta|^{|\mathcal{A}| \cdot |\mathcal{B}|}. \tag{G.8}$$

Combining (G.8) with Lemma D.4 and setting $\delta = 1/n$, we obtain that

$$\log N_1 \leq |\mathcal{A}| \cdot |\mathcal{B}| \cdot \log |\mathcal{N}_\delta| \leq |\mathcal{A}| \cdot |\mathcal{B}| \cdot (s^* + 1) \cdot \log\left[2n \cdot (L^* + 1) \cdot D^2\right]$$
$$\leq |\mathcal{A}| \cdot |\mathcal{B}| \cdot s^* \cdot L^* \max_{j \in [L^*]} \log(d_j^*) \lesssim |\mathcal{A}| \cdot |\mathcal{B}| \cdot n^{\alpha^*} \cdot (\log n)^{1+2\xi'}, \tag{G.9}$$

where $D = \prod_{\ell=1}^{L^*+1}(d_\ell^* + 1)$ and the second inequality follows from (4.7).

Finally, combining (G.1), (G.5), (G.7), and (G.9), we conclude the proof of Theorem E.4. $\qquad\square$

## G.1 PROOF OF THEOREM G.1

*Proof.* The proof is similar to the that of Theorem G.1. Before presenting the proof, we first introduce the following notation for simplicity. For any $k \in \{0, \ldots, K-1\}$, we denote $T\widetilde{Q}_k$ by $Q_{k+1}$ and define $\varrho_k = Q_k - \widetilde{Q}_k$. In addition, throughout the proof, for two action-value functions $Q_1$ and $Q_2$, we write $Q_1 \leq Q_2$ if $Q_1(s, a, b) \geq Q_2(s, a, b)$ for any $(s, a, b) \in \mathcal{S} \times \mathcal{A} \times \mathcal{B}$, and define $Q_1 \geq Q_2$ similarly. Furthermore, we denote by $(\pi_k, \nu_k)$ and $(\pi^*, \nu^*)$ the equilibrium policies with respect to $\widetilde{Q}_k$ by $Q^*$, respectively. Besides, in addition to the Bellman operators $T^{\pi,\nu}$ and $T$ defined in (E.4) and (E.5), for any policy $\pi$ of the first player, we define

$$T^\pi Q(s, a, b) = r(s, a, b) + \gamma \cdot \mathbb{E}_{s' \sim P(\cdot \mid s, a, b)}\left\{\min_{\nu' \in \mathcal{P}(\mathcal{B})} \mathbb{E}_{a' \sim \pi, b' \sim \nu'}\left[Q(s', a', b')\right]\right\}, \tag{G.10}$$

corresponds to the case where the first player follows policy $\pi$ and player 2 adopts the best policy in response to $\pi$. By this definition, it holds that $Q^* = T^{\pi^*}Q^*$. Unlike the MDP setting, here $T^\pi$ is a nonlinear operator due to the minimization in (G.10). Furthermore, for any fixed action-value function $Q$, we define the best-response policy against $\pi$ with respect to $Q$, denote by $\nu(\pi, Q)$, as

$$\nu(\pi, Q)(\cdot \mid s) = \operatorname*{argmin}_{\nu' \in \mathcal{P}(\mathcal{B})} \mathbb{E}_{a \sim \pi, b \sim \nu'}\left[Q(s, a, b)\right]. \tag{G.11}$$

Using this notation, we can write (G.10) equivalently as

$$T^\pi Q(s, a, b) = r(s, a, b) + \gamma \cdot \left(P^{\pi, \nu(\pi, Q)}\right)(s, a, b).$$

Notice that $P^{\pi, \nu(\pi, Q)}$ is a linear operator and that $\nu_Q = \nu(\pi_Q, Q)$ by definition.

Now we are ready to present the proof, which can be decomposed into three key steps.

**Step (i):** In the first step, we establish recursive upper and lower bounds for $\{Q^* - \widetilde{Q}_k\}_{0 \leq k \leq K}$. For each $k \in \{0, \ldots, K-1\}$, similar to the decomposition in (F.3), we have

$$Q^* - \widetilde{Q}_{k+1} = Q^* - T^{\pi^*}\widetilde{Q}_k + (T^{\pi^*}\widetilde{Q}_k - T\widetilde{Q}_k) + \varrho_{k+1}, \tag{G.12}$$

where $\pi^*$ is part of the equilibrium policy with respect to $Q^*$ and $T^{\pi^*}$ is defined in (G.10).

Similar to Lemma F.1, we utilize the following lemma to show $T^{\pi^*}\widetilde{Q}_k \geq T\widetilde{Q}_k$.

**Lemma G.3.** For any action-value function $Q : \mathcal{S} \times \mathcal{A} \times \mathcal{B} \to \mathbb{R}$, let $(\pi_Q, \nu_Q)$ be the equilibrium policy with respect to $Q$. Then for and any policy $\pi$ of the first player, it holds that

$$T^{\pi_Q}Q = TQ \geq T^\pi Q.$$

Furthermore, for any policy $\pi : \mathcal{S} \to \mathcal{P}(\mathcal{A})$ of player one and any action-value function $Q$, we have

$$T^{\pi, \nu(\pi, Q)}Q = T^\pi Q \leq T^{\pi, \nu}Q \tag{G.13}$$

for any policy $\nu : \mathcal{S} \to \mathcal{P}(\mathcal{B})$, where $\nu(\pi, Q)$ is the best-response policy defined in (G.11).

*Proof.* Note that for any $s' \in \mathcal{S}$, by the definition of equilibrium policy, we have

$$\max_{\pi' \in \mathcal{P}(\mathcal{A})} \min_{\nu' \in \mathcal{P}(\mathcal{B})} \mathbb{E}_{a' \sim \pi', b' \sim \nu'} \big[ Q(s', a', b') \big] = \min_{\nu' \in \mathcal{P}(\mathcal{B})} \mathbb{E}_{a' \sim \pi_Q, b' \sim \nu'} \big[ Q(s', a', b') \big].$$

Thus, for any state-action tuple $(s, a, b)$, taking conditional expectations of $s$ with respect to $P(\cdot \,|\, s, a, b)$ on both ends of this equation, we have

$$(T^{\pi_Q} Q)(s, a, b) = r(s, a, b) + \gamma \cdot \mathbb{E}_{s' \sim P(\cdot \,|\, s, a, b)} \Big\{ \min_{\nu' \in \mathcal{P}(\mathcal{B})} \mathbb{E}_{a' \sim \pi_Q, b' \sim \nu'} \big[ Q(s', a', b') \big] \Big\}$$

$$= r(s, a, b) + \gamma \cdot \mathbb{E}_{s' \sim P(\cdot \,|\, s, a, b)} \Big\{ \max_{\pi' \in \mathcal{P}(\mathcal{A})} \min_{\nu' \in \mathcal{P}(\mathcal{B})} \mathbb{E}_{a' \sim \pi', b' \sim \nu'} \big[ Q(s', a', b') \big] \Big\} = (TQ)(s, a, b),$$

which proves $T^{\pi_Q} Q = TQ$. Moreover, for any policy $\pi$ of the first player, it holds that

$$\max_{\pi' \in \mathcal{P}(\mathcal{A})} \min_{\nu' \in \mathcal{P}(\mathcal{B})} \mathbb{E}_{a' \sim \pi', b' \sim \nu'} \big[ Q(s', a', b') \big] \geq \min_{\nu' \in \mathcal{P}(\mathcal{B})} \mathbb{E}_{a' \sim \pi, b' \sim \nu'} \big[ Q(s', a', b') \big].$$

Taking expectations with respect to $s' \sim P(\cdot \,|\, s, a, b)$ on both ends, we establish $TQ \geq T^{\pi}Q$.

It remains to show the second part of Lemma G.3. By the definition of $\nu(\pi, Q)$, we have

$$\mathbb{E}_{a' \sim \pi, b' \sim \nu(\pi, Q)} \big[ Q(s', a', b') \big] = \min_{\nu' \in \mathcal{P}(\mathcal{B})} \mathbb{E}_{a' \sim \pi, b' \sim \nu'} \big[ Q(s', a', b') \big],$$

which, combined with the definition of $T^{\pi}$ in (G.10), implies that $T^{\pi, \nu(\pi, Q)} Q = T^{\pi} Q$. Finally, for any policy $\nu$ of player two, we have

$$\min_{\nu' \in \mathcal{P}(\mathcal{B})} \mathbb{E}_{a' \sim \pi, b' \sim \nu'} \big[ Q(s', a', b') \big] \geq \mathbb{E}_{a' \sim \pi, b' \sim \nu} \big[ Q(s', a', b') \big],$$

which yields $T^{\pi} Q \leq T^{\pi, \nu} Q$. Thus, we conclude the proof of this lemma. $\qquad \square$

Hereafter, for notational simplicity, for each $k$, let $(\pi_k, \nu_k)$ be the equilibrium joint policy with respect to $\widetilde{Q}_k$, and we denote $\nu(\pi^*, \widetilde{Q}_k)$ and $\nu(\pi_k, Q^*)$ by $\widetilde{\nu}_k$ and $\bar{\nu}_k$, respectively. Applying Lemma G.3 to (G.12) and utilizing the fact that $Q^* = T^{\pi^*} Q^*$, we have

$$Q^* - \widetilde{Q}_{k+1} \leq (Q^* - T^{\pi^*} \widetilde{Q}_k) + \varrho_{k+1} = (T^{\pi^*} Q^* - T^{\pi^*} \widetilde{Q}_k) + \varrho_{k+1}$$

$$\leq (T^{\pi^*, \widetilde{\nu}_k} Q^* - T^{\pi^*, \widetilde{\nu}_k} \widetilde{Q}_k) + \varrho_{k+1} = \gamma \cdot P^{\pi^*, \widetilde{\nu}_k} (Q^* - \widetilde{Q}_k) + \varrho_{k+1}, \qquad (\text{G.14})$$

where the last inequality follows from (G.13). Furthermore, for a lower bound of $Q^* - \widetilde{Q}_{k+1}$, similar to (F.5), we have

$$Q^* - \widetilde{Q}_{k+1} = (TQ^* - T^{\pi_k} Q^*) + (T^{\pi_k} Q^* - T^{\pi_k} \widetilde{Q}_k) + \varrho_{k+1}$$

$$\geq (T^{\pi_k} Q^* - T^{\pi_k} \widetilde{Q}_k) + \varrho_{k+1} \geq \gamma \cdot P^{\pi_k, \bar{\nu}_k} (Q^* - \widetilde{Q}_k) + \varrho_{k+1}, \qquad (\text{G.15})$$

where the both inequalities follow from Lemma G.3. Thus, combining (G.14) and (G.15) we have

$$\gamma \cdot P^{\pi^*, \widetilde{\nu}_k} (Q^* - \widetilde{Q}_k) + \varrho_{k+1} \geq Q^* - \widetilde{Q}_{k+1} \geq \gamma \cdot P^{\pi_k, \bar{\nu}_k} (Q^* - \widetilde{Q}_k) + \varrho_{k+1}. \qquad (\text{G.16})$$

for any $k \in \{0, \dots, K-1\}$. Similar to the proof of Lemma F.2 , by applying recursion to (G.16), we obtain the following upper and lower bounds for the error propagation of Algorithm 4.

**Lemma G.4** (Error Propagation). For any $k, \ell \in \{0, 1, \dots, K-1\}$ with $k < \ell$, we have

$$Q^* - \widetilde{Q}_\ell \leq \sum_{i=k}^{\ell-1} \gamma^{\ell-1-j} \cdot \big( P^{\pi^*, \widetilde{\nu}_{\ell-1}} P^{\pi^*, \widetilde{\nu}_{\ell-2}} \cdots P^{\pi^*, \widetilde{\nu}_{i+1}} \big) \varrho_{i+1}$$

$$+ \gamma^{\ell-k} \cdot \big( P^{\pi^*, \widetilde{\nu}_{\ell-1}} P^{\pi^*, \widetilde{\nu}_{\ell-2}} \cdots P^{\pi^*, \widetilde{\nu}_k} \big) (Q^* - \widetilde{Q}_k), \qquad (\text{G.17})$$

$$Q^* - \widetilde{Q}_\ell \geq \sum_{i=k}^{\ell-1} \gamma^{\ell-1-i} \cdot \big( P^{\pi_{\ell-1}, \bar{\nu}_{\ell-1}} P^{\pi_{\ell-2}, \bar{\nu}_{\ell-2}} \cdots P^{\pi_{i+1}, \bar{\nu}_{i+1}} \big) \varrho_{i+1}$$

$$+ \gamma^{\ell-k} \cdot \big( P^{\pi_{\ell-1}, \bar{\nu}_{\ell-1}} P^{\pi_{\ell-2}, \bar{\nu}_{\ell-2}} \cdots P^{\pi_{i+1}, \bar{\nu}_k} \big) (Q^* - \widetilde{Q}_k). \qquad (\text{G.18})$$

*Proof.* The desired results follows from applying the inequalities in (G.16) multiple times and the linearity of the operator $P^{\pi,\nu}$ for any joint policy $(\pi,\nu)$. □

The above lemma establishes recursive upper and lower bounds for the error terms $\{Q^* - \widetilde{Q}_k\}_{0 \le k \le K-1}$, which completes the first step of the proof.

**Step (ii):** In the second step, we characterize the suboptimality of the equilibrium policies constructed by Algorithm 4. Specifically, for each $\pi_k$, we denote by $Q_k^*$ the action-value function obtained when agent one follows $\pi_k$ while agent two adopt the best-response policy against $\pi_k$. In other words, $Q_k^*$ is the fixed point of Bellman operator $T^{\pi_k}$ defined in (G.10). In the following, we obtain an upper bound of $Q^* - Q_k^*$, which establishes the a notion of suboptimality of policy $(\pi_k, \nu_k)$ from the perspective of the first player.

To begin with, for any $k$, we first decompose $Q^* - Q_k^*$ by

$$Q^* - Q_k^* = \left(T^{\pi^*}Q^* - T^{\pi^*}\widetilde{Q}_k\right) + \left(T^{\pi^*}\widetilde{Q}_k - T^{\pi_k}\widetilde{Q}_k\right) + \left(T^{\pi_k}\widetilde{Q}_k - T^{\pi_k}Q_k^*\right). \tag{G.19}$$

Since $\pi_k$ is the equilibrium policy with respect to $\widetilde{Q}_k$, by Lemma G.3, we have $T^{\pi^*}\widetilde{Q}_k \le T^{\pi_k}\widetilde{Q}_k$. Recall that $(\pi^*, \nu^*)$ is the joint equilibrium policy with respect to $Q^*$. The second argument of Lemma G.3 implies that

$$T^{\pi^*}Q^* \le T^{\pi^*,\widetilde{\nu}_k}Q^*, \qquad T^{\pi_k}\widetilde{Q}_k \le T^{\pi_k,\widehat{\nu}_k}\widetilde{Q}_k, \tag{G.20}$$

where $\widetilde{\nu}_k = \nu(\pi^*, \widetilde{Q}_k)$ and we define $\widehat{\nu}_k = \nu(\pi_k, Q_k^*)$. Thus, combining (G.19) and (G.20) yields that

$$0 \le Q^* - Q_k^* \le \gamma \cdot P^{\pi^*,\widetilde{\nu}_k}\left(Q^* - \widetilde{Q}_k\right) + \gamma \cdot P^{\pi_k,\widehat{\nu}_k}\left(\widetilde{Q}_k - Q_k^*\right)$$
$$= \gamma \cdot \left(P^{\pi^*,\widetilde{\nu}_k} - P^{\pi_k,\widehat{\nu}_k}\right)\left(Q^* - \widetilde{Q}_k\right) + \gamma \cdot P^{\pi_k,\widehat{\nu}_k}\left(Q^* - Q_k^*\right). \tag{G.21}$$

Furthermore, since $I - \gamma \cdot P^{\pi_k,\widehat{\nu}_k}$ is invertible, by (G.21) we have

$$0 \le Q^* - Q_k^* \le \gamma \cdot \left(I - \gamma \cdot P^{\pi_k,\widehat{\nu}_k}\right)^{-1} \cdot \left[P^{\pi^*,\widetilde{\nu}_k}\left(Q^* - \widetilde{Q}_k\right) - P^{\pi_k,\widehat{\nu}_k}\left(Q^* - \widetilde{Q}_k\right)\right]. \tag{G.22}$$

Now we apply Lemma G.4 to the right-hand side of (G.22). Then for any $k \le \ell$, we have

$$P^{\pi^*,\widetilde{\nu}_\ell}\left(Q^* - \widetilde{Q}_\ell\right) \le \sum_{i=k}^{\ell-1} \gamma^{\ell-1-j} \cdot \left(P^{\pi^*,\widetilde{\nu}_\ell}P^{\pi^*,\widetilde{\nu}_{\ell-1}}\cdots P^{\pi^*,\widetilde{\nu}_{i+1}}\right)\varrho_{i+1}$$
$$+ \gamma^{\ell-k} \cdot \left(P^{\pi^*,\widetilde{\nu}_\ell}P^{\pi^*,\widetilde{\nu}_{\ell-1}}\cdots P^{\pi^*,\widetilde{\nu}_k}\right)\left(Q^* - \widetilde{Q}_k\right), \tag{G.23}$$

$$P^{\pi_\ell,\widehat{\nu}_\ell}\left(Q^* - \widetilde{Q}_\ell\right) \ge \sum_{i=k}^{\ell-1} \gamma^{\ell-1-i} \cdot \left(P^{\pi_\ell,\bar{\nu}_\ell}P^{\pi_\ell,\bar{\nu}_{\ell-2}}\cdots P^{\pi_{i+1},\bar{\nu}_{i+1}}\right)\varrho_{i+1}$$
$$+ \gamma^{\ell-k} \cdot \left(P^{\pi_\ell,\bar{\nu}_\ell}P^{\pi_{\ell-1},\bar{\nu}_{\ell-1}}\cdots P^{\pi_k,\bar{\nu}_k}\right)\left(Q^* - \widetilde{Q}_k\right). \tag{G.24}$$

Thus, setting $\ell = K$ and $k = 0$ in (G.23) and (G.24), we have

$$Q^* - Q_K^* \le \left(I - \gamma \cdot P^{\pi_K,\widehat{\nu}_K}\right)^{-1} \cdot \tag{G.25}$$

$$\left\{\sum_{i=0}^{K-1} \gamma^{K-1} \cdot \left[\left(P^{\pi^*,\widetilde{\nu}_K}P^{\pi^*,\widetilde{\nu}_{K-1}}\cdots P^{\pi^*,\widetilde{\nu}_{i+1}}\right) - \left(P^{\pi_K,\bar{\nu}_K}P^{\pi_{K-1},\bar{\nu}_{K-1}}\cdots P^{\pi_{i+1},\bar{\nu}_{i+1}}\right)\right]\varrho_{i+1}\right.$$

$$\left. + \gamma^{K+1} \cdot \left[\left(P^{\pi^*,\widetilde{\nu}_K}P^{\pi^*,\widetilde{\nu}_{K-1}}\cdots P^{\pi^*,\widetilde{\nu}_0}\right) - \left(P^{\pi_K,\bar{\nu}_K}P^{\pi_{K-1},\bar{\nu}_{K-1}}\cdots P^{\pi_0,\bar{\nu}_0}\right)\right]\left(Q^* - \widetilde{Q}_0\right)\right\}.$$

To simplify the notation, we define $\{\alpha_i\}_{i=0}^K$ as in (F.18). Note that we have $\sum_{i=0}^K \alpha_i = 1$ by definition. Moreover, we define $K + 1$ linear operators $\{O_k\}_{k=0}^K$ as follows. For any $i \le K - 1$, let

$$O_i = \frac{1-\gamma}{2} \cdot \left(I - \gamma \cdot P^{\pi_K,\widehat{\nu}_K}\right)^{-1}\left[\left(P^{\pi^*,\widetilde{\nu}_K}P^{\pi^*,\widetilde{\nu}_{K-1}}\cdots P^{\pi^*,\widetilde{\nu}_{i+1}}\right) - \left(P^{\pi_K,\bar{\nu}_K}P^{\pi_{K-1},\bar{\nu}_{K-1}}\cdots P^{\pi_{i+1},\bar{\nu}_{i+1}}\right)\right].$$

Moreover, we define $O_K$ by

$$O_K = \frac{1-\gamma}{2} \cdot (I - \gamma \cdot P^{\pi_K, \widehat{\nu}_K})^{-1} \Big[ \big( P^{\pi^*, \widetilde{\nu}_K} P^{\pi^*, \widetilde{\nu}_{K-1}} \cdots P^{\pi^*, \widetilde{\nu}_0} \big) - \big( P^{\pi_K, \bar{\nu}_K} P^{\pi_{K-1}, \bar{\nu}_{K-1}} \cdots P^{\pi_0, \bar{\nu}_0} \big) \Big].$$

Therefore, taking absolute values on both sides of (G.25), we obtain that

$$\big| Q^*(s,a,b) - Q_K^*(s,a,b) \big|$$
$$\leq \frac{2\gamma(1 - \gamma^{K+1})}{(1-\gamma)^2} \cdot \Bigg[ \sum_{i=0}^{K-1} \alpha_i \cdot \big( O_i | \varrho_{i+1} | \big)(s,a,b) + \alpha_K \cdot \big( O_K | Q^* - \widetilde{Q}_0 | \big)(s,a,b) \Bigg], \quad \text{(G.26)}$$

for any $(s,a,b) \in \mathcal{S} \times \mathcal{A} \times \mathcal{B}$, which concludes the second step of the proof.

**Step (iii):** We note that (G.26) is nearly the same as (F.19) for the MDP setting. Thus, in the last step, we follow the same proof strategy as in **Step (iii)** in §F.1. For notational simplicity, for any function $f \colon \mathcal{S} \times \mathcal{A} \times \mathcal{B} \to \mathbb{R}$ and any probability distribution $\mu \in \mathcal{P}(\mathcal{S} \times \mathcal{A} \times \mathcal{B})$, we denote the expectation of $f$ under $\mu$ by $\mu(f)$. By taking expectation with respect to $\mu$ in (G.26), we have

$$\| Q^* - Q_K^* \|_{1,\mu} \leq \frac{2\gamma(1 - \gamma^{K+1})}{(1-\gamma)^2} \cdot \Bigg[ \sum_{i=0}^{K-1} \alpha_i \cdot \mu \big( O_i | \varrho_{i+1} | \big) + \alpha_K \cdot \mu \big( O_K | Q^* - \widetilde{Q}_0 | \big) \Bigg]. \quad \text{(G.27)}$$

By the definition of $O_i$, we can write $\mu(O_i | \varrho_{i+1} |)$ as

$$\mu \big( O_i | \varrho_{i+1} | \big) = \frac{1-\gamma}{2} \cdot \mu \Bigg\{ \sum_{j=0}^{\infty} \gamma^j \cdot \Big[ \big( P^{\pi_K, \widehat{\nu}_K} \big)^j \big( P^{\pi^*, \widetilde{\nu}_K} P^{\pi^*, \widetilde{\nu}_{K-1}} \cdots P^{\pi^*, \widetilde{\nu}_{i+1}} \big) \qquad \text{(G.28)}$$
$$+ \big( P^{\pi_K, \widehat{\nu}_K} \big)^j \big( P^{\pi_K, \bar{\nu}_K} P^{\pi_{K-1}, \bar{\nu}_{K-1}} \cdots P^{\pi_{i+1}, \bar{\nu}_{i+1}} \big) \Big] | \varrho_{i+1} | \Bigg\}.$$

To upper bound the right-hand side of (G.28), we consider the following quantity

$$\mu \big[ P^{\tau_m} \cdots P^{\tau_1} ) f \big] = \int_{\mathcal{S} \times \mathcal{A} \times \mathcal{B}} \big[ (P^{\tau_m} P^{\tau_{m-1}} \cdots P^{\tau_1}) f \big](s,a,b) \mathrm{d}\mu(s,a,b),$$

where $\{ \tau_t \colon \mathcal{S} \to \mathcal{P}(\mathcal{A} \times \mathcal{B}) \}_{t \in [m]}$ are $m$ joint policies of the two-players. By Cauchy-Schwarz inequality, it holds that

$$\mu \big[ P^{\tau_m} \cdots P^{\tau_1} ) f \big] \leq \Bigg[ \int_{\mathcal{S} \times \mathcal{A} \times \mathcal{B}} \bigg| \frac{\mathrm{d}(P^{\tau_m} P^{\tau_{m-1}} \cdots P^{\tau_1} \mu)}{\mathrm{d}\sigma}(s,a,b) \bigg|^2 \mathrm{d}\sigma(s,a,b) \Bigg]^{1/2}$$
$$\cdot \Bigg[ \int_{\mathcal{S} \times \mathcal{A} \times \mathcal{B}} |f(s,a,b)|^2 \mathrm{d}\sigma(s,a,b) \Bigg]^{1/2} \leq \kappa(m; \mu, \sigma) \cdot \| f \|_\sigma,$$

where $\kappa(m; \mu, \sigma)$ is the $m$-th concentration parameter defined in (E.9). Thus, by (G.28) we have

$$\mu \big( O_i | \varrho_{i+1} | \big) \leq (1-\gamma) \cdot \sum_{j=0}^{\infty} \gamma^j \cdot \kappa(K - i + j; \mu, \nu) \cdot \| \varrho_{i+1} \|_\sigma. \quad \text{(G.29)}$$

Besides, since both $Q^*$ and $\widetilde{Q}_0$ are bounded by $R_{\max}/(1-\gamma)$ in $\ell_\infty$-norm, we have

$$\mu \big( O_K | Q^* - \widetilde{Q}_0 | \big) \leq 2 \cdot R_{\max}/(1-\gamma). \quad \text{(G.30)}$$

Finally, combining (G.27), (G.29), and (G.30), we obtain that

$$\| Q^* - Q^{\pi_K} \|_{1,\mu}$$
$$\leq \frac{2\gamma(1 - \gamma^{K+1})}{(1-\gamma)} \cdot \Bigg[ \sum_{i=0}^{K-1} \sum_{j=0}^{\infty} \alpha_i \cdot \gamma^j \cdot \kappa(K - i + j; \mu, \nu) \cdot \| \varrho_{i+1} \|_\sigma \Bigg] + \frac{4\gamma(1 - \gamma^{K+1})}{(1-\gamma)^3} \cdot \alpha_K \cdot R_{\max}$$
$$\leq \frac{2\gamma(1 - \gamma^{K+1})}{(1-\gamma)} \cdot \Bigg[ \sum_{i=0}^{K-1} \sum_{j=0}^{\infty} \frac{(1-\gamma)\gamma^{K-i-1}}{1 - \gamma^{K+1}} \cdot \gamma^j \cdot \kappa(K - i + j; \mu, \nu) \Bigg] \cdot \varepsilon_{\max} + \frac{4\gamma^{K+1}}{(1-\gamma)^2} \cdot R_{\max},$$

where the last inequality follows from the fact that $\varepsilon_{\max} = \max_{i\in[K]} \|\varrho_i\|_\sigma$. Note that in (F.29) we show that it holds under Assumption E.3 that

$$\sum_{i=0}^{K-1}\sum_{j=0}^{\infty} \frac{(1-\gamma)\gamma^{K-i-1}}{1-\gamma^{K+1}} \cdot \gamma^j \cdot \kappa(K-i+j;\mu,\nu) \leq \frac{\phi_{\mu,\nu}}{(1-\gamma^{K+1})(1-\gamma)}.$$

Hence, we obtain (G.1) and thus conclude the proof of Theorem G.1. □

