# OpenReview forum: "A Theoretical Analysis of  Deep Q-Learning"
_ICLR.cc/2020/Conference — Reject_

### Official Review · AnonReviewer2 · 2019-10-16
**Official Blind Review #2**

**Rating:** 3

**Review:**

This paper analyze the off-policy policy improvement algorithm with a very limited sparse ReLU function class. Although some of the results are interesting, I have lots of concern on the motivation of this work. I believe this paper is technically correct, but the authors focus on a very simplified case: they just use samples generated from a fixed sampling policy, which gets rid of the analysis of exploration and sample complexity that is a main focus of the reinforcement learning community. From my point of view, this may not be some analysis of reinforcement learning, at least not for Deep Q-learning, but more likely to be some learning theory of off-policy FQI. The main theorem investigates the statistical error and convergence rate of this problem, which can be of individual interest. But overall, I think the problem the authors want to solve is not a traditional reinforcement learning algorithm, and it is not appropriate to introduce the result as the theoretical analysis of Deep Q-Learning.

Detailed comments:
1. I think the assumption of Sparse ReLU network is too strong and generally not held in practice. Also, the optimization of such kind of network is painful, as the ell_0 constraint makes the optimization problem NP-hard. In other words, I think the authors only handle a very specific case under very ideal condition like assuming an oracles that can return the optimal network each turn.
2. The equivalence between FQI and target network is well-known and may not occupy so many places in Sec 3. Also, the results in Appendix B can be simply derived follows the recent development of neural network optimization. As this may be not the main contribution of this paper, I think it is better to omit these parts to make the paper more neat.
3. Moreover, in appendix B, the authors assumed the function class as two-layer ReLU network, which is different from the assumption in the main text and cannot justify the global convergence of (3.4).
4. It is somewhat strange of assume a sampling distribution, as when we say Q-learning, we want to balance the exploration and exploitation given current estimation Q. Even in Deep Q-Learning, the data are sampled with \epsilon-greedy policy w.r.t the current Q network. This kinds of problems are more like off-policy policy improvement. I think call it the analysis of Deep Q-Learning is somehow not accurate and over-claimed. Maybe better called off-policy policy improvement with deep neural networks.
5. Theorem 4.4 is an interesting result as it shows that the error of the proposed algorithm can be decomposed into the a statistical error which depends on the smoothness of the operator Tf and an algorithm error that depend on the number of iterations. I am wondering what's the main technical differences between this work and [1], as I find the main difference is [1] don't give K-dependent algorithm error, instead assuming K have a order of log 1/epsilon to ensure algorithm error is smaller than \epsilon. I feel it's not so hard to derive a bound that combines statistical error and algorithm error for [1]. Also, FVI in [1] is not in spirit totally different from FQI in this paper given that [1] use the maximum operator over action when do FVI, not take expectation over the target policy. I hope the authors can clarify in their paper.
6. The authors don’t mention much of the target network in the main theorem. I know the generalization to the update with target network is not so hard, but as the authors mentioned so much time in the main text, shall it be better to include the result with target network?

Still, in my opinion, the main theorem has its own value. However, it is not proper to claim as a theoretical analysis of Deep Q-Learning. Also, I feel the function class is too restricted and the optimization issue in the proposed algorithms cannot be simply solved, and the main analysis is similar to [1] with little generalization to Holder smoothness. Thus, I tend to reject this paper.

[1] Munos, Rémi, and Csaba Szepesvári. "Finite-time bounds for fitted value iteration." Journal of Machine Learning Research 9.May (2008): 815-857.

**Experience Assessment:**

I have read many papers in this area.

**Review Assessment: Checking Correctness Of Derivations And Theory:**

I assessed the sensibility of the derivations and theory.

**Review Assessment: Checking Correctness Of Experiments:**

N/A

**Review Assessment: Thoroughness In Paper Reading:**

I read the paper at least twice and used my best judgement in assessing the paper.

---

> ### Author Response · Authors · 2019-11-15
> **Response to Reviewer 2 regarding general comments**
>
>
> We appreciate the valuable comments from the reviewer. We first address the concern on the assumption of i.i.d. sampling from a fixed behavioural policy and then address each detailed comments separately.
>
>
> Sampling i.i.d. data from a behavioural policy:
>
> As also pointed out by Reviewer 4, the challenges of DQN involves exploration and generalization. In this work, we avoid the exploration problem by assuming sampling i .i.d. data from the behavioural policy and concentrability coefficients are bounded. We did not tackle exploration as provably efficient RL algorithms under the general function approximation setting remains an open problem. To study this problem, a standard metric is called ``regret'', and normally we need to modify the algorithm by constructing some ``optimistic value functions''. We believe that this is beyond the scope of this work. We only want to understand the vanilla version of DQN, which uses the $\epsilon$-greedy approach for exploration.
>
>  Here, our assumption of i.i.d. sampling from a behavioural policy is motivated by the common practice of having an extremely large memory buffer and sample i.i.d. data from the replay memory. Since the size of reply memory is huge, the sampling distribution of data changes very slowly as we update the replay memory. This empirical trick essentially aims to create i.i.d. data from sampling distribution, and is captured by our simplification of sampling i.i.d data from a fixed distribution.
>
> Besides, in DQN training, the target network is usually fixed for a long time with only the Q network updated by gradient descent. Then the target network is updated using the weights of the Q-network. This is essentially solving a regression problem with a fixed target network and use the Q network as the regressor. Then the learned Q network is used to update the target network. Thus we recover the FQI algorithm.
>
> Therefore, with a slight simplification of the tricks of experience replay and target network, we arrive at the FQI algorithm studied in our work. This motivates the study of our algorithm.
>
> Moreover, we tend not to agree with the reviewer on that "FQI is not a traditional reinforcement learning algorithm". In fact, FQI belongs to the family of batch reinforcement learning methods, which has lots of existing work. Batch RL is motivated by the fact that we would like to solve the reinforcement learning problem purely from historical data without the help of a simulator. Such a type of RL problems arises in applications such as recommendation system and prescription medicine, where it is challenging to run a trial-and-error algorithm or build a simulator. Please kindly find [Lange et al, 2012] for a survey and [Chen and Jiang, 2019] for recent understandings of the challenges of this problem.

---

> > ### Author Response · Authors · 2019-11-15
> > **Response to Reviewer 2 regarding detailed comments**
> >
> > Due to space limits, we address the detailed comments in a separate reply as follows.
> >
> > 1. (Sparse ReLU). We agree that finding the ERM solution in each iteration of FQI is computationally challenging. Here our focus is on the theoretical properties. We adopt sparse NNs because they are a family of universal function approximators, and we consider general MDPs with a weak assumption on the smooth of the transition.
> >
> > To alleviate the computational problem, we can instead focus on the family of overparametrized neural networks. However, it remains open that the representation power of this class of NNs. In other words, it might incur a large bias when applying the Bellman operator $\mathcal{T}$ to this function class, i.e., $\inf_{f\in \mathcal{F}} \sup_{g \in \mathcal{F} } \| f - Tg \|$ is large when $\mathcal{F}$ is the family of overparametrized NN. Nevertheless, we have also characterized the statistical error of FQI under this setting in Appendix B.
> >
> > 2.  We thank the reviewer's suggestion on the paper presentation. We write Section 3 in details to explain that reducing DQN to the version of FQI considered in our work is reasonable. The main message is that although DQN has the tricks of experience replay with a large memory size and target network that is fixed for a long time, the ideal version of this DQN reduces to our FQI, which motivates our analysis in Section 4.
> >
> > We will revise this section to highlight our motivation and also try to make the presentation neat.
> >
> > 3. We did not claim that the estimator in Appendix B solves the ERM in (3.4). Instead, we acknowledge that this problem is computationally intractable. In Appendix B, we would like to provide another neural FQI algorithm which can be computed. We also analyze the statistical error of this setting in Appendix B.
> >
> > 4. As mentioned previously, the reason that we called it deep Q-learning is because our algorithm is a reasonable simplification of the DQN algorithm. Our algorithm is fitted Q-iteration with deep neural networks. Note that sampling from a fixed distribution is a standard assumption in  FQI ([Munos and Szapesvari, 2008]). We have revised the abstract to make it clear that we focus on FQI.
> >
> > 5. The proof of Theorem 4.4 consists of three parts: 1) error propagation, which studies how the regression error in each iteration accumulates as the FQI algorithm proceeds, 2) the regression error in each iteration, and 3) balance the error terms to get the final statistical error.
> >
> > The general error propagation in our work essentially follows from the results in FVI ([Munos and Szapesvari, 2008]). However, the regression error analysis in the second part involves the particular structure of the Q-network. We need to analysis control the bias and variance separately. Moreover, in the last step, we explicit characterize the bias term of applying Bellman operator to the class of Q-networks $\mathcal{F}$, $\inf_{f\in \mathcal{F}} \sup_{g \in \mathcal{F} } \| f - Tg \|$. Thus, the last two steps of the analysis are specific to the deep neural networks and are not covered in [Munos and Szapesvari, 2008].
> >
> > 6. In our FQI algorithm, the target network is just the Q-network in the last iteration. That is, we fixed the previous Q-network as the target network and learn a new Q-network via regression using DNN. Then the new Q-network is used to replace the target network.
> >
> >
> > References:
> > [Lange et al, 2012] Batch Reinforcement Learning.  Sascha Lange, Thomas Gabel, and  Martin Riedmiller, 2012. \textit{https://link.springer.com/chapter/10.1007/978-3-642-27645-3_2}
> >
> > [Chen and Jiang, 2019] Information-Theoretic Considerations in Batch Reinforcement Learning. Jinglin Chen and Nan Jiang,  2019. \textit{https://arxiv.org/abs/1905.00360}
> >
> > [Barron and Klusowski, 2018] Approximation and Estimation for High-Dimensional Deep
> > Learning Networks. Andrew R. Barron and Jason M. Klusowski, 2018. \textit{https://arxiv.org/abs/1809.03090}
> >
> > [Munos and Szapesvari, 2008]  Finite-time bounds for fitted value iteration. Remi Munos and Csaba Szepesvari. Journal of Machine Learning Research. 2008

---

### Official Review · AnonReviewer1 · 2019-10-22
**Official Blind Review #1**

**Rating:** 8

**Review:**

The authors provide a theoretical analysis of deep Q-learning based on the neural fitted Q-iteration (FQI) algorithm [1]. Their analysis justifies the techniques of experience replay and target network, both of which are critical to the empirical success of DQN. Moreover, the authors establish the algorithmic and statistical errors of the neural FQI algorithm.
Then, the authors propose the Minimax-DQN algorithm for the zero-sum Markov game with two players. They further establish the algorithmic and statistical convergence rates of the sequence of action-value functions obtained by the Minimax-DQN algorithm.

[1] Martin Riedmiller. Neural fitted Q iteration–first experiences with a data efficient neural reinforcement learning method. In European Conference on Machine Learning, pp. 317–328. Springer, 2005.

The strengths of this paper are as follows.
1. This paper is theoretically sound. The authors establish the convergence rates with detailed proofs step by step.
2. It is the first theoretical analysis that provides the errors of the neural FQI algorithm with a ReLU network. This analysis provides a rigorous approach to understand deep q-learning algorithms.
3. The authors propose an extension of DQN for the zero-sum Markov game with two players. They further analyze the convergence rates of the sequence of action-value functions obtained by the proposed algorithm.

Minor comments:
1. Page 2: In Notation, "$\|f\|_{2,v}$" may be "$\|f\|_{v,2}$"。
2. Page 3: In the 2th line of Section 2.2, "$\{d_j\}_{i=0}^{L+1}$" may be "$\{d_i\}_{i=0}^{L+1}$".



**Experience Assessment:**

I have read many papers in this area.

**Review Assessment: Checking Correctness Of Derivations And Theory:**

I did not assess the derivations or theory.

**Review Assessment: Checking Correctness Of Experiments:**

N/A

**Review Assessment: Thoroughness In Paper Reading:**

I made a quick assessment of this paper.

---

> ### Author Response · Authors · 2019-11-15
> **Response to Reviewer 1**
>
>
> We thank the reviewer for appreciating this work and for pointing out the typos. We have addressed the issues raised by the other reviewers and revised our work accordingly.

---

### Official Review · AnonReviewer4 · 2019-11-05
**Official Blind Review #4**

**Rating:** 3

**Review:**

Overview:

This paper provides an analysis of fitted Q-iteration in the off-policy reinforcement learning setting, for the setup where the value function class of interest is a class of neural networks. The provide bounds on the rate at which fitted Q iteration converges to a near-optimal policy under the assumption that the transition dynamics satisfy a certain notion of Holder smoothness. This result is motivated by the problem of understanding why deep Q-learning works, which the authors relate to the problem above via certain simplifying assumptions. The authors also extend this result to give similar guarantees for two-player zero-sum stochastic games.

Review:

This paper addresses an important and challenging problem, and the results appear technical sound. It is also fairly thorough and well-written. However, my overall feelings toward the result are mixed, because I believe the assumptions the authors make are so strong that they essentially remove most of the interesting problem structure, and consequently the results follow by straightforward application of known techniques.

Two major challenges in understanding DQN are as follows:
1) Exploration: Why does the algorithm successfully explore and solve MDPs with large state spaces?
2) Generalization: How do the overparameterized neural networks used for value function approximation help with generalization (or exploration)?

The issue of exploration is assumed away by the authors, as they work in the batch/offline RL setting where examples (s,a,r,s') are i.i.d., and the assume that the so-called "concentrability coefficient", which measures mismatch between the data distribution and the data induced by the optimal policy, is bounded. This assumption is standard in the analysis of fitted Q-iteration for off-policy RL (eg, Munos and Szepesvari '08), but it implies that the algorithm does not need to solve a challenging exploration problem, since the data-gathering policy has good coverage. Unfortunately, the authors do not justify why this assumption should hold for DQN.

The standard analysis for off-policy fitted Q-iteration does not simply require that the concentratability coefficient is bounded, but also requires another strong assumption, which is that the function class is closed/complete under bellman updates. In general, this is a difficult property to verify, and it is well-known that fitted Q-iteration can cycle and fail to converge when it does not hold. This leads to the issue of generalization: The way the authors get around the issue of closedness/completeness is to work in the fully nonparametric regime: They take the class of neural nets under consideration to be large enough to approximate any Holder smooth function, then show that under mild assumptions on the dynamics this class of Holder smooth functions is closed under bellman updates. This is a good trick, but it has an unfortunate consequence, which is that by blowing up the class of neural networks, the generalization bound one can prove is quite weak. Ultimately, the generalization bound the authors give follows the standard rate for Holder-smooth functions in nonparametric statistics, which is exponential in dimension whenever the function class is $p$th order smooth for constant $p$. For example, when the class of functions is lipschitz the rate is $n^{-1/(2+d)}$, where $n$ is the number of examples. Since we are paying the fully nonparametric rate for generalization here, this begs the question of why neural nets were even used to begin with, which is not addressed.

To conclude, this is a certainly a challenging problem, but I don't think the paper is transparent about the limitations of the techniques (as described above) and I believe the title of the paper, "A theoretical analysis of deep Q-learning", is too strong given the shortcomings of the results.

**Experience Assessment:**

I have read many papers in this area.

**Review Assessment: Checking Correctness Of Derivations And Theory:**

I assessed the sensibility of the derivations and theory.

**Review Assessment: Checking Correctness Of Experiments:**

N/A

**Review Assessment: Thoroughness In Paper Reading:**

I read the paper at least twice and used my best judgement in assessing the paper.

---

> ### Author Response · Authors · 2019-11-15
> **Response to Reviewer 4**
>
>
> We appreciate the valuable comments from the reviewer. We first address the general concerns of the reviewer on the assumptions we made and then address each detailed comments separately.
>
> Assumptions regarding exploration and generalization:
>
> Our work aims to understand the empirical success of deep Q-learning from a theoretical perspective. To fully understand DQN, one indeed needs to tackle the challenges of exploration and generalization simultaneously. However, in this work, we pursue a more modest goal by only focusing on the generalization. Particularly, we view DQN as a sequence of fitted Q-iterations with neural networks. We aim to understand how the error incurred in each iteration of FQI affects the final generalization error (statistical error) of the DQN algorithm.
>
> Moreover, it is known that DQN adopts two tricks that have not been well understood -- i) experience replay and ii) target network. In practice, the memory size of experience replay is extremely large, and the target network is usually fixed for a large number of parameter updates of the Q network. As we show in Section 3, this motivates us to study the statistical error by looking into the problem of FQI.
>
>  Address the detailed comments:
>
>  1.  The assumption on i.i.d. data and concentrability coefficients.
>
>  As pointed out by the reviewer, we avoided the exploration problem by assuming sampling i .i.d. data from the behavioural policy and concentrability coefficients are bounded. We did not tackle exploration as provably efficient RL algorithms under the general function approximation setting remains an open problem. To study this problem, a standard metric is called ``regret'', and normally we need to modify the algorithm by constructing some ``optimistic value functions''. We believe that this is beyond the scope of this work. We only want to understand the vanilla version of DQN, which uses the $\epsilon$-greedy approach for exploration.
>
>  Here, our assumption of i.i.d. sampling from a behavioural policy is motivated by the common practice of having an extremely large memory buffer and sample i.i.d. data from the replay memory. Since the size of reply memory is huge, the sampling distribution of data changes very slowly as we update the replay memory. This empirical trick essentially aims to create i.i.d. data from sampling distribution, and is captured by our simplification of sampling i.i.d data from a fixed distribution.
>
>  In addition, we admit that the `` bounded concentrability coefficients'' is a technical assumption, which is used to capture the distributional shift caused by having different policies. As shown in [Chen and Jiang, 2019], concentrability is a necessary assumption for theoretical analysis. Moreover, they show that, even when the function class is closed under the Bellman operator, the reinforcement learning problem is computationally hard when the concentrability assumption is missing. Thus, we adopt this assumption for the aim of theoretical analysis. This assumption holds true when the transition kernel of the MDP has some nice properties. For hard exploration problems such that this assumption fails to hold, the hardness result in [Chen and Jiang, 2019] show that it is also not hopeful to solve efficiently using DQN.
>
>  2. Holder smooth assumption, nonparametric rate, and the usage of neural network.
>
>  As the reviewer has pointed out, we do not assume that the neural network function class is closed under the Bellman operator. Instead, we show that the target function is Holder smooth when the transition kernel satisfies certain smoothness conditions. Then we show that the neural network class yields a nonparametric rate of convergence.
>
>  It is true that the nonparametric rate can also be obtained by RKHS regression. Thus, this result does not exhibit the superiority of using neural networks. However, we adopt the deep neural network as the parametrization of the Q function of RL because our goal is to understand DQN. From that perspective, we show that, DQN roughly works as good as  FQI with other nonparametric regressors. Note that this type of theoretical guarantees of DQN is not known before.
>
>  Moreover, even in supervised learning, it seems that, without extra problem structures, the theoretical guarantees of deep learning is at most as good as kernels. Here we also assume Holder smooth for generality. Suppose we are willing to assume more problem structures, we can obtain a much faster $1/ \sqrt{n}$ rate. For example, suppose the Bellman operator is closed for the family of DNNs, by extending the analysis in [Barron and Klusowski, 2018], we obtain a  $\sqrt{L^3 \log d / n}$ rate, where $L$ is the number of layers and $d$ is the input dimension.

---

> > ### Author Response · Authors · 2019-11-15
> > **Response Continued (references)**
> >
> > Due to space limits, we list the references as follows.
> >
> > References:
> >
> > [Chen and Jiang, 2019] Information-Theoretic Considerations in Batch Reinforcement Learning. Jinglin Chen and Nan Jiang,  2019.
> >
> > [Barron and Klusowski, 2018] Approximation and Estimation for High-Dimensional Deep Learning Networks. Andrew R. Barron and Jason M. Klusowski, 2018.

---

### Author Response · Authors · 2019-11-15
**We appreciate the valuable reviews by the reviewers and have updated a revised version**

We appreciate the valuable reviews by the reviewers and the efforts the reviewers have dedicated. it seems that both Reviewers 2 and 4 are concerned with the motivation of our FQI algorithm. In the revised version, we clearly state in the abstract that the algorithm we consider is FQI with deep neural networks, which is a simplification of DQN that captures the features of experience replay and the target network. In the introduction and section 3, we explain in detail why such a simplification is reasonable.

---

### Decision · Program_Chairs · 2019-12-19

**Decision:**

Reject

**Comment:**

The authors offer theoretical guarantees for a simplified version of the deep Q-learning algorithm. However, the majority of the reviewers agree that the simplifying assumptions are so many that the results do not capture major important aspects of deep Q-Learning (e.g. understanding good exploration strategies, understanding why deep nets are better approximators and not using neural net classes that are so large that can capture all non-parametric functions). For justifying the paper to be called a theoretical analysis of deep Q-Learning some of these aspects need to be addressed, or the motivation/title of the paper needs to be re-defined.